# Cell state-specific cytoplasmic density controls spindle architecture and scaling

Tobias Kletter [1,2,9], Omar Muñoz[3,4,5], Sebastian Reusch[2], Abin Biswas [1,3,5], Aliaksandr Halavatyi [6], Beate Neumann [6], Benno Kuropka [7], Vasily Zaburdaev[3,4,5] & Simone Reber [1,2,8] ✉

Mitotic spindles are dynamically intertwined with the cytoplasm they assemble in. How the physicochemical properties of the cytoplasm affect spindle architecture and size remains largely unknown. Using quantitative biochemistry in combination with adaptive feedback microscopy, we investigated mitotic cell and spindle morphology during neural differentiation of embryonic stem cells. While tubulin biochemistry and microtubule dynamics remained unchanged, spindles changed their scaling behaviour; in differentiating cells, spindles were considerably smaller than those in equally sized undifferentiated stem cells. Integrating quantitative phase imaging, biophysical perturbations and theory, we found that as cells differentiated, their cytoplasm became more dilute. The concomitant decrease in free tubulin activated CPAP (centrosomal P4.1-associated protein) to enhance the centrosomal nucleation capacity. As a consequence, in differentiating cells, microtubule mass shifted towards spindle poles at the expense of the spindle bulk, explaining the differentiation-associated switch in spindle architecture. This study shows that cell state-specific cytoplasmic density tunes mitotic spindle architecture. Thus, we reveal physical properties of the cytoplasm as a major determinant in organelle size control.

The mitotic spindle is a prime example to study subcellular organizing principles and link physical laws to organelle function[1]. While spindles showcase a staggering morphological diversity, their primary function remains a mechanical one: to precisely partition the genetic material into two daughter cells[2]. Generally, spindles show two scaling regimes adapting to changes in cell size. In small cells, spindle size scales linearly with cell size and in large cells, the size of the spindle becomes decoupled from cell size and reaches an upper limit. Such key observations of spindle scaling mostly stem from fast reductive divisions in early mouse, frog, fish and worm embryos[3–10]. However, later in development, cellular differentiation poses a similar challenge: differentiating cells change size, morphology and function and therefore must reorganize

and adapt their internal architecture. Changing spindle architecture while maintaining spindle function poses a structural and mechanical challenge to the cell and may require adaptation of the organizing principles[11]. Whether and how mitotic spindles adjust their morphology in differentiating cells remains largely unknown.

While spindles can function in a variety of cells across orders of magnitude of cell sizes, they all assemble mainly from one building block: αβ-tubulin. Tubulin assembles into dynamic microtubules that self-organize with the help of microtubule-associated proteins (MAPs) and motors into a spindle with a steady-state length. How can more or less identical building blocks assemble spindles with varying sizes and architectures? Our current understanding is that small spindles

[1]Max Planck Institute for Infection Biology, Berlin, Germany. [2]IRI Life Sciences, Humboldt-Universität zu Berlin, Berlin, Germany. [3]Max-Planck-Zentrum für Physik und Medizin, Erlangen, Germany. [4]Friedrich-Alexander-Universität Erlangen-Nürnberg (FAU), Erlangen, Germany. [5]Max Planck Institute for the Science of Light, Erlangen, Germany. [6]Advanced Light Microscopy Facility, EMBL Heidelberg, Heidelberg, Germany. [7]Freie Universität Berlin, Core Facility BioSupraMol, Berlin, Germany. [8]Berliner Hochschule für Technik, Berlin, Germany. [9]Present address: i3S - Instituto de Investigação e Inovação em Saúde, Universidade do Porto, Porto, Portugal. ✉e-mail: reber@mpiib-berlin.mpg.de

modulate microtubule dynamics to adjust spindle size, whereas large spindles require additional microtubule nucleation[9,12,13]. Spindle microtubule nucleation and dynamics can be modulated in many ways, by changes in tubulin biochemistry[14], activity of MAPs and motors[8,15–17], by the presence of a limiting component[18,19] or changes in the biochemical composition of the cytoplasm[6,9,20]. In the context of differentiation, adjustments to spindle architecture have been described[21,22]. Whether these architectural adjustments occur via modified microtubule dynamics or spatial regulation of microtubule nucleation is currently unknown. We therefore lack a systematic understanding how microscopic processes collectively give rise to a mesoscale spindle that is adaptive to the differentiating cellular context.

It is increasingly appreciated that spindles are dynamically intertwined with the cytoplasm they assemble in. However, mechanisms of how cytoplasmic properties affect spindle assembly and function are only starting to emerge. For example, it has been shown that cytoplasmic viscosity affects the rates of microtubule polymerization and depolymerization[23], spindle function[24] and spindle positioning[25,26]. The mixing of the nucleoplasm and cytoplasm upon mitotic entry affects free tubulin concentration globally and consequently microtubule dynamics[27]. At the same time, local enrichment of tubulin by organelle-exclusion zones[28,29] or at centrosomes[30,31] spatially supports microtubule nucleation and growth. In addition, the mitotic cytoplasm softens[32] and is fluidized by microtubule polymerization dynamics[33]. Both changes are thought to maintain diffusivity through the heterogenous metaphase cytoplasm. Although differentiation-mediated changes in cytoplasmic properties have been observed[34–40], we do not know how these changes directly affect spindle assembly and scaling.

Here, we use a cell culture model of neural differentiation to study changes in spindle architecture, while cells change size, morphology and function. Studying neural spindle scaling was motivated by the observation that neurodevelopment is particularly susceptible to mutations in spindle genes, and by pathological evidence that links spindle morphology and mitotic susceptibility[22,41–43]. To do so, we established a noninvasive, long-term image acquisition and analysis workflow that allowed us to image more than 4,000 cells at single-cell resolution throughout neural differentiation. We find that spindle size subscales with cell volume; in early-differentiated cells spindles are 24% smaller when compared with spindles in equally sized undifferentiated stem cells, but still follow a similar functional dependence on cell size. Despite this difference in size, tubulin biochemistry and microtubule dynamics remain unchanged. We show that cytoplasmic dilution and a drop in tubulin concentration increase centrosomal nucleation capacity. This is dependent on centrosomal P4.1-associated protein (CPAP), a centrosomal regulator[44–46]. As a consequence, microtubule growth is redistributed to the spindle poles, resulting in a shift in spindle architecture. Consistently, differentiation-independent cytoplasmic dilution or the specific inhibition of the CPAP–tubulin interaction in undifferentiated embryonic stem cells (ESCs) phenocopied spindle architecture and size characteristic of early-differentiated cells. Our data are consistent with a theory in which microtubule number scales with cell volume and the inhibition of a centrosomal regulator determines the distribution between astral and bulk microtubules. Ultimately, our study links cell state-specific cytoplasmic material properties to spindle architecture and size. More generally, this work shows how local intracellular environments can exert control over organelle morphology and scaling.

## Results

### Spindle size subscales with cell volume upon differentiation

Mechanistic insights into spindle scaling stem primarily from fast, reductive cell divisions during early animal development[3,4,8,9,13]. It thus remains largely unknown whether differentiating cells tune spindle size relative to cell size and if so, whether the observed physicochemical mechanisms apply to this cellular context. This question is particularly

relevant in neurally differentiating cells where spindle defects have documented pathological implications[41]. An experimentally accessible model is the differentiation of neurons from mouse ESCs in adherent cell culture[47]. To quantitatively study spindle scaling, we differentiated ESCs that stably expressed tubulin::GFP towards the neural lineage (Fig. 1a). This allowed us to image cells from pluripotency until terminal differentiation at single-cell resolution on a single substrate and in large numbers. We found that within 48–72 h of neural induction, the differentiating cells downregulated the expression of the pluripotency marker OCT-4 and started expressing nestin (Fig. 1a–d) and PAX6 (Extended Data Fig. 1a), consistent with previous reports[47,48]. Notably, we observed the formation of neural rosettes (Supplementary Video 1), which showed hallmarks of neuroepithelial physiology in terms of marker expression, cell polarity and interkinetic nuclear migration[49,50] (Fig. 1a, Extended Data Fig. 1b–g and Supplementary Video 2). We concluded that this system represented a suitable model to study spindle morphology in a differentiation context.

Using this model system, we designed an experimental and analytical methodology to track individual cell families through differentiation while simultaneously detecting and quantifying spindle morphology. First, to overcome the challenge of generating sufficient quantitative three-dimensional (3D) imaging data of metaphase cells with minimal phototoxicity, we developed an adaptive feedback microscopy pipeline for live-cell imaging (Fig. 1e and Methods). Throughout the differentiation process, the fully autonomous microscope located metaphase cells within the larger population and launched high-content recordings exclusively of cells of interest (Fig. 1f). Next, using Spindle3D[51] and Ilastik[52], we extracted 3D morphometric parameters, including spindle volume, width, pole-to-pole length, cell volume and cell surface area from 1,084 undifferentiated and 2,920 differentiating mitotic cells (Fig. 1g,h and Extended Data Fig. 2a–d). To obtain the volumetric relationship between spindle size and cell size, we pixel-classified, segmented and validated cell volume through cytoplasmic tubulin::GFP. To segment spindles in 3D, the Spindle3D pipeline robustly finds a critical signal threshold value above which tubulin voxels are incorporated into the spindle bulk mask (Extended Data Fig. 2a). Thus, spindle volume serves as a proxy for spindle microtubule mass. Within the differentiating population, the average volume of mitotic cells decreased by 30% when compared with undifferentiated ESCs ($V_{ESC} = 2,719 \pm 567\ \mu m^3$ and $V_{DIF} = 1,942 \pm 424\ \mu m^3$; Fig. 1h). This was independent of cell confluency and medium conditions (Extended Data Fig. 2e,f). Spindle volume decreased and scaled with cell volume ($r_s = 0.8$; Fig. 1h). However, spindles in the early-differentiating cells (up to 5 days of differentiation) occupied significantly less cell volume than spindles in undifferentiated ESCs (Extended Data Fig. 2g). This became even more apparent when we binned cell volumes: in cells with comparable cell volume, the median spindle volumes were up to 24% smaller in cells undergoing differentiation (Fig. 1i). This subscaling behaviour was independent of cell geometry or microtubule density within the spindle (Extended Data Fig. 2h–j). Thus, spindle size is not specified solely by cell volume. Taken together, we established a noninvasive, long-term image acquisition and analysis workflow, which allowed us to quantify cell and spindle morphologies over many generations to show that spindle volume subscales in early-differentiating cells. This meant that cell volume cannot be the only determinant of spindle size. Rather, our findings implied a cell state-specific spindle scaling mechanism that was responsive to the biochemical or physical changes imposed by cellular differentiation.

### Spindle architecture switches upon differentiation

Cells can change spindle size by changing the balance between microtubule nucleation, growth and turnover, which is known to alter microtubule polymer mass and spindle volume[9,53,54]. To test whether spindles in neurally differentiating cells change their size because of altered microtubule dynamics, we measured spindle microtubule turnover via fluorescence recovery after photobleaching (FRAP) measurements

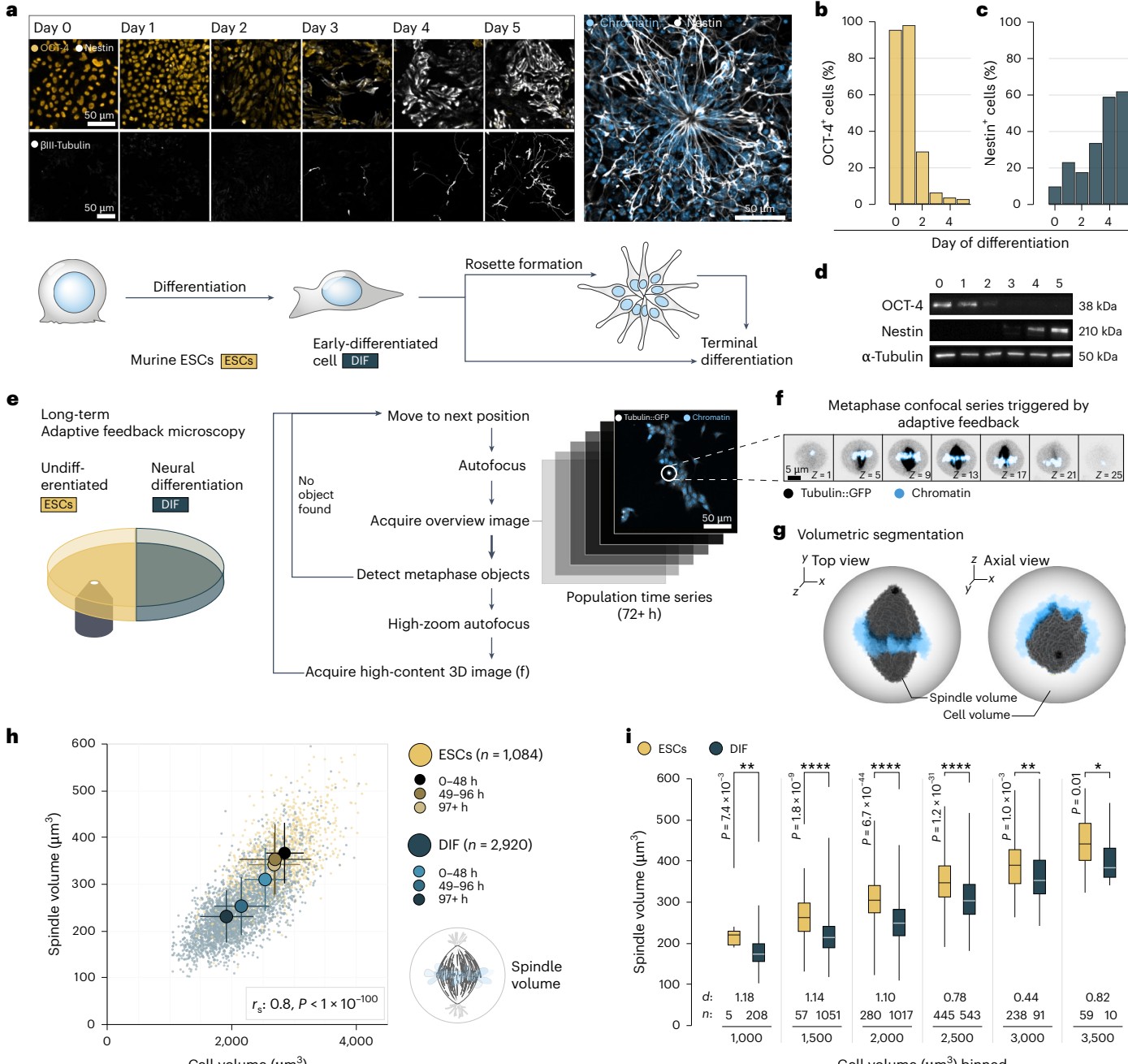

**Fig. 1 | Spindle size subscales with cell volume upon differentiation.**
**a**, Immunofluorescence staining of mouse ESCs driven towards neural fates in adherent monolayer. Every 24 h, a replicate culture was co-stained for OCT-4[88] (yellow, top row) and nestin (grey, top row) or stained for βIII-tubulin (grey, bottom row) (left). Neural rosette after 6 days of differentiation stained with Hoechst (blue) and against nestin (grey) (right). All images are maximum projections. Scale bars, 50 μm. Illustration showing the differentiation of ESCs towards neural progenitors (bottom). **b**, Percentage of OCT-4-positive cells (as in **a**), covering the first 5 days of neural differentiation (day 1, $n = 418$ cells; day 2, $n = 1,316$ cells; day 3, $n = 1,199$ cells; day 4, $n = 2,666$ cells; day 5, $n = 3,976$ cells from one experiment). **c**, As in **b** but showing percentage of nestin-positive cells (same experiment as **b**). **d**, Immunoblots probing for differentiation markers on a series of cell lysates ($n = 1$ experiment), covering 5 days of neural differentiation. α-Tubulin as a loading control. **e**, Automated microscopy setup. ESCs were either kept undifferentiated ('ESCs') or driven towards neural differentiation ('DIF'). Adaptive feedback microscopy pipeline for live-cell imaging with high-content confocal series of metaphase cells expressing tubulin::GFP and chromatin stained with SiR-DNA. Scale bar, 50 μm. **f**, Confocal raw high-resolution data of a metaphase cell ($n = 9$ experiments) as described above. Tubulin::GFP (inverted grey) and SiR-DNA (blue). Scale bar, 5 μm. **g**, 3D-rendered metaphase spindles (grey) after automated volumetric segmentation and morphometry using the Fiji plugin Spindle3D[51] and the pixel-classification tool Ilastik[52]. Chromosomes are shown in blue. The cell volume is illustrated as a cartoon for clarity.
**h**, Spindle volumes scale with cell volumes during differentiation. Each data point represents an individual cell (ESCs $n = 1,084$ (yellow) and DIF $n = 2,920$ (blue)). Big circles represent the mean of each differentiation time bin. Error bars show the s.d. Data are pooled from nine independent experiments. $r_s$: Spearman's correlation, $P < 1 \times 10^{-100}$. **i**, In equally sized cells, spindle volume subscales in DIF (blue) when compared with ESCs (yellow). $n$ (ESCs, DIF) = 5, 208; 57, 1,051; 280, 1,017; 445, 543; 238, 91; 59, 10 cells binned from 1,000 μm³ through 3,500 μm³ in 500-μm³ bins, with the same experiments as in **h**. White lines inside boxes denote medians, boxes show interquartile ranges, whiskers show minima and maxima. $d$, Cohen's $d$ (effect size). Welch's $t$-test (two-sided) per bin. ****$P < 0.0001$; **$P < 0.01$; *$P < 0.05$.

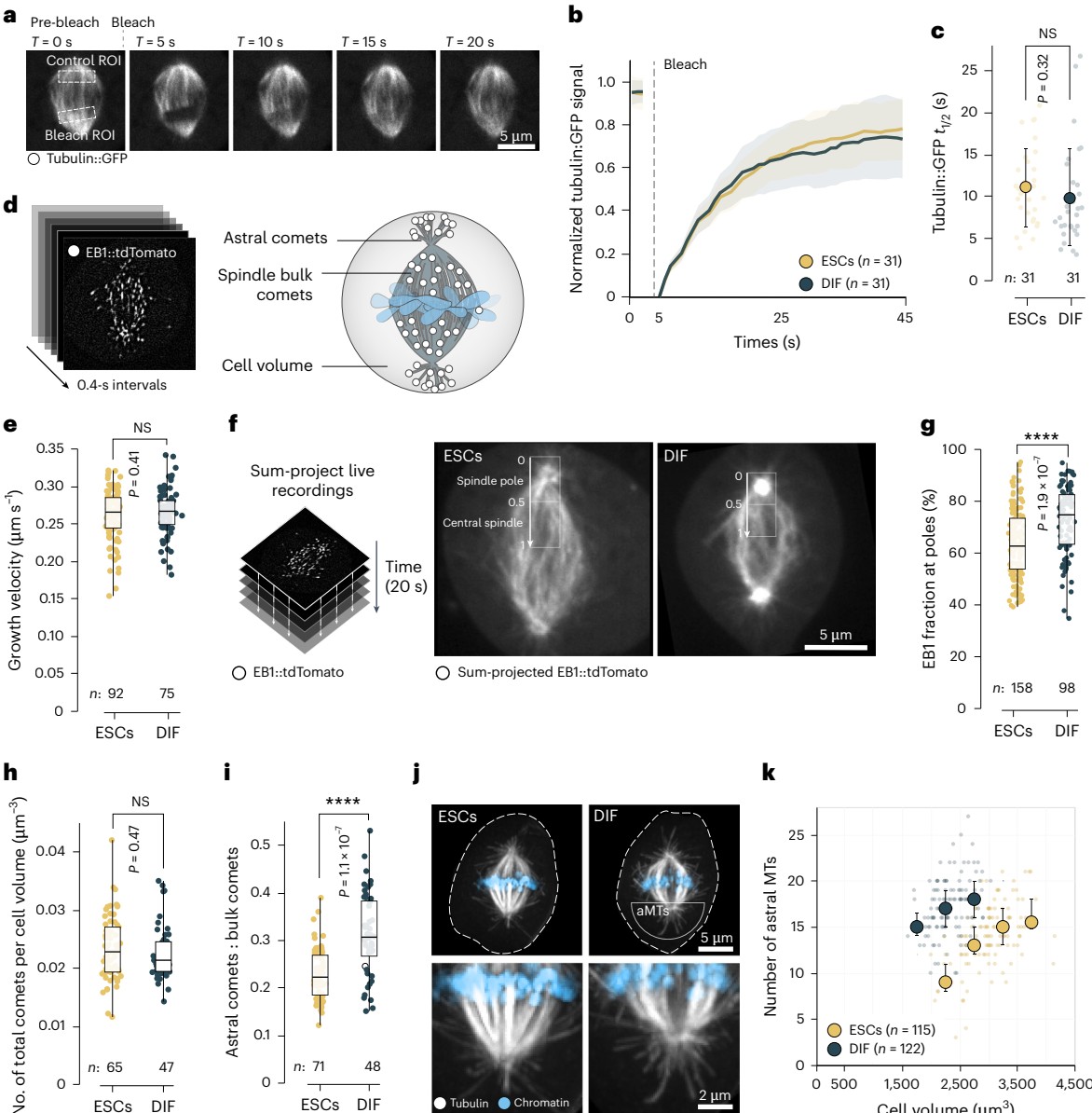

**Fig. 2 | Spindles switch to pronounced astral architectures in early-differentiated cells. a**, FRAP (*n* = 2 experiments) of tubulin::GFP turnover in spindles. Selected frames pre-bleach and post-bleach are shown. Scale bar, 5 μm. **b**, Normalized FRAP recovery curves from **a**, lines show the mean (ESCs *n* = 31 cells, DIF *n* = 31 cells pooled from two independent experiments), bands show the s.d. **c**, Recovery half-times of tubulin::GFP derived from **b**, large circles show means, error bars indicate s.d., small circles show individual cells. Welch's *t*-test (two-sided), *P* = 0.32. **d**, Mitotic cells expressing EB1::tdTomato imaged live in 0.4-s intervals (left) to determine growth speed and distribution of growing microtubules before and after differentiation (right). **e**, Average growth speed of EB1::tdTomato-labelled microtubules. Data points show individual cells (ESCs *n* = 92 cells, DIF *n* = 75 cells pooled from 6 independent experiments). Boxes show interquartile ranges, black lines inside boxes denote medians, whiskers show minima and maxima. Welch's *t*-test (two-sided), *P* = 0.41. **f**, EB1::tdTomato videos (20 s) were sum-projected (left). Half-spindle sum intensity profiles were drawn and subdivided into spindle poles (normalized distance 0–0.5) and central spindle (0.5–1) (right). **g**, Percentage of summed up EB1::tdTomato signal at the spindle pole (normalized half-spindle distance 0–0.5). Data points show individual cells (ESCs *n* = 158 cells, DIF *n* = 98 cells pooled from six independent

experiments). Boxes show interquartile ranges, black lines inside boxes denote medians, whiskers show minima and maxima. Welch's *t*-test (two-sided) *P* = 1.9 × 10⁻⁷. **h**, Total number of EB1 comets per unit cell volume. Data points show ratio in individual cells (ESCs *n* = 65, DIF *n* = 47 from six independent experiments). Boxes show interquartile ranges, black lines inside boxes denote medians, whiskers show minima and maxima. Welch's *t*-test (two-sided), *P* = 0.47. **i**, Ratio of astral and spindle bulk EB1 comets. Data points show individual cells (ESCs *n* = 71 cells, DIF *n* = 48 cells pooled from six independent experiments). Boxes show interquartile ranges, black lines inside boxes denote medians, whiskers show minima and maxima. Welch's *t*-test (two-sided), *P* = 1.1 × 10⁻⁷. **j**, Max-projected micrographs showing immunostained microtubules (grey). Chromatin counterstained by Hoechst (blue). Dotted lines show cell boundaries. Scale bar, 5 μm. aMTs, astral microtubules. Bottom row, zoomed-in details (Scale bar, 2 μm). **k**, Differentiating cells increase their number of astral microtubules (as determined in **f**). Each data point represents a single cell (ESCs *n* = 115 cells, DIF *n* = 122 cells pooled from three independent experiments), large circles denote medians in each cell volume bin, error bars show interquartile ranges. ****P* < 0.0001; NS, not significant, *P* > 0.05.

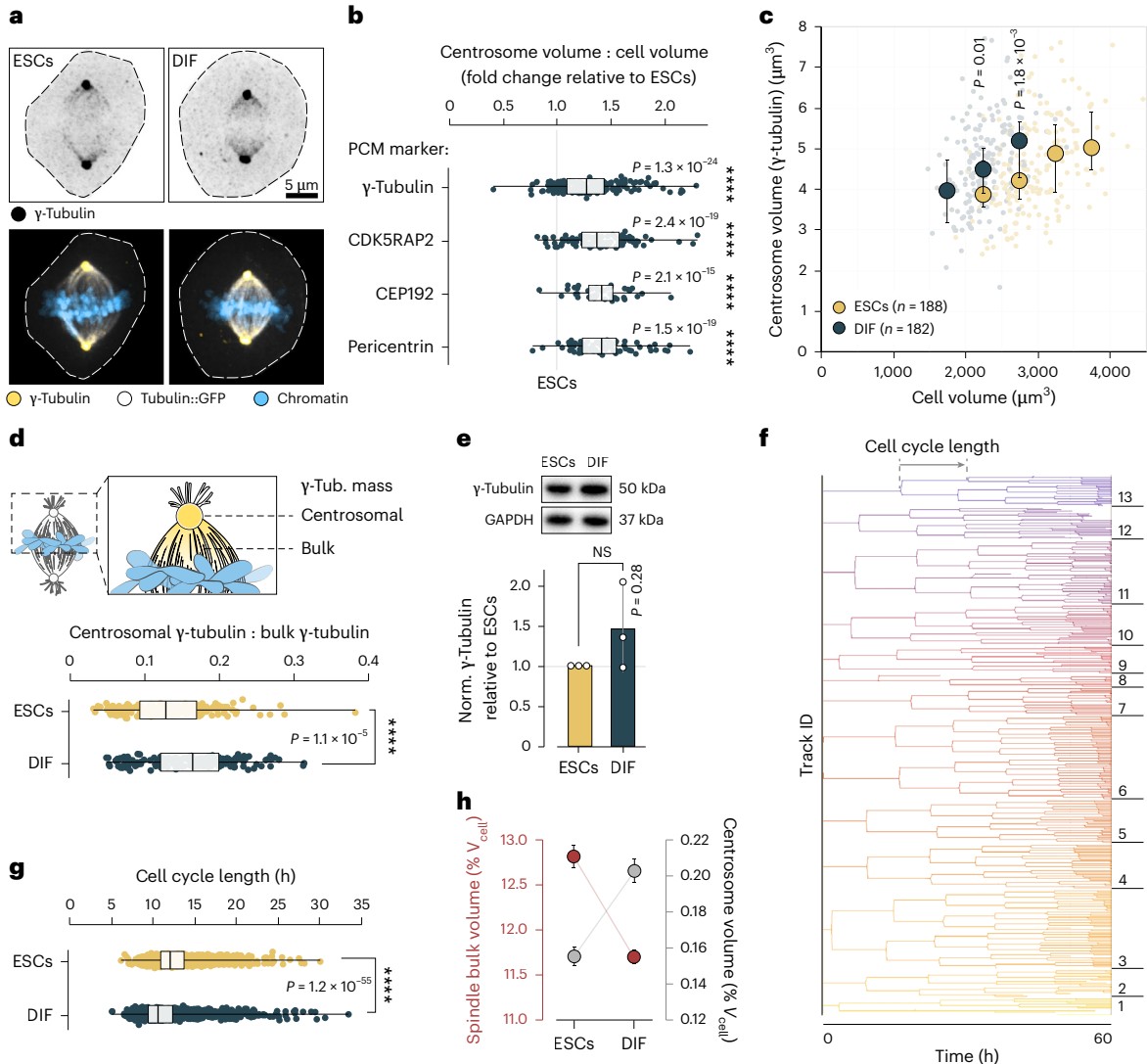

**Fig. 3 | Centrosomes superscale upon differentiation. a**, Confocal micrographs (maximum projected, representative of $n = 6$ experiments) of fixed ESCs or DIF at metaphase. Immunostained γ-tubulin signal (inverted grayscale) (top), immunostained γ-tubulin (yellow), tubulin::GFP (grey) and chromatin (Hoechst, blue) (bottom). Dotted lines indicate cell boundaries. Scale bar, 5 μm. **b**, Fold change in centrosome occupancy (centrosome volume:cell volume) in DIF relative to ESCs, comparing four centrosomal markers. Data points represent individual cells, boxes show interquartile ranges, vertical lines show medians and whiskers show the minima and maxima. $n = 188, 182; 113, 100; 56, 44; 89, 73$ cells (ESCs, DIF) stained for γ-tubulin, CDK5RAP2, CEP192 and pericentrin, each from six, three, one and three independent experiments, respectively. Significances against ESCs were tested with Welch's $t$-tests (two-sided; γ-tubulin, $P = 1.3 \times 10^{-24}$; CDK5RAP2, $P = 2.4 \times 10^{-19}$; CEP192, $P = 2.1 \times 10^{-15}$; pericentrin, $P = 1.3 \times 10^{-24}$). **c**, Centrosome volume (γ-tubulin) scales with cell volume. Each data point represents a single cell (ESCs $n = 188$ cells and DIF $n = 182$ cells pooled from six independent experiments), large circles denote medians in each cell volume bin (bin size = 500 μm³), error bars show the interquartile ranges. Statistics per cell volume bin by Welch's $t$-test (two-sided; 2,000–2,500 μm³, $P = 0.01$; 2,500–3,000 μm³, $P = 1.8 \times 10^{-3}$). **d**, γ-Tubulin localization (top). Centrosomal to bulk γ-tubulin mass ratio in immunostained cells (see **a**) (bottom). Data

points represent individual cells (sample sizes as in **c**), boxes show interquartile ranges, vertical lines show medians and whiskers show the minima and maxima. Welch's $t$-test (two-sided) $P = 1.1 \times 10^{-5}$. **e**, Top: representative western blots against γ-tubulin in ESCs or DIF. GAPDH as loading control. γ-Tubulin (normalized to GAPDH), relative fold change (bottom). Bars show mean ($n = 3$ biological replicates), error bars show s.d., circles show replicates. Welch's $t$-test (two-sided), $P = 0.28$. **f**, Cell cycle lengths determined by tracking of individual cell families. Visual representation of tracking data of an exemplary adaptive feedback recording (Fig. 1) of ESCs. **g**, Cell cycle length changes upon differentiation[89,90] Intermitotic times from time-lapse recordings (Fig. 1) of ESCs (yellow) or DIF (blue). Data points represent individual cells ($n = 2,614$ and 2,230 from ESCs and DIF, respectively, pooled from nine independent replicates). Boxes show interquartile ranges, black lines inside boxes denote medians, whiskers show the minima and maxima. Welch's $t$-test (two-sided), $P = 1.2 \times 10^{-55}$. **h**, As spindles subscale, centrosomes superscale upon differentiation. Comparing spindle bulk scaling to cell volume (ESCs $n = 1,084$ cells and DIF $n = 2,920$ cells pooled from nine independent experiments) and centrosome scaling to cell volume (data as in **c**) between ESCs and DIF. Circles show means, error bars show 95% confidence intervals. ****$P < 0.0001$, NS, not significant, $P > 0.05$.

(Fig. 2a,b). We found the half-time of microtubule recovery in both spindle types to be comparable: $11.1 \pm 4.6$ s in stem cells and $9.7$ s $\pm 5.8$ s in differentiating cells (Fig. 2c). To measure microtubule growth velocities, we used the plus end-tracking protein EB1::tdTomato (Fig. 2d). We acquired time-lapse recordings of undifferentiated ESCs and

early-differentiated cells and tracked growing microtubule plus ends (Supplementary Video 3). We found growth velocity to be independent of spindle size and mitotic cell volume (Extended Data Fig. 3a) and identical between the differentiation states ($v_{p\_ESC} = 0.26 \pm 0.034$ μm s⁻¹ and $v_{p\_DIF} = 0.26 \pm 0.031$ μm s⁻¹; Fig. 2e). Consistently, the cellular levels

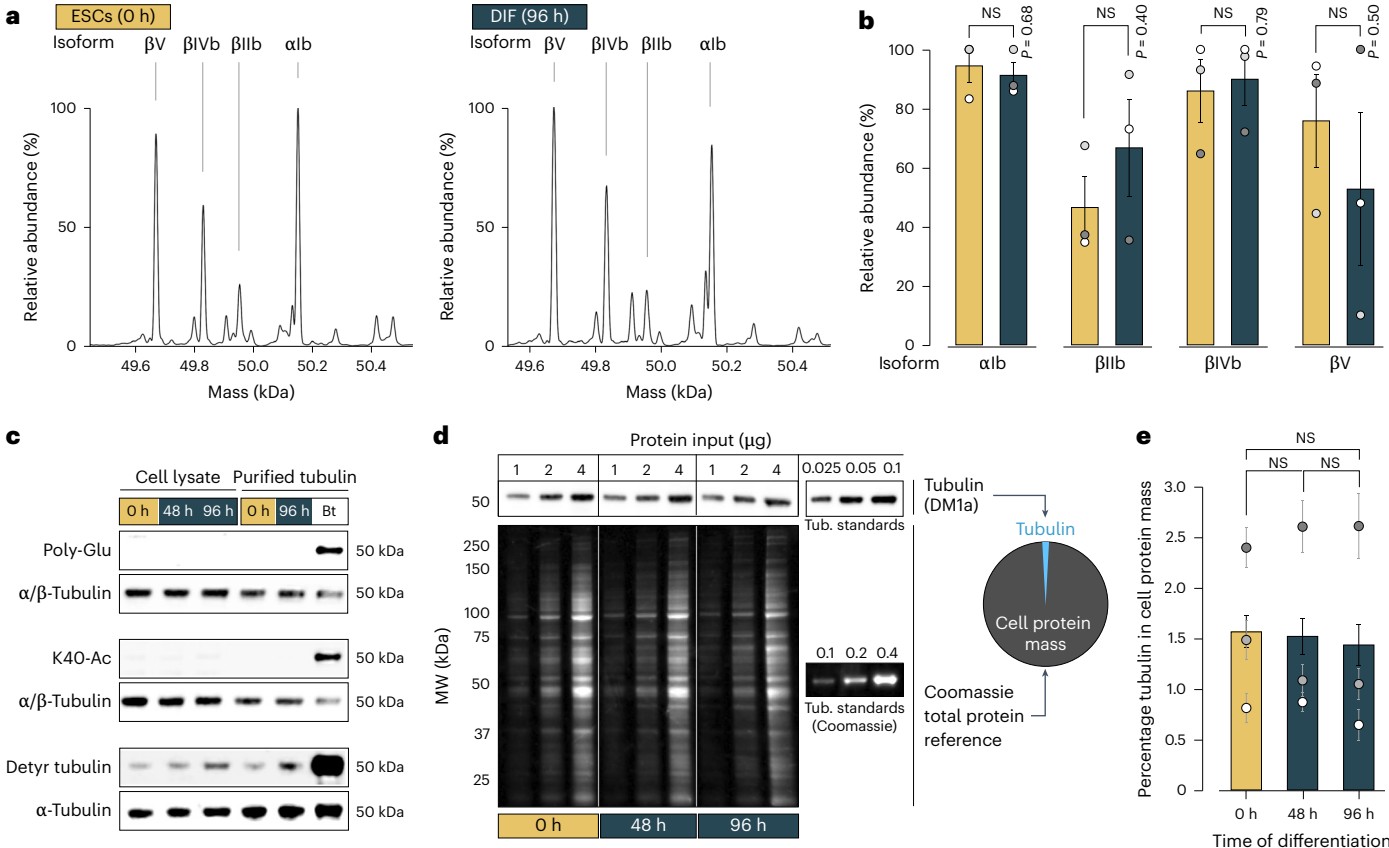

**Fig. 4 | Constant tubulin biochemistry between the differentiation states.**
**a**, Deconvoluted mass spectra of tubulins purified from ESCs (left) or DIF (right). Average masses of the most abundant signals and corresponding tubulin isoforms, βV (MW = 49,670.3 Da, UniProt P99024), βIVb (MW = 49,830.5 Da, UniProt P68372), βIIb (MW = 49,952.6 Da, UniProt Q9CWF2), αIb (MW = 50,151.1 Da, UniProt P05213). **b**, Mean relative abundances of the four most dominant isoforms purified from ESCs versus 96 h DIF (*n* = 3 biological replicates) measured by intact protein mass spectrometry. Error bars show s.e.m. Circles show replicates. Welch's *t*-test (two-sided), αIb: *P* = 0.68, βIIb: *P* = 0.40, βIVb: *P* = 0.79, βV: *P* = 0.50. **c**, Tubulin PTMs are comparable between differentiation states. Western blots using whole-cell lysates and purified tubulin (*n* = 2 biological replicates). Affinity-purified *Bos taurus* (Bt) brain tubulin loaded

as positive control. Poly-Glu, poly-glutamylation; K40-Ac, tubulin lysine 40 acetylation; Detyr, detyrosinated tubulin. **d**, Representative tubulin western blot of three differentiation time points and whole-cell lysates (*n* = 3 biological replicates) (top). For downstream calibration, defined masses of purified tubulin were blotted onto the same membrane (three technical replicates per batch of lysates). Coomassie-stained whole protein content on a replica gel (bottom). **e**, Tubulin consistently represents 1.5% of the cellular protein mass. Relative tubulin content in total cellular protein mass in whole-cell lysates of ECSs (0 h) versus 48 h or 96 h DIF cells. Bars show mean ± s.e.m. of *n* = 3 biological replicates. Circles show mean ± s.e.m. of three technical replicates per experiment. One-way analysis of variance (ANOVA), F-statistic = 0.1263, *P* = 0.88. NS, not significant, *P* > 0.05.

of the major microtubule polymerases CKAP5, which belongs to the TOG/XMAP215 family[55], and CKAP2 (ref. 56) remained comparable (Extended Data Fig. 3b–e). Thus, despite the difference in spindle size, the microtubule growth dynamics and turnover remained unchanged.

However, we found a considerable switch in spindle architecture: at 48 h of differentiation, microtubules increasingly grew from spindle poles but less in the spindle bulk (Fig. 2f,g and Extended Data Fig. 3a). This led to a shift in the ratio of astral to spindle bulk microtubules at a constant number of microtubules per cell volume (Fig. 2h,i). To complement the live data, we additionally visualized microtubule populations via 3D immunofluorescence (Fig. 2j), which confirmed that the number of astral microtubules was significantly higher in early-differentiated cells when compared with undifferentiated ESCs of equal cell volumes (Fig. 2k). Notably, in ESCs the number of astral microtubules seemed to saturate at a critical cell volume above ~3,000 μm³ (Fig. 2k). Together, these data indicated that the observed cell state-specific spindle scaling resulted from a redistribution of microtubules towards the asters away from the spindle bulk.

The switch in spindle architecture is analogous to observations in the developing mouse neocortex, where early neurogenic spindles displayed pronounced astral microtubules[22]. In this system, TPX2, a

MAP that stabilizes microtubules and stimulates augmin-mediated microtubule nucleation[57–59], has been identified as a main contributor of spindle morphology switches. In our system, however, neither cellular TPX2 nor augmin levels, nor their localization on spindles changed between undifferentiated ESCs and early-differentiated cells (Extended Data Fig. 4). Thus, we concluded that the switch in spindle architecture was not a consequence of diminished TPX2- or augmin-loading upon differentiation. Alternatively, microtubule number could be modulated by microtubule severing[60–62]. While total cellular concentrations of spastin and katanin p60 and p80 did not or only slightly change between stem cells and differentiating cells, we found katanin p80 to be enriched on spindle poles of stem cells (Extended Data Fig. 5a–g). However, while katanin knockdown did produce aberrant spindle phenotypes with buckled microtubules within the bulk[63] (Extended Data Fig. 5h–k), it had no effect on spindle scaling (Extended Data Fig. 5l,m), ruling out microtubule severing as the main factor of spindle subscaling in our system.

Taken together, these data suggested that differentiating cells changed their spindle architecture and size by redistributing microtubule growth to the spindle poles. This resulted in smaller spindles relative to cell size when comparing differentiating and undifferentiated cells.

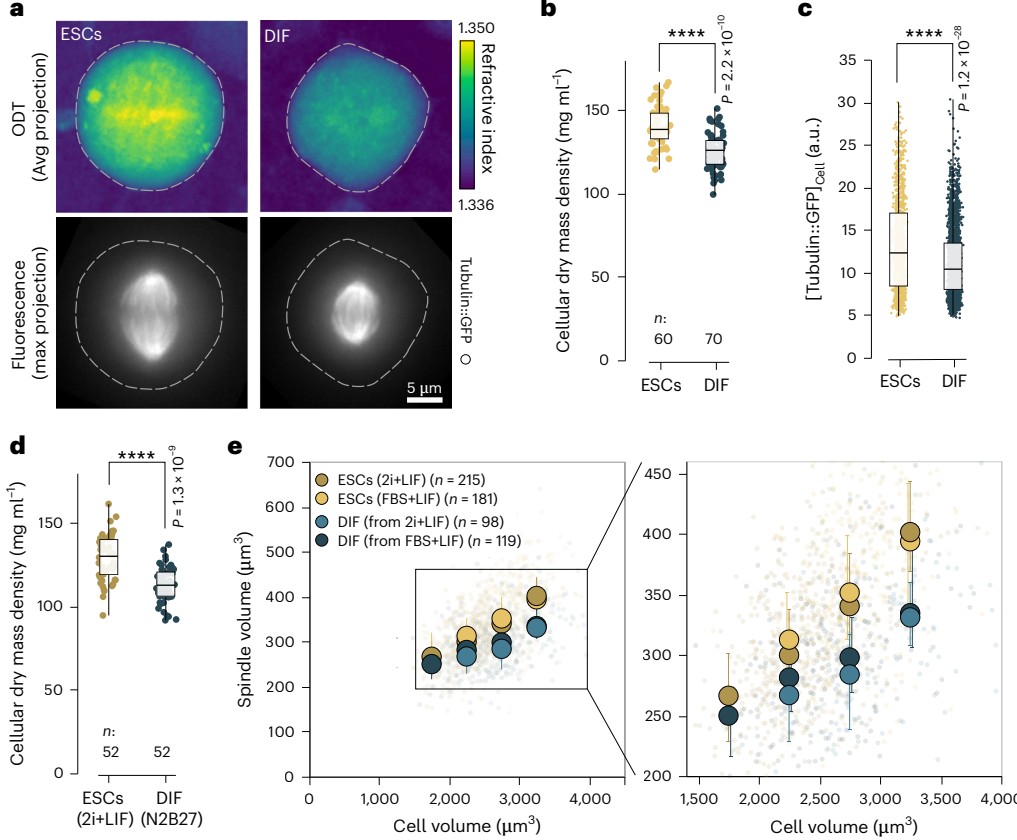

**Fig. 5 | The cytoplasm is diluted in early-differentiating cells. a**, Average (Avg) z-projections of 3D refractive index (RI) maps derived from optical diffraction tomography of mitotic ESCs or DIF (top). Colour-coded according to RI. Maximum-projected epi-fluorescent micrographs showing the tubulin::GFP signal (bottom). Dotted lines show cell boundaries. Representative images from $n = 3$ experiments. Scale bars, 5 µm. **b**, Cellular mass density decreases upon differentiation. 3D cellular mass densities of mitotic ESCs versus DIF. Data points show individual cells (ESCs $n = 60$ cells and DIF $n = 70$ cells pooled from three independent experiments). Boxes show interquartile ranges, black lines inside boxes denote medians, whiskers show the minima and maxima. Welch's *t*-test (two-sided), $P = 2.2 \times 10^{-10}$. **c**, Average tubulin::GFP signal (3D total cell) decreases during differentiation. Data from long-term automated imaging (Fig. 1). Boxes show interquartile ranges, black lines inside boxes denote medians, whiskers

show the minima and maxima. Data points show individual cells (ESCs $n = 1,084$ cells and DIF $n = 2,920$ cells pooled from nine independent experiments). Welch's *t*-test (two-sided), $P = 1.2 \times 10^{-28}$. **d**, As in **b** but showing cellular mass density of DIF from 2i + LIF ESCs cultures (ESCs $n = 52$ cells and DIF $n = 52$ cells pooled from two independent experiments). Welch's *t*-test (two-sided), $P = 1.3 \times 10^{-9}$. **e**, Spindle volume subscaling is differentiation intrinsic. Comparing spindle bulk volume scaling to cell volume in DIF originating from 2i + LIF ESCs cultures or from FBS + LIF ESCs cultures, as determined by confocal live-cell imaging (left). Data points show individual cells (ESCs (2i + LIF) $n = 215$, ESCs (FBS + LIF) $n = 181$, DIF (from 2i + LIF) $n = 98$, DIF (from FBS + LIF) $n = 215$, cells pooled from five independent experiments). Large circles represent medians in 500 µm³ cell volume bins. Error bars show interquartile ranges. Zoomed-in detail (right). ****$P < 0.0001$.

## Centrosomes superscale in early-differentiated cells

The apparent increase in nucleation capacity at spindle poles could simply be a result of increased centrosome size[64]. Centrosomes are the major microtubule nucleating centres in mammalian cells and important for mitotic spindle assembly, asymmetric cell division, and cell polarity[65,66]. Therefore, we directly visualized and measured centrosomes by staining γ-tubulin (Fig. 3a) and components of the pericentriolar material (PCM), that is CDK5RAP2, CEP192 and pericentrin (Extended Data Fig. 6a–c). We found that centrosomes superscaled upon differentiation, where the relative volume occupied by centrosomal proteins increased by approximately 1.4-fold in early-differentiated cells when compared with the undifferentiated stem cells (Fig. 3b). Centrosome volume scaled with cell volume for both cell states (Fig. 3c), as has been observed in early *Caenorhabditis elegans* and Zebrafish embryogenesis[4,67,68]. However, centrosomes of undifferentiated ESCs became independent of cell size and approached an upper limit in cells with volumes above ~3,000 µm³ (Fig. 3c and Extended Data Fig. 6d–f). Indeed, in early-differentiated cells, more γ-tubulin was recruited to the centrosomes when compared with stem cells (Fig. 3d), while overall γ-tubulin levels did not change (Fig. 3e). The relative increase in

centrosome size paralleled the increase in nucleation capacity and astral microtubule number in early-differentiated cells (Fig. 2k). This suggested a direct relation between centrosome volume and nucleation capacity, as has been reported previously[64].

Commonly, to prepare for mitosis, microtubule remodelling starts with a dramatic increase in microtubule nucleation at the centrosomes driven by the recruitment and local activation of γ-TuRC[64,69]. Our long-term imaging strategy allowed us to track individual cells across several generations (Fig. 3f). This showed that cells classified as early-differentiated cells had on average divided three to four times (Fig. 3g) and thus dynamically remodelled their PCM. Taken together, these data suggested that the relative increase in centrosome size shifts the microtubule nucleation capacities towards the spindle poles at the expense of the spindle bulk (Fig. 3h) leading to smaller spindles with larger asters in early-differentiated cells.

## Constant tubulin biochemistry between the differentiation states

Another way of changing spindle architecture in differentiating cells could be through fundamental changes in tubulin biochemistry.

Therefore, we measured tubulin isoform composition, tubulin post-translational modifications (PTMs) and tubulin levels. To determine tubulin isoforms and PTMs, we purified tubulin from either ESCs or early-differentiated cells[70]. Intact protein mass spectrometry revealed no significant changes in isoform composition between day 0 and 4 of differentiation (Fig. 4a,b). Considerable βIII-tubulin expression (as neuronal marker) started only with terminal differentiation later than day 4 (Fig. 1a) as reported previously[47]. Similarly, there were no differences between the PTM patterns in ESCs and early-differentiated cells (Fig. 4c). Together, this showed that neither tubulin isoform composition nor tubulin PTMs change during early differentiation. Next, we measured cellular tubulin levels by quantitative immunoblots (Fig. 4d and Methods). For each time point during differentiation, tubulin constituted approximately 1.5% of total cellular protein mass (Fig. 4e). From these data we concluded that changes in spindle morphology were not driven by tubulin biochemistry, and thus other spindle scaling mechanisms must be operating.

### The cytoplasm is diluted in early-differentiating cells

Differentiation-induced changes in cell function have been correlated with reductions in cellular mass density in a variety of cellular systems[34–39]. To test whether in our system cells change their mass density (defined as dry mass/volume; Methods)[29] during differentiation, we measured subcellular distributions of refractive indices using correlative fluorescence and optical diffraction tomography[71]. Notably, we found that cellular mass density in early-differentiated cells was reduced by 10% relative to the undifferentiated ESCs ($\rho_{ESC} = 140 \pm 12$ mg ml$^{-1}$ versus $\rho_{DIF} = 125 \pm 11$ mg ml$^{-1}$, Fig. 5a,b). This differentiation-associated dilution of the cytoplasm was observed across all cell volumes (Extended Data Fig. 7). Consistently, we measured a reduction in total cellular (spindle and cytoplasmic) tubulin::GFP fluorescence using live-cell imaging (Fig. 5c). This indicated a drop of total tubulin concentration concomitant with cytoplasmic dilution. While the reduction in total tubulin reduced spindle volume, it did not lead to a reduction in microtubule density within the spindle bulk

(Extended Data Figs. 2j and 3a). Notably, both the observed decrease in cellular mass density as well as changes in spindle scaling were independent of culturing and differentiation conditions (Fig. 5d,e). Taken together, these data imply that changes in cellular mass density could have consequences for spindle architecture and scaling during differentiation.

### Cytoplasmic dilution shifts spindle architecture by increasing centrosomal nucleation capacity

The above data suggested that tuning cellular mass density could contribute to changes in spindle volume. If cytoplasmic dilution altered spindle morphology universally, we would expect a differentiation-independent drop in cellular mass density to affect spindle size in undifferentiated ESCs. To test this, we diluted the cytoplasm of stem cells by lowering the osmolality of the culturing medium (Fig. 6a,b). As expected, a decrease in osmolality led to an increase in cell volume and a reduction in cellular mass density ($\rho_{ISO} = 118 \pm 10$ mg ml$^{-1}$ versus $\rho_{HYPO} = 112 \pm 9$ mg ml$^{-1}$; Fig. 6c,d). This drop in cellular mass density led to an increase in γ-tubulin at the centrosomes (Fig. 6e,f), consequently an increase in centrosomal EB1 concentration (Fig. 6g) and higher numbers of astral microtubules (Fig. 6h,i). Taken together, these data confirmed that a reduction in cellular mass density is sufficient to increase the centrosome's nucleation capacity at the expense of the spindle bulk (Fig. 6j), resulting in a reduced spindle volume with constant microtubule density. Indeed, we were able to gradually adjust spindle scaling and architecture in stem cells by modulating cellular mass densities (Extended Data Figs. 8a–f). In the same manner, we rescued spindle subscaling in differentiating cells by concentrating the cytoplasm by hyperosmotic challenges (Extended Data Fig. 8g–j). These data imply that the spindle assembly mechanism is intertwined with the physical properties of the cytoplasm. But how is this implemented at the molecular level?

So far, there is no established link between cytoplasmic mass density and centrosome function. However, it has been reported that centrosomes can increase their nucleation capacity when free

**Fig. 6 | Cytoplasmic dilution shifts spindle architecture by increasing centrosomal nucleation capacity. a**, Osmotic perturbation of ESCs (i) and liberating CPAP from its inhibitory binding to tubulin in ESCs (ii). **b**, Average (Avg) z-projections of 3D RI maps derived from optical diffraction tomography (ODT) imaging of mitotic ESCs after adding isotonic medium (Iso, 337 mOsmol kg$^{-1}$) or 25% ultrapure water (Hypo, 250 mOsmol kg$^{-1}$) (top). Colour-coded according to RI. Maximum-projected epi-fluorescent micrographs showing tubulin::GFP signal (bottom). Dotted lines show cell boundaries. Scale bars, 5 μm. **c**, ODT-derived cell volumes after hypo-osmotic treatment of ESCs. Boxes show interquartile ranges, black lines inside boxes denote medians, whiskers show the minima and maxima. Data points show individual cells (*n* isosmotic: 107, *n* hypo-osmotic: 96 cells pooled from four independent experiments). Welch's *t*-test (two-sided), *P* = 0.02. **d**, As **c** but showing mitotic cellular mass density. *P* = 6.2 × 10$^{-6}$. **e**, Maximum-projected confocal micrographs showing immunostained γ-tubulin signals (yellow, or inverted grey (cropped images), respectively) in fixed ESCs, tubulin::GFP in grey, chromatin in blue. Cells after hypo-osmotic challenge ('Hypo') versus control ('Iso') (top), cells after CCB02 treatment versus control (dimethylsulfoxide 'DMSO') (bottom). Scale bars, 5 μm. **f**, Fraction of total γ-tubulin signals residing at centrosomes, comparing iso- versus hypo-osmotically treated ESCs (*n* = 71 and 60 cells from iso- and hypo-osmotically treated ESCs, respectively, pooled from two independent experiments, normalized to isosmotic control) and comparing DMSO- versus CCB02-treated ESCs (*n* = 109 and 121 cells from DMSO- and CCB02-treated ESCs, respectively, pooled from three independent experiments, normalized to DMSO control) and comparing ESCs (*n* = 188 cells) with DIF (*n* = 182 cells) (pooled from six independent experiments (Fig. 3b), normalized to ESCs). Boxes show interquartile ranges, black lines inside boxes denote medians, whiskers show the minima and maxima. Welch's *t*-test (two-sided), Iso versus Hypo, *P* = 2.7 × 10$^{-4}$; DMSO versus CCB02, *P* = 1.4 × 10$^{-5}$; ESCs versus DIF, *P* = 2.9 × 10$^{-8}$. **g**, Left: Sum-projected videos (20 s) of mitotic ESCs expressing EB1::tdTomato,

after iso- or hypo-osmotic treatment (top), or after treatment with CCB02 or DMSO (bottom). Rectangle indicates region for sum intensity profiles, subdivided into spindle pole and central spindle. Boxplots as in **f** but showing fraction of EB1::tdTomato sum at spindle poles (right). Iso- (*n* = 53 cells) or hypo-osmotically treated (*n* = 67 cells) ESCs (pooled from *n* = 2 independent experiments), and of ESCs after CCB02 treatment (*n* = 40 cells) (versus DMSO control, *n* = 49 cells) (pooled from two independent experiments) and ESCs (*n* = 158 cells) or DIF (*n* = 98 cells) (same as Fig. 2g, pooled from six independent experiments). Iso versus Hypo, *P* = 7.2 × 10$^{-4}$; DMSO versus CCB02, *P* = 8.7 × 10$^{-4}$; ESCs versus DIF, *P* = 2.0 × 10$^{-7}$. **h**, Confocal micrographs (max-projected) of metaphase control ('Iso') ESCs or after hypo-osmotic treatment ('Hypo') (top row) or CCB02- or DMSO-treated ESCs (bottom row), stained with anti-tubulin antibodies (grey). Chromatin in blue. Dotted lines show cell boundaries. Scale bar, 5 μm. **i**, Boxplots as in **f** but showing number of astral microtubules, comparing iso- (*n* = 37 cells) versus hypo-osmotically treated ESCs (*n* = 37 cells) (pooled from two independent experiments, normalized to isosmotic control), and comparing DMSO- (*n* = 54 cells) versus CCB02-treated (*n* = 70 cells) ESCs (pooled from two independent experiments, normalized to DMSO control), and comparing ESCs (*n* = 115 cells) with DIF (*n* = 122 cells) (pooled from three independent experiments (same data as Fig. 2k), normalized to ESCs). Iso versus Hypo, *P* = 1.1 × 10$^{-7}$; DMSO versus CCB02, *P* = 1.1 × 10$^{-6}$; ESCs versus DIF, *P* = 2.5 × 10$^{-12}$. **j**, Boxplots as in **f** but showing percentage of cell volume occupied by spindle, comparing iso- (*n* = 67 cells) versus hypo-osmotically treated ESCs (*n* = 71 cells) (pooled from *n* = 2 experiments), and comparing DMSO- (*n* = 136 cells) versus CCB02-treated (*n* = 119 cells) ESCs (pooled from three independent experiments), and comparing ESCs (*n* = 181 cells) with DIF (*n* = 119 cells) (pooled from five independent experiments (data as Fig. 5e)). Iso versus Hypo, *P* = 1.3 × 10$^{-5}$; DMSO versus CCB02, *P* = 3.8 × 10$^{-20}$; ESCs versus DIF, *P* = 2.3 × 10$^{-6}$. *\*P < 0.05; \*\*\*P < 0.001; \*\*\*\*P < 0.0001.*

cytoplasmic tubulin decreases. This is because of the inhibitory interaction of soluble tubulin with the centrosomal protein CPAP[72]. We speculated that the observed decrease in free tubulin liberated CPAP, which in turn increased PCM recruitment and centrosomal microtubule nucleation. To mimic a decrease in free tubulin in stem cells, we liberated CPAP from tubulin using a small molecule inhibitor (CCB02 (ref. [73]); Fig. 6a). This should result in an increase in astral microtubule number and a decrease in spindle bulk without changing cell size. Indeed, upon inhibitor treatment, spindles in undifferentiated ESCs partitioned more γ-tubulin to the centrosomes (Fig. 6e,f), which showed an increase in nucleation capacity as evidenced by a higher

centrosomal EB1 concentration (Fig. 6g) and astral microtubule number (Fig. 6h,i), leading to a reduction in spindle volume in equally sized cells (Fig. 6j). Thus, we propose that CPAP is a molecular determinant of centrosomal nucleation capacity that is responsive to cytoplasmic dilution. Together, our physical perturbation via cytoplasmic dilution and biochemical perturbation using a small molecule inhibitor allowed us to phenocopy the spindle architecture characteristic of early-differentiated cell states in undifferentiated stem cells.

Notably, our data are consistent with quantitative models of spindle scaling[9,18,53]. In the Supplementary Note, we provide a theoretical model that allows us to link cell volume, spindle volume and

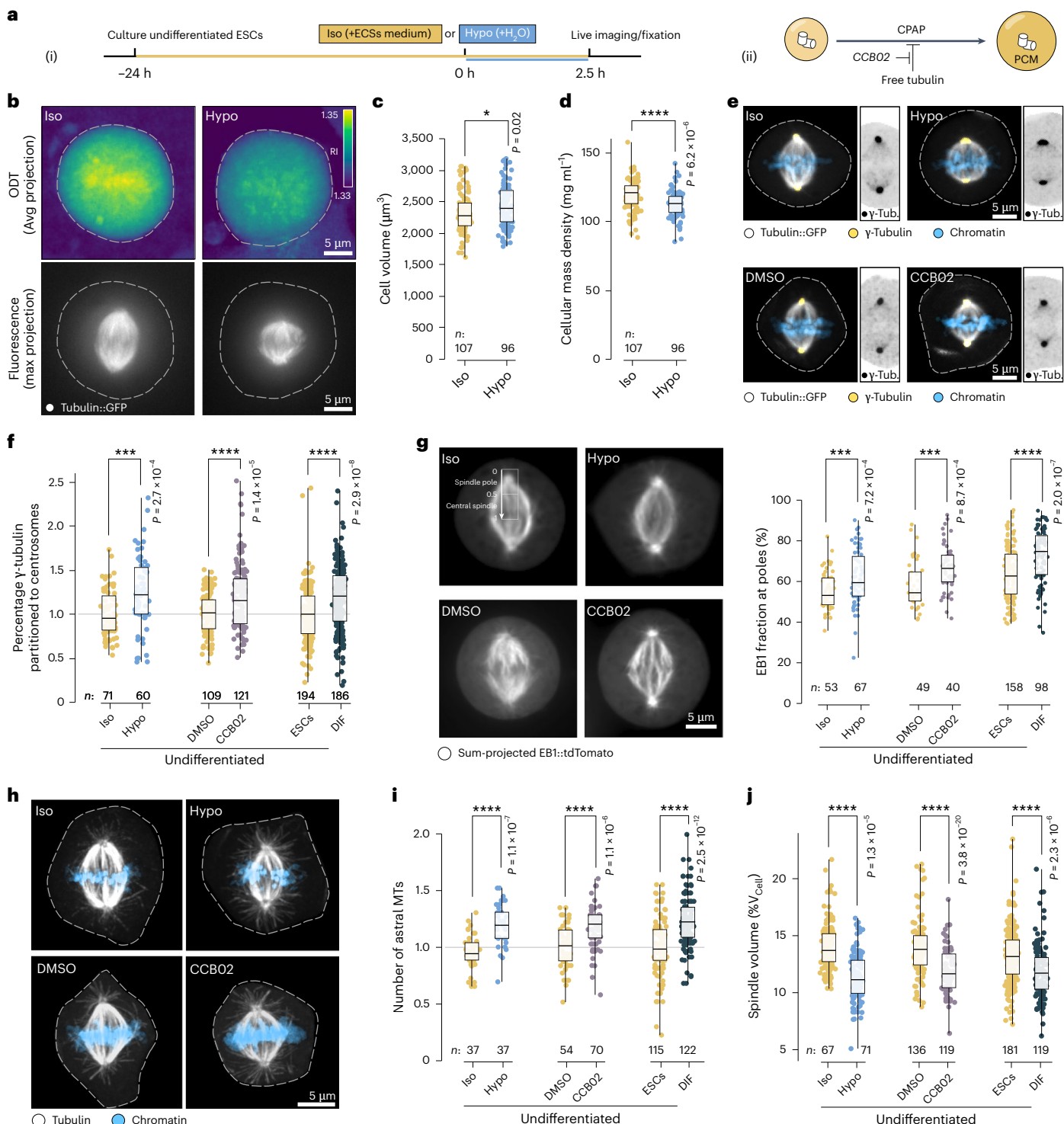

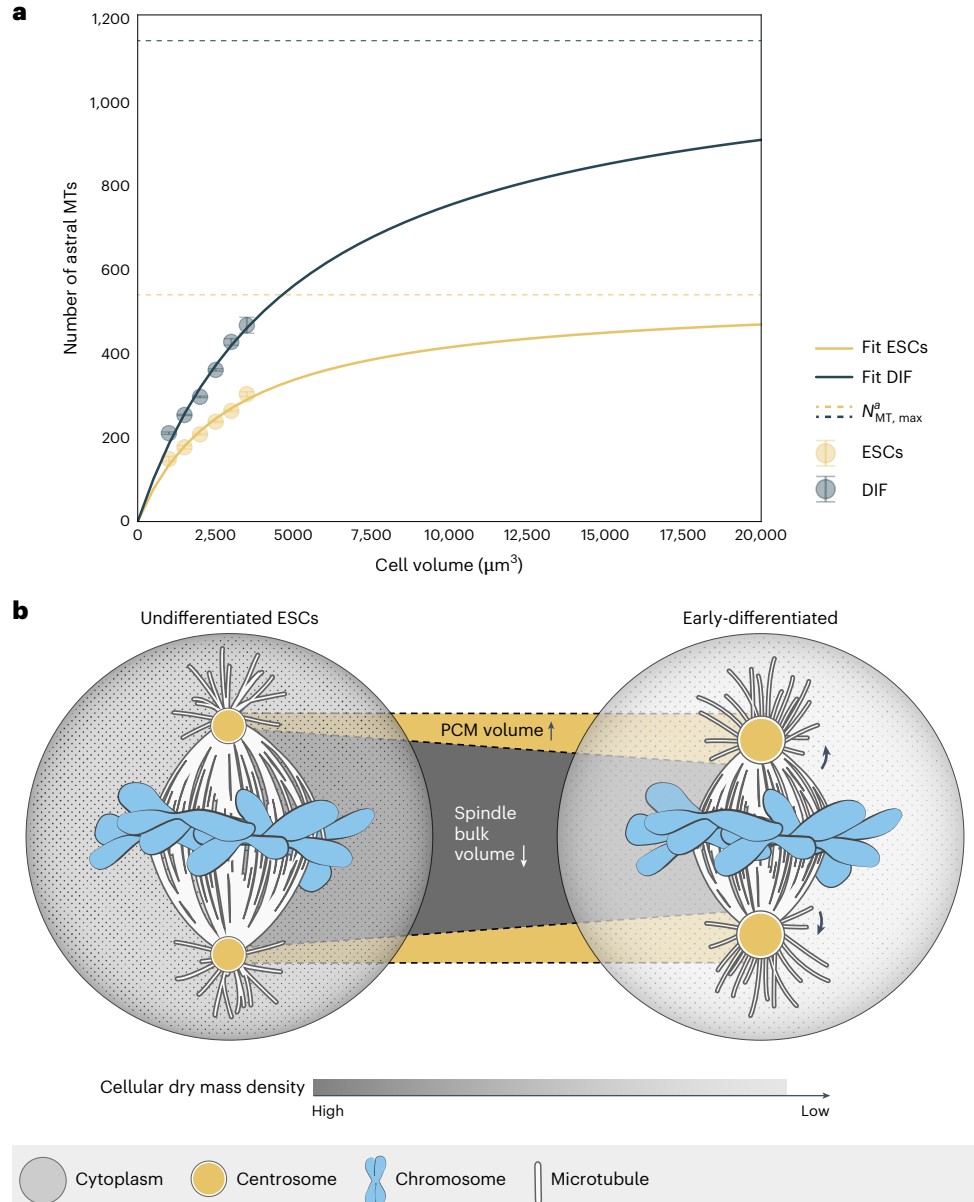

**Fig. 7 | Cytoplasmic dilution-driven changes in mitotic architecture in early-differentiated cells. a**, For both cell states (ESCs, yellow; DIF, blue), number of astral microtubules is estimated from spindle volume data (experimental data as Fig. 1h and Supplementary Note Fig. 3) and grouped by cell volume bins of 500 μm³. Large circles depict averages for each cell volume bin (modelled data) and are fitted to equation (6), error bars show s.d. Solid curves represent fit curves. Dashed lines depict predicted saturation values for astral microtubule number. **b**, Upon differentiating of ESCs, a reduction in cellular mass density leads to enlargement of the pericentriolar material (PCM), increasing centrosomal nucleation capacity and redistribution of microtubule mass away from the spindle bulk towards the asters.

cytoplasmic density to estimate the number of astral microtubules (also Supplementary Table 1). In our model, through the inhibition of a centrosomal regulator by free tubulin, the astral microtubule number has a Michaelis–Menten dependence on cell volume and is proportional to the concentration of free CPAP. This model is able to reproduce our experimental observations in stem cells and differentiating cells (Fig. 7a). Of note, this shows that spindle architecture can change considerably while total microtubule number does not. To ensure spindle size regulation during differentiation, this separation allows fine-tuning of spindle architecture without altering gross microtubule dynamics. Thus, by combining our experimental data and theory, we propose that the fundamental physical property of the cytoplasm (its mass density) provides a mechanism for regulating spindle architecture and size during differentiation.

## Discussion

The remarkable capacity of the spindle to scale with cell size has been observed across eukaryotes with spindle lengths ranging from 2 μm (yeast) to >50 μm (frog oocytes). Spindles showcase a large range in size not only between species but also within a single organism[74]. Yet, how spindles adapt to changes in cell size, shape and cytoplasmic properties as cells undergo differentiation was unclear. Here, we systematically quantify changes in spindle morphology and size during differentiation. Our data are consistent with a spindle scaling mechanism in which a reduction in cellular mass density leads to a relative increase in centrosome size in early-differentiated cells. As a consequence, microtubule nucleation shifts towards the spindle poles in these cells, promoting the generation of astral microtubules at the expense of the spindle bulk (Fig. 7b). This mechanism explains how a fundamental physical

property of the cell (mass density) can lead to changes in the spindle's nucleation profile and consequently spindle architecture.

## Spindle scaling mechanisms

Based on the current knowledge of spindle assembly, a number of biophysical mechanisms has been established that explain the regulation of spindle size and the coupling to cell size to achieve spindle scaling. In the limiting component (volume-sensing) model, the finite availability of critical components has been implicated in the regulation of organelle size. In the context of spindle assembly, limiting components can be structural[18,19] or regulatory[8,13]. A special case of component limitation is the cell surface-sensing mechanism, where the active cytoplasmic concentration of a spindle assembly factor is regulated through the sequestration of importin-α to the membrane[6,9,20]. Both the volume-sensing and surface-sensing scaling models describe physical mechanisms where the coupling of spindle size to cell size is inherently self-correcting. Here, we propose that another physical property of the cell is essential to spindle size regulation. We propose that changes in cellular mass density explain the cell state-specific scaling phenotypes in a differentiating system (Fig. 7). Indeed, a relationship between decreasing mass density and progressing differentiation has been proposed in many systems, including chondrocytes, keratinocytes, myeloid precursors and neurons[34–39]. Still, it is not immediately evident why and how neurally differentiating cells would change their mass density, in particular before terminal differentiation. One idea is that single-cell properties affect the overall mechanical properties of the developing tissue and this, in turn, influences differentiation[75–77]. This, however, was not the scope of this study and will therefore be part of future investigations.

In contrast to early stages of development, which typically are transcriptionally silent[78,79], differentiation is not. Why would a physical mechanism for spindle scaling be in place? Cell-state transitions are often asynchronous within a tissue[80–82]. Thus, programmed scaling mechanisms based only on developmental timing or cytoplasmic composition would not robustly couple spindle size to cell size, potentially leading to errors in spindle positioning and chromosome segregation. This might be particularly relevant in neural development, a process that is notoriously sensitive to mutations in spindle and centrosome genes[42,43].

We suggest CPAP as a molecular determinant of centrosomal nucleation capacity that is responsive to cytoplasmic dilution. Mutations in the CPAP gene link centrosomal defects to primary autosomal recessive microcephaly, a disorder characterized by severely reduced brain size and cognitive disability[83]. Consistently, mitotic spindle orientation defects have been described in cells depleted of CPAP[84,85]. While a direct link between microcephaly and spindle orientation defects remains to be definitely established, our data provide an explanation how interfering with CPAP function can have consequences for spindle architecture. Moreover, our findings are consistent with studies in neural stem cells of the embryonic developing mouse neocortex. Here, metaphase spindles displayed differences in their architecture from early to late neurogenic stages. At early neurogenic stages (comparable to the early-differentiated cells in our system), spindles displayed long astral microtubules and a reduced microtubule density near the chromosomes[22]. Together, these observations suggest that centrosome function via CPAP and resulting switches in spindle architecture could be a potential source for major developmental defects.

## Coordination of microtubule nucleation pathways

Mitotic spindle assembly mainly relies on two pathways that generate microtubules: centrosome- and chromatin-driven microtubule nucleation. Both pathways depend on the recruitment of γ-tubulin, either to the centrosome or to pre-assembled microtubules. The current understanding is that in very large cells, spindle scaling is mainly achieved via microtubule nucleation to create sufficient amounts of polymer mass, while in small cells regulating microtubule dynamics can be sufficient to scale spindle size[9]. Indeed, the dominance of nucleation pathways has been shown to shift in the course of mitosis[86] or even gradually over the course of development[6,10]. This cooperation of pathways is thought to provide robustness and mitotic fidelity[87], but as we show here, it can also result in the construction of spindles with differing architectures, providing the necessary plasticity to adapt spindles to evolving cellular environments.

## Online content

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

## Methods

### Quantification and statistical analysis

In each figure legend, details about the quantifications are provided, including the number of analysed cells and spindles measured (n). The effect size (Cohen's $d$) was calculated by: $d = (mean\_of\_ESCs − mean\_of\_DIFs)/s$ where s is the pooled s.d. of the parameter. No statistical methods were used to pre-determine sample sizes. No randomization was performed, as experiments were conducted with independent cell culture populations. Data collection and analysis were not performed blind to the conditions of the experiments. No data points were excluded from the analyses.

Data analysis was performed using the pandas library[91] and the NumPy library[92] in Jupyter Notebooks (using Python v.3.7.1). Data distribution was assumed to be normal, but this was not formally tested. Statistical tests were performed using the *SciPy* library (Welch's *t*-test for unequal variances, Wilcoxon signed-rank test)[93] or the Statsmodels package (ANOVA)[94]. Spearman's correlation coefficients were determined using the SciPy stats.spearman package. Minimum–maximum normalization (EB1 half-spindle profiles) was performed using the Scikit-learn library[95].

### Stem cell culture and differentiation

R1/E mouse ESCs stably transfected with bacterial artificial chromosomes harbouring the eGFP-fused coding region of human β5-tubulin and its regulatory sequences for native expression levels (a gift from the Hyman laboratory)[96] were cultured in FBS + LIF (16% FBS (Gibco), nonessential amino acids (Gibco), 50 μM β-mercaptoethanol, pen–strep (Invitrogen), $10^3$ U ml$^{-1}$ recombinant mouse LIF (Sigma-Aldrich) in high-glucose DMEM supplemented with L-Gln, pyruvate). After thawing from $N_2$ storage, ESCs were passaged every 48 h. Tissue culture dishes (100 × 25 mm) were coated with 0.1% gelatin. The cells were washed with pre-warmed 1× PBS (pH 7.4) and detached from the vessel (Accutase (Invitrogen) at room temperature (RT)). Detachment was stopped by addition of FBS + LIF medium. Cells were centrifuged for 2 min at 200*g*. Cells were replated at a density of 25,000 cells per cm$^2$ and incubated at 37 °C, 95% humidity and 5% $CO_2$.

Alternatively, ESCs were cultured using 2i + LIF medium (N2 in DMEM/F12 (Gibco) mixed 50:50 with B27 (200×, without vitamin A, Gibco) in neurobasal medium (Gibco), supplemented with L-Gln and 50 μM β-mercaptoethanol, pen–strep (Invitrogen), 3 μM CHIR99021 and 1 μM PD0325901) prepared from N2 supplement (100×) made in-house (0.6 mg ml$^{-1}$ progesterone (Sigma-Aldrich), 1.6 mg ml$^{-1}$ putrescine (Sigma-Aldrich), 10 mg ml$^{-1}$ apo-transferrin (Sigma-Aldrich), 3 μM sodium selenite (Sigma-Aldrich), 2.5 mg ml$^{-1}$ insulin (Sigma-Aldrich) and 0.5% bovine albumin fraction V (Gibco) in DMEM/F12 (Gibco)). The cells were passaged as described above but using 2i + LIF instead of FBS + LIF and at a reduced cell seeding density of 15,000 cells per cm$^2$.

For neural differentiation, ESCs were taken up in N2B27 medium (N2 (see above) in DMEM/F12 (Gibco) mixed 50:50 with B27 (200×, plus vitamin A, Gibco) in neurobasal medium (Gibco), supplemented with L-Gln and 50 μM β-mercaptoethanol, pen–strep (Invitrogen), 500 nM retinoic acid (Roth)) and seeded onto laminin-511 (L511, 2.5 μg ml$^{-1}$ in 1× PBS (containing Ca$^{2+}$ and Mg$^{2+}$)) coated dishes or imaging wells at a density of 19,000 cells per cm$^2$ and incubated at 37 °C, 95% humidity, 5% $CO_2$. For differentiation times longer than 48 h, cells were passaged after 48 h and replated at a density of 200,000 cells per cm$^2$.

Cells were routinely tested for *Mycoplasma* contamination using a commercial detection kit.

### RNAi

For siRNA transfection, ESCs were seeded at a seeding number of 3,500 cells per cm$^2$ on L511-coated dishes. After 24 h, the cells were transfected using the Lipofectamine 3000 kit (Thermo) and 20 nM of siRNAs (katnb1 5′-GGACUACAGGAGAUAUCAAtt-3′ or scrambled 5′-UUCUCCGAACGUGUCACGUtt-3′) according to the manufacturer's instructions. After 48 h of knockdown, the cells were either fixed, immunostained and imaged or lysed and prepared for western blotting.

### Live-cell confocal microscopy

For single time point imaging of mitotic cells, cells were seeded and differentiated as described above in L511-coated imaging plates (polymer bottom, Ibidi). Cells were imaged using a Nikon spinning-disk confocal equipped with an incubation chamber (37 °C, 95% humidity, 5% $CO_2$), an Andor Revolution SD System (CSU-X), an iXon3 DU-888 Ultra EMCCD camera and a ×60 Plan Apo oil objective (numerical aperture (NA) 1.40), or on a Nikon Crest X-Light V3 spinning-disk confocal equipped with an incubation chamber (37 °C, 95% humidity, 5% $CO_2$) and CFI Plan Apo VC ×60 water objective (NA 1.2).

Confocal series (z-range of 23 μm at a step size of 0.3 μm) were recorded using a multi-wavelength filter and the 488 nm and 640 nm laser lines at 5% laser power and 200-ms exposure time.

### Adaptive feedback microscopy

Datasets were recorded on two laser-scanning confocal microscopes, the Zeiss LSM 800 and LSM 900, equipped with custom-built, temperature, $CO_2$ and humidity-controlled incubation chambers (electronic and mechanical workshops, European Molecular Biology Laboratory (EMBL) Heidelberg). A C-Apochromat ×40 water objective (NA 1.2) was used. Signals were detected using GaAsP photomultiplier tubes. The objective was equipped with a custom-built automated water immersion system (electronic and mechanical workshops, EMBL Heidelberg).

The pipeline automatically locates mitotic cells and images them in 3D with high resolution. The pipeline consists of four image acquisition tasks and three image analysis tasks as described below. For settings, see Supplementary Table 2. The pipeline was driven by Zeiss ZEN Blue macros and an analysis protocol in Fiji[97] using the AutoMicTools package.

(1) Low-zoom autofocus. For each position in each imaging cycle, the focal plane was re-determined by using a reflection signal from coverslip surface. A fast XZ scan was performed using the piezo drive and the reflection signal was calculated by online image analysis and fed back to the microscope to trigger the acquisition of the low-zoom population image in focus.

(2) Low-zoom population image. Two-channel (DNA, GFP) overview images of cells at predefined positions were acquired by recording 2 × 2-tile scans with a 10% overlap stitched in the ZEN Blue software. The stitched image was processed to identify positions of mitotic cells.

(3) High-zoom autofocus. After receiving the *xy* coordinates of the positive hits, the pipeline determined the central focal plane for each mitotic cell for (4). For that purpose, a stack at low lateral resolution, high zoom factor and extended z-range was automatically acquired for each cell in question. The online image analysis in AutoMicTools identified the mask of the tubulin::GFP-labelled spindle and found its brightest z-section to correct lateral and axial position of a high-resolution acquisition.

(4) High-zoom single-cell image. The positive hits were recorded in two channels with high spatial resolution.

For one imaging cycle (interval of 10 min), the microscope moved to the first predefined *xy* position and recorded the low-zoom autofocus image (1) that was analysed using the AutoMicTools plugin in Fiji. The correct focal *z* plane was sent back to the microscope that subsequently performed task (2). In the dedicated AutoMicTools online image analysis task the GFP channel in the overview raw image was maximum projected. Positions of mitotic cells were identified using the Multi-Template Matching plugin[98] where we provided a template for object detection of tubulin::GFP-labelled spindles. The list with the *xy* coordinates was fed back to the microscope to proceed with

tasks (3) and (4). Once completed, the cycle started anew at the next predefined position.

## Quantification of cell cycle duration

Cell families were traced using the manual mode in TrackMate[99] in the max-projected and median (sigma = 2)-filtered time-lapse videos generated via adaptive feedback microscopy. The time interval between two track division events was read out as the cell cycle duration.

## Quantification of cell and spindle morphology

Cell volumes and cell surface areas were calculated via MorphoLibJ[100] in Fiji after pixel-classifying and segmenting cells in 3D based on tubulin::GFP signal using Ilastik[52] or using the Volume Manager Fiji plugin (Dresden, https://sites.imagej.net/SCF-MPI-CBG/). Spindle volume, length and width were determined using Spindle3D (v.0.80)[51] after excluding all voxels outside the cell binary masks. In brief, Spindle3D finds a tubulin threshold by subdividing the tubulin voxels that are in contact with the chromatin mask into two populations, the cytoplasmic pool and the spindle pool. Thus, spindle volume is a proxy for spindle tubulin mass. Other geometrical parameters such as spindle length and width are based on the spindle volume mask. For clarity, microscopic images shown in the figures only show voxels within the cell binary masks.

## Quantification of spindle tubulin turnover via fluorescence recovery after photobleaching

Tubulin::GFP signals in live spindles were bleached in Leica LAS software using a Leica TCS SP8 (Leica Microsystems) laser-scanning confocal system equipped with an incubation chamber (37 °C, 95% humidity, 5% $CO_2$) and a ×63 Plan Apo glycerol objective (NA 1.30) and an Argon laser (set to 10%). A bleaching region of interest (ROI) of 4.4 × 1 µm spanning the spindle width at the half-distance between equator and pole was defined. Before and after bleaching, cells were imaged using the 488 nm laser (6.84 %), a photomultiplier tube with a gain of 760 V, in a field of view of 512 × 512, a zoom factor of 5.00 and a scanning speed of 400. One acquisition encompassed three frames pre-bleach at 1 s per frame, 1 s of bleach (488 nm at 100%) and 35 frames post-bleach at 1 s per frame. Signals within the bleached ROI were corrected for regular photobleaching during capture using a control ROI within the other half of the spindle, and min–max normalized. To determine recovery half-times, the corrected data were fit to an exponential recovery function $A \times (1 - e^{(-t/\text{tau})}) + C$ using the scipy.optimize library. Half-times were calculated by $t_{1/2} = \text{tau} \times \ln 2$.

## Quantification of microtubule growth velocity and EB1 comet numbers

ESCs were transiently transfected following manufacturer's instructions (Lipofectamine 3000, Thermo) with plasmids carrying the EB1::tdTomato coding sequence under a CMV promoter (Addgene 50825). After overnight transfection, cells were plated onto L511-coated wells of a 24-well polymer bottom imaging plate (Ibidi) at a seeding density of 19,000 cells per $cm^2$, and either maintained in FBS + LIF medium, or N2B27 medium for neural differentiation.

After 48 h, cells were imaged on a Nikon spinning-disk confocal equipped with an incubation chamber (37 °C, 95% humidity, 5% $CO_2$), a Andor Revolution SD System (CSU-X), an iXon3 DU-888 Ultra EMCCD camera and a ×100 Plan Apo oil objective (NA 1.45). A confocal series of the tubulin::GFP signal was recorded (step size of 0.75 µm), using a multi-band filter and the 488 nm laser at 5% power and 200 ms of excitation duration.

Fluorescent EB1 comets were imaged in the most central position within the spindle (400 ms frames for 1 min) using a single-band filter in combination with the 561 nm laser at 5% and 300 ms of excitation duration.

For automated single-particle tracking using the Fiji plugin TrackMate[99], the recordings were processed using a median filter

(sigma = 1 µm) and a rolling-ball background subtraction (radius of 1 µm). In TrackMate, particles were detected using the Laplacian of Gaussian algorithm (subpixel localization, estimated blob diameter 0.5 µm, no median filter and threshold of 100–130). To generate tracks, the LAP tracker algorithm was used with the following settings: frame linking max distance 0.3 µm, gap closing not allowed, track splitting not allowed, track merging not allowed. Tracks were filtered by duration (2–4 s), median track quality (200–600) and track start and end within the video (frames 0–30).

Comet numbers were determined manually. For each recording, the numbers of spindle bulk EB1 comets and astral EB1 comets were determined in frame number 1, 20 and 40 and averaged. Last, the EB1::tdTomato signals were sum-projected in Fiji. Using the Fiji line tool at an averaging of 15 pixels, half-spindle profiles were drawn starting from one pole to the spindle equator. Projections were only included if the centre of the spindle and at least one pole was clearly in focus. For each profile, the $x$ axis and the $y$ axis were normalized using the min_max_scaler.fit_transform function in the Scikit-learn library[95].

## Quantification of pericentriolar material and spindle-localized proteins

Cells were chemically fixed using 3.2% PFA (in 1× PBS) for 10 min or fixed with ice-cold methanol for 4 min at −20 °C. After three washes with 1× PBS and 10 min quenching with 0.1 M glycine at RT (PFA-fixation only), the cells were treated with BSA blocking buffer (3% BSA/0.5% TX-100 in 1× PBS) for 1 h at RT. Primary antibody incubation was performed in BSA blocking buffer for 1 h at RT (CDK5RAP5 (rabbit, Sigma 06-1398), 1:500; CEP192 (rabbit, Proteintech 18832-1-AP), 1:500; pericentrin (rabbit, Abcam 4448), 1:500; γ-tubulin (mouse, Sigma T6557), 1:100; katanin p60 (rabbit, Proteintech 17560-1-AP), 1:125; katanin p80 (rabbit, Proteintech 14969-1-AP), 1:250, HAUS6 (rabbit, a gift from L. Pelletier), 1:200, TPX2 (mouse, a gift from A. Bird), 1:500, CKAP2 (mouse, Proteintech 25486-1-AP) and 1:200, KIF2C/MCAK (rabbit, Abcam ab71706), 1:500). Secondary antibody incubation was performed for 1 h at RT using goat anti-mouse (Thermo A-21235) or anti-rabbit (Thermo A-21244) Alexa Fluor 647 secondary antibodies diluted 1:1,000 in BSA blocking buffer.

Cells were imaged on a Nikon spinning-disk confocal microscope at RT, equipped with an Andor Revolution SD System (CSU-X), an iXon3 DU-888 Ultra EMCCD camera and a ×100 Plan Apo oil objective (NA 1.45) or a ×60 Plan Apo oil objective (NA 1.40). Confocal series were recorded using a z-step size of 0.3 µm. Using single-band filters, stacks of the individual channels were recorded serially (405 nm, 5% laser power; 488 nm, 18% laser power; and 640 nm, 5% laser power). Excitation duration per slice was 200 ms. Cells and spindles were segmented as described above.

The spindle binary mask was used to collect fluorescent signals of spindle-localized proteins (HAUS6, TPX2, CKAP2, KIF2C/MCAK, γ-tubulin). For 3D segmentation of the PCM, the PCM signal was thresholded using the maximum entropy algorithm in Fiji, considering the histogram from all optical sections. This method was robust because all noncell voxels were already eliminated from the confocal stack. Volumes of the centrosomes were calculated from the centrosome binary mask using the Analyze Regions 3D function in MorphoLibJ[100]. The average cellular fluorescence was determined using the Fiji 3D Suite[101] and the cell binary mask. Analogously, this was repeated for the average signal in the centrosome binary mask/the spindle binary mask. The average signals from the centrosome mask and the spindle mask, respectively, were normalized using the cell average signal.

## Quantification of astral microtubules

For this work, we define astral microtubules as the subpopulation of mitotic microtubules that originate at the centrosomes and are not incorporated into the spindle bulk. To visualize astral microtubules, cells were fixed and stained with anti-tubulin antibodies. Cells were

chemically fixed for 10 min (3.2% paraformaldehyde/0.1% glutaraldehyde in 1× PBS), washed three times with 1× PBS, quenched with 0.1% $NaBH_4$ in 1× PBS for 7 min at RT and for 10 min with 100 mM glycine in 1× PBS and blocked (3% BSA / 0.5% TX-100 in 1× PBS) for 1 h at RT. The cells were incubated for 1 h at RT with BSA blocking buffer supplemented with anti-α-tubulin antibodies (DM1a (Sigma T-6199, mouse origin), diluted 1:1,000 and Yol1/34 (Bio-Rad MCA78G, rat origin), diluted 1:1,000). After three washes with 1× PBS, cells were incubated with secondary antibodies (goat anti-mouse Alexa Fluor 647 (Thermo A-21235) 1:1,000 in blocking buffer and goat anti-rat Alexa Fluor 647 (Thermo A-21247) 1:1,000 in blocking buffer). After DNA staining with Hoechst 33342 (1:10,000 in 1× PBS) for 5 min at RT, cells were washed three times with 1× PBS.

Cells were imaged on a Nikon spinning-disk confocal microscope at RT. Confocal series were recorded using a z-step size of 0.3 μm. Using single-band filters, stacks of the individual channels were recorded serially (405 nm, 5% power; 488 nm, 18% power; and 640 nm, 5% power). Excitation duration per slice was 200 ms.

The stained microtubule signals were maximum projected in Fiji. The individual astral microtubules were traced using the free-hand selection tool.

### Western blotting and tubulin quantification

Whole-cell lysates were prepared from adherent ESCs throughout differentiation. The lysis of cells from different stages (24–120 h) of differentiation was synchronized by staggering the differentiation onset. Using the differentiation procedures as described above, cells were seeded onto L511-coated six-well plates, three replica wells per differentiation day. For the '0 h' time point, the ESCs were seeded at a density of 19,000 cells per $cm^2$ and incubated in FBS + LIF for 48 h.

Plates were placed on ice and cells were incubated with cold, protease inhibitor-supplemented RIPA lysis buffer (150 mM NaCl/50 mM Tris-HCl (pH 8.0)/1% Nonidet P-40/0.5% sodium deoxycholate/0.1% SDS in ddH₂O) and scraped off. Lysates were collected and incubated for 15 min on ice. The lysates were sonicated in a water bath sonicator three times for 2 s each, with 1-min recovery periods on ice between the pulses. The lysates were incubated another 15 min on ice before centrifuging at 13,000g for 5 min at 4 °C. Protein concentrations of supernatants were determined using a bicinchoninic acid (BCA) assay (Thermo).

Lysates were diluted 1:4 with SDS sample buffer (4×) (200 mM Tris-Cl (pH 6.8), 400 mM dithiothreitol, 8% SDS, 0.4% bromophenol blue and 40% glycerol) and heated for 5 min at 95 °C. Sample volumes corresponding to 4 μg, 2 μg and 1 μg of total protein, alongside defined masses (25 ng, 50 ng and 100 ng) of tubulin purified from ESCs, were loaded onto NuPAGE 4–12% Bis-Tris protein gels (Thermo) and mounted in gel running chambers filled with 1× NuPAGE SDS running buffer (Thermo). A replicate gel (with adjusted masses of tubulin controls of 100 ng, 200 ng and 400 ng) was loaded in parallel for Coomassie staining of total protein for normalization[102].

The original gel was blotted onto a nitrocellulose membrane using 1× NuPAGE transfer buffer and a wet transfer protocol in an ice-cold transfer chamber for 1 h at 40 V. The membrane was blocked in TBS blocking buffer (LI-COR) for 1 h at RT. Staining with anti-α-tubulin antibodies (DM1a (Sigma T-6199), 1:5,000) was carried out in TBS blocking buffer supplemented with 0.2% Tween-20 overnight at 4 °C. Next, the membranes were washed three times using TBS-T and then incubated with secondary antibodies (goat anti-mouse Alexa Fluor 800 (Thermo A-32730), 1:5,000) for 1 h at RT. Finally, the membranes were washed again three times in TBS-T. Bands were fluorescently detected on an Odyssey XF system (LI-COR) with a 2-min integration time. The signals were quantified using Fiji's rectangular selection tool. First, the sum signals of the tubulin standards were measured to generate a calibration curve. The sum signals of the lysate lanes were determined (using the rectangular selection tool) and calibrated using the calibration curve.

For normalization, the replicate gel was stained using an Instant-Blue Coomassie protein stain (Abcam) according to the manufacturer's instructions. Coomassie can be used as a sensitive fluorescent in-gel protein stain[103]. Protein was fluorescently detected in a ChemiDoc system (Bio-Rad) using 700 nm excitation light. The raw total protein reference image was de-noised in Fiji using a median filter (sigma = 3 pixels). The background was subtracted using the Fiji rolling-ball algorithm at a radius of 50 pixels. For each of the lanes including the tubulin calibration lanes, the raw pixel sum was measured using the rectangular selection tool.

### Tubulin purification

Tubulin was purified as previously described[70,104]. In brief, cultures of undifferentiated ESCs and early-differentiated ESCs were collected and centrifuged for 5 min at 300g at 4 °C. Lysis was performed on ice using a Dounce homogenizer and supernatants were loaded onto a TOG column and tubulin was eluted using high salt.

### Intact protein analysis by liquid chromatography mass spectrometry

Intact protein masses were determined by liquid chromatography mass spectrometry (LC–MS) as described previously[105]. In brief, samples were analysed using the Ultimate 3000 liquid chromatography system connected to a Q Exactive HF mass spectrometer via the ion max source with HESI-II probe (Thermo). The proteins were desalted and concentrated by injection on a reversed-phase cartridge (MSPac DS-10, 2.1 × 10 mm, Thermo). Full MS spectra were acquired using the following parameters: intact protein mode on, mass range $m/z$ 600–2500, resolution 15,000, AGC target $3 × 10^6$, microscans 5 and maximum injection time 200 ms. The software tool UniDec was used for data processing and deconvolution of protein masses[106]. First, an averaged spectrum was generated from each measurement followed by spectral deconvolution using the default settings except for the following adjustments: charge range 38–65, mass range 45–55 kDa, sample mass every 1 Da and peak FWHM 0.1.

### Osmotic challenge

For hypo-osmotic challenges, the growth medium was diluted down to 80%, 75% or 50% of its original concentration with ultrapure DNase/RNase-free distilled water (Invitrogen). As isosmotic control, full growth medium was added. To allow spindle recovery, live-cell imaging or chemical fixation was performed after 2.5 h post-treatment. Media osmolalities were measured using freezing point osmometry via the Osmomat 3000 basic (Gonotec).

For hyperosmotic challenge, 50 mM or 100 mM sorbitol to the differentiation medium. For isosmotic control, the corresponding volume of regular differentiation medium was added. Before imaging, the cells were left for 1 h to recover.

### Quantification of cellular mass density

Cellular mass density was measured using optical diffraction tomography as described by Biswas et al.[29]. In brief, a coherent laser beam was split into two using a beam splitter. While one beam was used as a reference beam, the sample was illuminated with the other using a ×60 water-dipping objective lens. Using a dual-axis galvanometer mirror, the sample was illuminated from 150 incident angles. After passing through the sample, the scattered light was gathered by a ×60 objective. At the image plane, the sample beam interfered with the reference beam to generate spatially modulated holograms. These were recorded by a charge-coupled device camera. To generate 3D tomograms, the holograms were reconstructed using a custom-written MATLAB script (github.com/OpticalDiffractionTomography)[107]. In Fiji, the cells were 3D segmented as described above, using the refractive index (RI) intensities in the tomograms. The cell average RI value (corresponding to the average RI value of all voxels within the 3D cell mask) was determined

using the Fiji 3D Suite[101]. For translating the raw RI value into mass density (in mg ml$^{-1}$), we applied:

$$(((RI_{raw\ voxel\ average}/10,000) - RI_{medium}) \times 1,000)/0.19$$

The RI of the medium ($RI_{medium}$) was determined for each experiment using an Abbe refractometer (Arcada ABBE-2WAJ).

## Reporting summary

Further information on research design is available in the Nature Portfolio Reporting Summary linked to this article.

## Data availability

Source data are provided alongside the paper. Supporting data are available in the BioStudies database[108] (http://www.ebi.ac.uk/biostudies) under accession code S-BIAD1680. Mass spectrometry data that support the findings of this study have been deposited to the ProteomeXchange Consortium under accession code PXD061228. All other data supporting the findings of this study are available from the corresponding author on reasonable request. Cell lines are available from S.R. under a material transfer agreement with the Max Planck Society. Source data are provided with this paper.

## Code availability

Source code, usage and installation guidelines for the implemented adaptive feedback microscopy pipeline is available at https://git.embl.de/grp-almf/automictools-zenblue-spindle-kletter. Analytical codes used for this study are deposited under https://github.com/TobiasKletter/Scaling.

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

## Acknowledgements

The authors thank all past and current members of the Zaburdaev and Reber laboratories. We thank staff at the Advanced Medical Bioimaging Core Facility (Charité, Berlin), the Advanced Light Microscopy Facility at the EMBL and Zeiss for imaging support. The authors acknowledge the support of i3S Scientific Platform Advanced Light Microscopy, a site of PPBI Euro-Bioimaging Node. For mass spectrometry, we acknowledge the assistance of staff at the Core Facility BioSupraMol supported by the Deutsche Forschungsgemeinschaft. We thank H. Maiato and laboratory for their support. We thank J. Gopalakrishnan for sharing reagents. We thank R. Basto, E. Taverna and F. Mora Bermudez for helpful discussions and advice on neural differentiation and M. Taylor, V. Štimac and M. Mendoza for critical comments on the manuscript. For funding, the authors acknowledge the Max Planck Society (to O.M. and S.R.), the Horizon 2020 Framework Programme of the European Union (iNEXT grant 653706, project PID 3503 to S.R.) and the Deutsche Forschungsgemeinschaft Project-ID 528483508 – FIP 12 (to S.R.).

## Author contributions

T.K. and S.R. conceived and designed the study. T.K. performed all experiments including data and image analysis. S.R. contributed to method development. A.B. supported ODT setup development and quantitative phase imaging. B.N. supported long-term live-cell imaging and RNAi experiments. A.H. designed the adaptive microscopy feedback pipeline. B.K. performed and interpreted the mass-spectrometric analyses. O.M. and V.Z. conceptualized and wrote the Supplementary Note. T.K. and S.R. wrote the manuscript with input from all authors.

## Funding

## Competing interests

The authors declare no competing interests.

## Additional information

**Extended data** is available for this paper at

**Supplementary information** The online version contains supplementary
material available at https://doi.org/10.1038/s41556-025-01678-x.

**Correspondence and requests for materials** should be addressed to
Simone Reber.

**Peer review information** *Nature Cell Biology* thanks Nicolas Minc and
the other, anonymous, reviewer(s) for their contribution to the peer
review of this work. Peer reviewer reports are available.

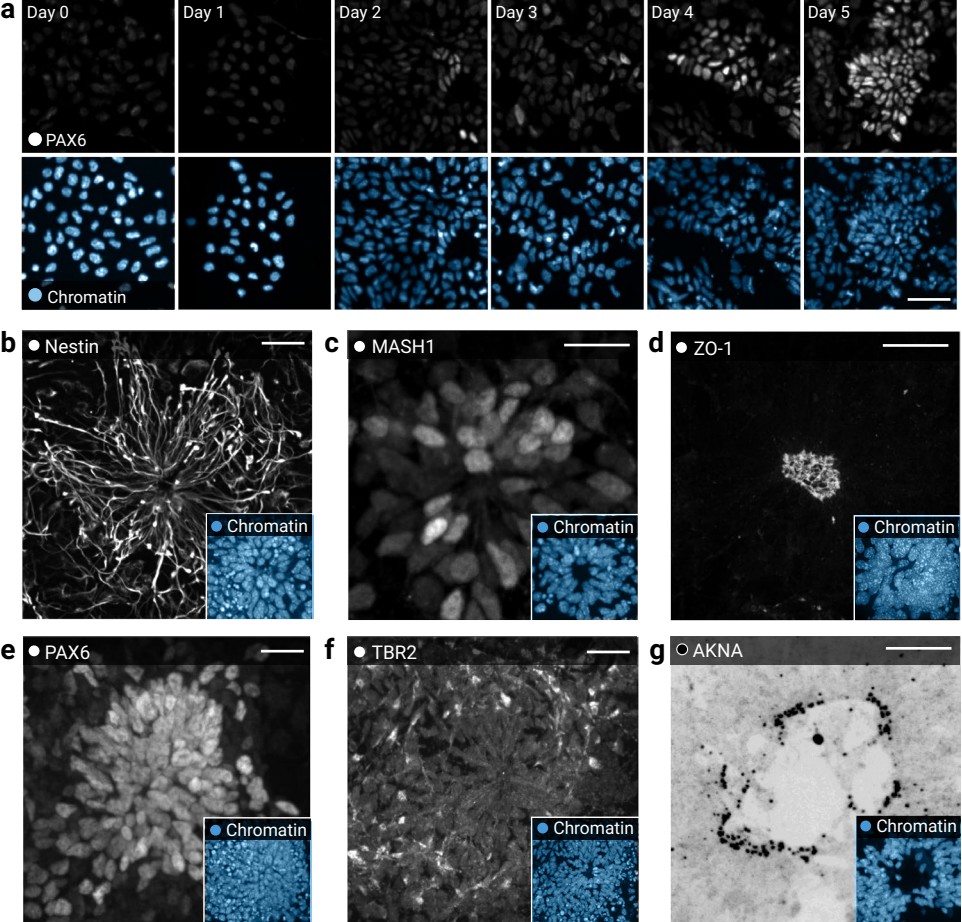

**Extended Data Fig. 1 | Marker expression and tissue architecture confirm successful differentiation of ESCs into neural rosettes. a**. Top: Confocal data showing immunostaining of PAX6[109] in fixed ESCs ("Day 0") and during 5 days of neural differentiation. Bottom: Nuclei stained by Hoechst. Scale bar: 50 μm. **b**. Immunofluorescent detection of nestin[110] in ESCs-derived rosette cells after 5 days of differentiation. Inlets show nuclei stained by Hoechst. Scale bars: 20 μm. **c**. As in (**b**) but detecting MASH1 (also known as ASCL1), a proneural transcription factor[111]. **d**. As in (**b**) but detecting ZO-1, marking tight junctions in the rosette centre[112]. **e**. As in (**b**) but detecting PAX6, uniformly expressed in the rosette bulk. **f**. As in (**b**) but detecting TBR2[113], sporadically expressed by cells in the rosette periphery. **g**. As in (**b**) but detecting AKNA, a centrosomal protein in neural stem and progenitor cells. AKNA[+] centrosomes strictly localize to the "apical" pole during interphase[114]. Lookup table of this panel is inverted in comparison with other panels for better visibility. All stainings (**a-g**) were performed for N = 2 independent experiments.

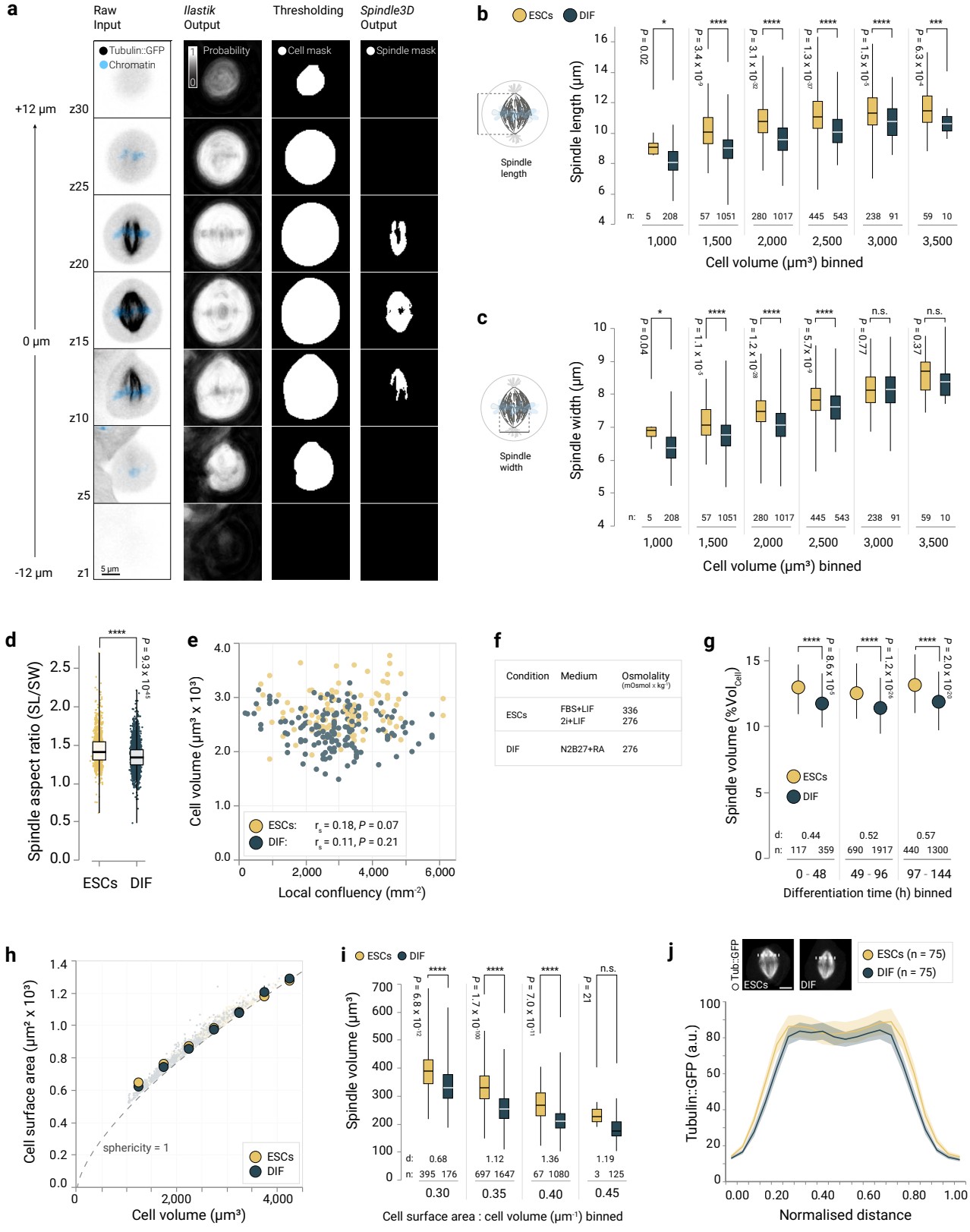

**Extended Data Fig. 2 | See next page for caption.**

**Extended Data Fig. 2 | Spindle size subscaling in differentiating embryonic stem cells (ESCs) is independent of confluency and cell geometry.**
**a.** Volumetric cell analysis, based on training pixel-classifier models in *Ilastik*[52] to distinguish mitotic cytoplasm via tubulin::GFP. Spindle volume masks were generated via *Spindle3D*[51], based on adaptive thresholding of tubulin::GFP. **b.** Spindle length after binning data into cell volume bins (bin size 500 μm³). Numbers inside the bins show respective n of cells (data sets as in Fig. 1h, n = 5, 208; 57, 1051; 280, 1017; 445, 543; 238, 91; 59, 10 cells for ESCs, DIF conditions binned from 1000 μm³ through 3500 μm³ in 500 μm³ bins, pooled from 9 independent replicates). Boxes denote interquartile ranges, horizontal lines inside boxes show medians, whiskers indicate minima and maxima. Welch's t-test (two-sided). **c.** Same as c) but showing spindle width. **d.** Spindle aspect ratio (spindle length / spindle width). Boxes denote interquartile ranges, horizontal lines inside boxes show medians, whiskers indicate minima and maxima. Data points show individual cells (ESCs n = 1,084 cells, DIF n = 2,920 cells pooled from 9 independent experiments, see Fig. 1h). Welch's t-test (two-sided), *P* = 9.3 x 10⁻⁴⁵. **e.** Mitotic cell volume is independent of the local cell confluency. Single data points represent individual cells (ESCs n = 110 cells (yellow) and DIF n = 151 cells (blue) from 3 experiments). $r_s$: Spearman's correlation coefficient ($P_{ESCs}$ = 0.07, $P_{DIF}$ = 0.21). **f.** Osmolality of the ESCs culturing medium and of the differentiation medium. **g.** Cell volume occupied by the spindle in ESCs (yellow) and DIF (blue) per differentiation time bin. The circles show the means, error bars show the standard deviation. Welch's t-test (two-sided). n: data points in each bin. **h.** Mitotic cells in both differentiation states are spherical. Each data point represents an individual cell (ESCs n = 1,084 (yellow) and DIF n = 2,920 (blue), data pooled from 9 independent experiments). Big circles represent the median of each cell volume bin, the error bars show the interquartile range. The dotted line represents the behaviour of a perfect sphere. **i.** In cells with comparable cell surface area : cell volume ratios, spindle volume subscales in DIF (blue) when compared with ESCs (yellow). Boxes show interquartile ranges, white lines inside boxes denote medians, whiskers show minima and maxima. n = 395, 176; 697, 1647; 67, 1080; 445, 543; 238, 91; 59, 10 cells for ESCs, DIF conditions binned from 0.3 μm⁻¹ through 0.45 μm⁻¹ in 0.05 μm⁻¹ bins, pooled from 9 independent replicates. d: Cohen's d. Welch's t-test (two-sided) for data in each bin. **j.** Top: Maximum-projected confocal images of mitotic ESCs or DIF (tubulin::GFP in grey) acquired during our automated live-cell imaging protocol (Fig. 1e). Bottom: tubulin::GFP-based intensity profiles of a cross-section between spindle equator and pole in maximum projected confocal images (top, dotted white lines). For consistency, cells were randomly drawn from a pool of cells within the 2500-3000 μm³ cell volume bin (ESCs n = 75 cells (yellow) and DIF n = 75 cells (blue) pooled from 6 independent experiments). Lines denote the means, shaded areas denote the 95% confidence intervals. *: *P* < 0.05, ***: *P* < 0.001, ****: *P* < 0.0001, n.s.: not significant, *P* > 0.05. d: Cohen's d.

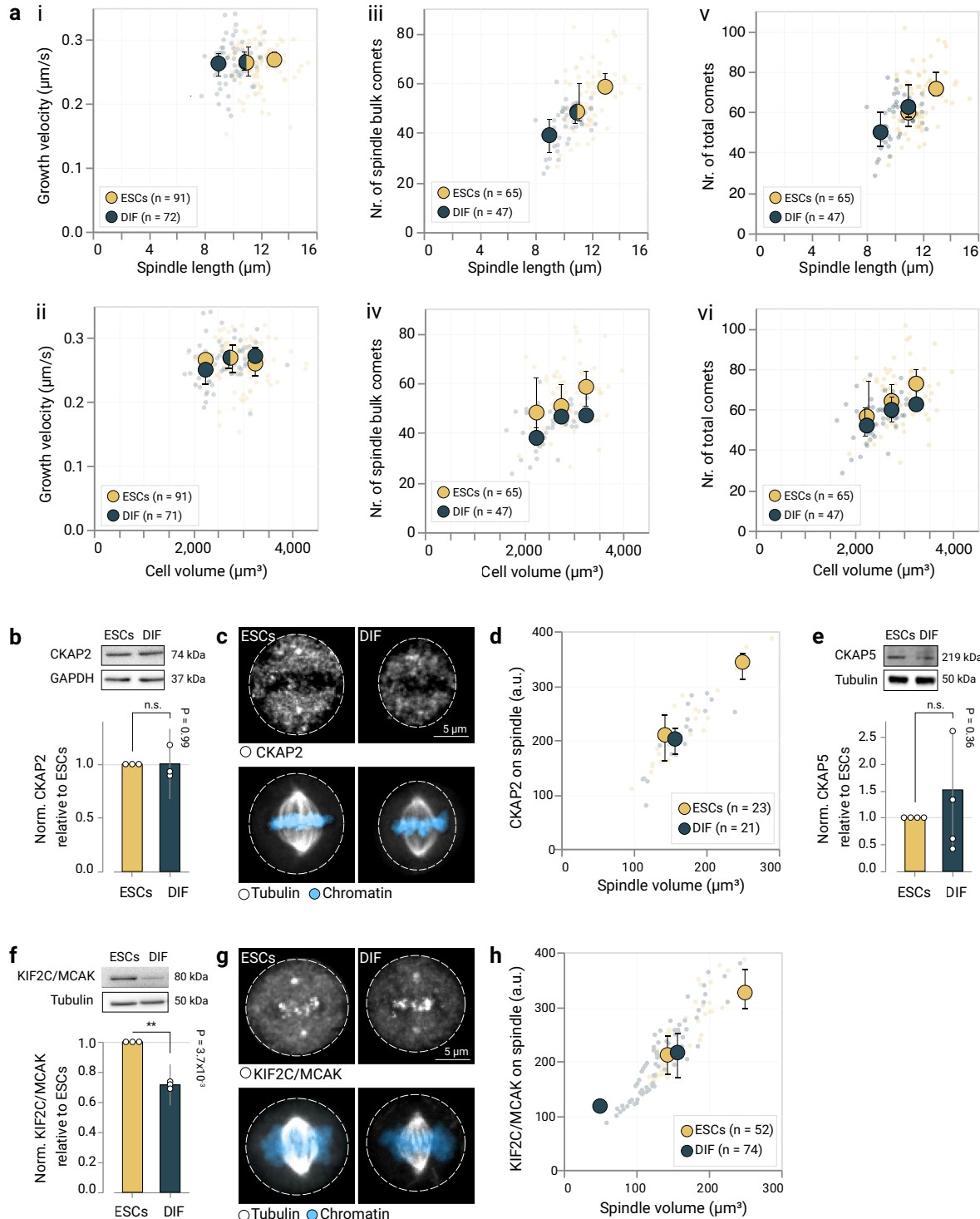

**Extended Data Fig. 3 | See next page for caption.**

**Extended Data Fig. 3 | Microtubule growth speed is independent of spindle length and cell volume. a**. Average single-cell microtubule growth velocities (EB1::tdTomato comets) as a (i) function of spindle length and (ii) cell volume in ESCs (n = 91 cells) or DIF (n = 72 cells). Average single-cell number of EB1::tdTomato spindle bulk comets as a function of (iii) spindle length and (iv) cell volume in ESCs (n = 65 cells) or DIF (n = 47 cells). Average single-cell number of total EB1::tdTomato comets as a function (v) of spindle length and (vi) cell volume in ESCs (n = 65 cells) or DIF cells (n = 47 cells). Data from pooled from 6 independent experiments. Small circles show averages of individual cells, large circles represent medians of binned data, error bars represent interquartile ranges. **b**. Cellular levels of CKAP2 after 48 h of differentiation relative to ESCs, probed by western blotting and normalized to GAPDH (N = 3, each time loading 2 independent batches of protein extracts). Bars show mean, errors show standard deviation. Welch's t-test (two-sided), $P = 0.99$. **c**. Confocal micrographs (maximum projected, from N = 2 experiments) of methanol-fixed ESCs or DIF at metaphase. Top: Immunostained CKAP2 signal, bottom: tubulin::GFP (grey) and chromatin counterstained by Hoechst (blue). Dotted lines indicate cell boundaries. Scale bar: 5 μm. **d**. Fluorescent CKAP2 signal on spindle (normalized by total cell CKAP2) summed, as a function of spindle volume. Data points represent single cells (ESCs n = 23 cells and DIF n = 21 cells, data from 2 independent experiments.), large circles denote medians in each bin, the error bars show interquartile ranges. Note that spindle volumes in general are smaller because of methanol fixation. **e**. Cellular levels of CKAP5 after 48 h of differentiation relative to ESCs, probed by western blotting and normalized to tubulin (N = 4 biological replicates). Bars show mean, errors show the standard deviation, circles show replicates. Welch's t-test (two-sided), $P = 0.36$. **f**. As in b) but showing microtubule depolymerase KIF2C/MCAK. $P = 3.7 \times 10^{-3}$. **g**. As in c) but showing staining for KIF2C/MCAK (N = 2 experiments). **h**. As in d) but showing KIF2C/MCAK summed spindle signals (ESCs n = 52 cells and DIF n = 74 cells, pooled from 2 independent experiments).

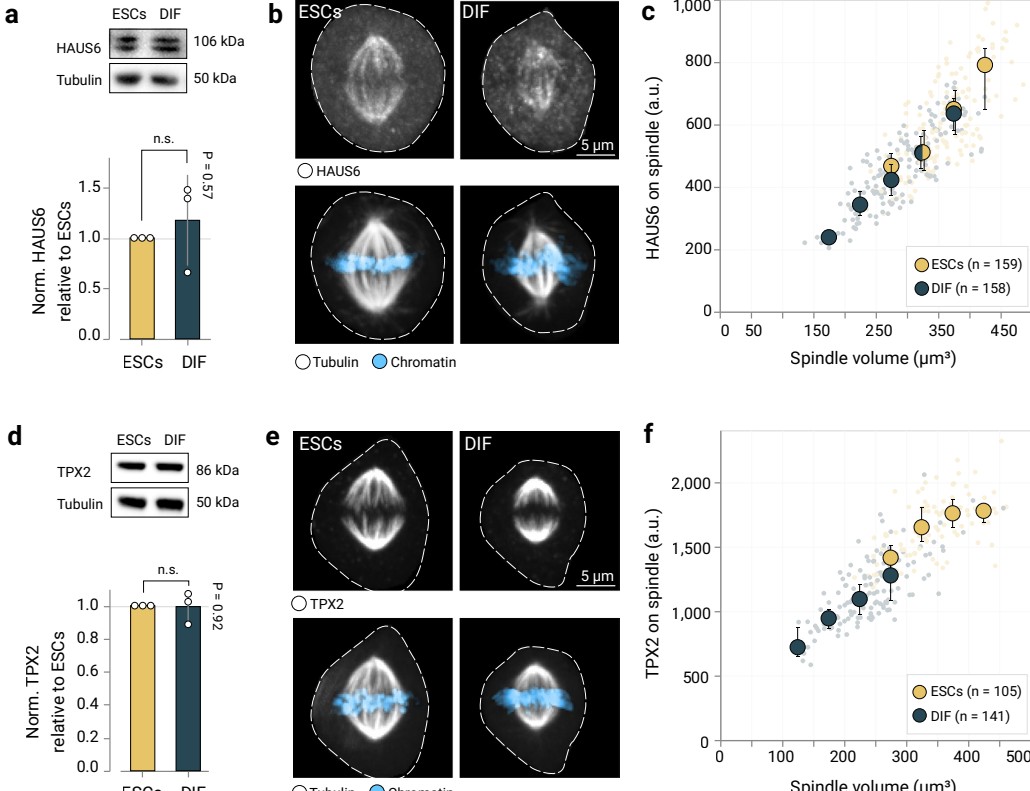

**Extended Data Fig. 4 | Augmin and TPX2 spindle localization is comparable between the differentiation states. a.** Cellular levels of augmin (subunit HAUS6) after 48 h of differentiation relative to ESCs, probed by western blotting and normalized to tubulin (N = 3 biological replicates). Bars show the mean, errors show the standard deviation, circles show replicates. Welch's t-test (two-sided), *P* = 0.57. **b.** Confocal micrographs (maximum projected, N = 2 experiments) of fixed ESCs or DIF at metaphase. Top: Immunostained HAUS6 signal, bottom: tubulin::GFP (grey) and chromatin counterstained by Hoechst (blue). Dotted lines indicate the cell boundaries. Scale bar: 5 µm. **c.** Fluorescent HAUS6 signal on spindle (normalized by total cell HAUS6) as a function of spindle volume. Each data point represents a single cell (ESCs n = 159 cells and DIF n = 158 cells pooled from 2 independent experiments), large circles denote medians in each cell volume bin, error bars show interquartile ranges. **d.** As in a) but showing TPX2 levels (N = 3 biological replicates). *P* = 0.92. **e.** As in b) but showing TPX2 immunofluorescence (N = 3 experiments). **f.** As in c) but showing TPX2 immunofluorescence (ESCs n = 105 cells and DIF n = 141 cells pooled from 3 independent experiments). n.s.: not significant, *P* > 0.05.

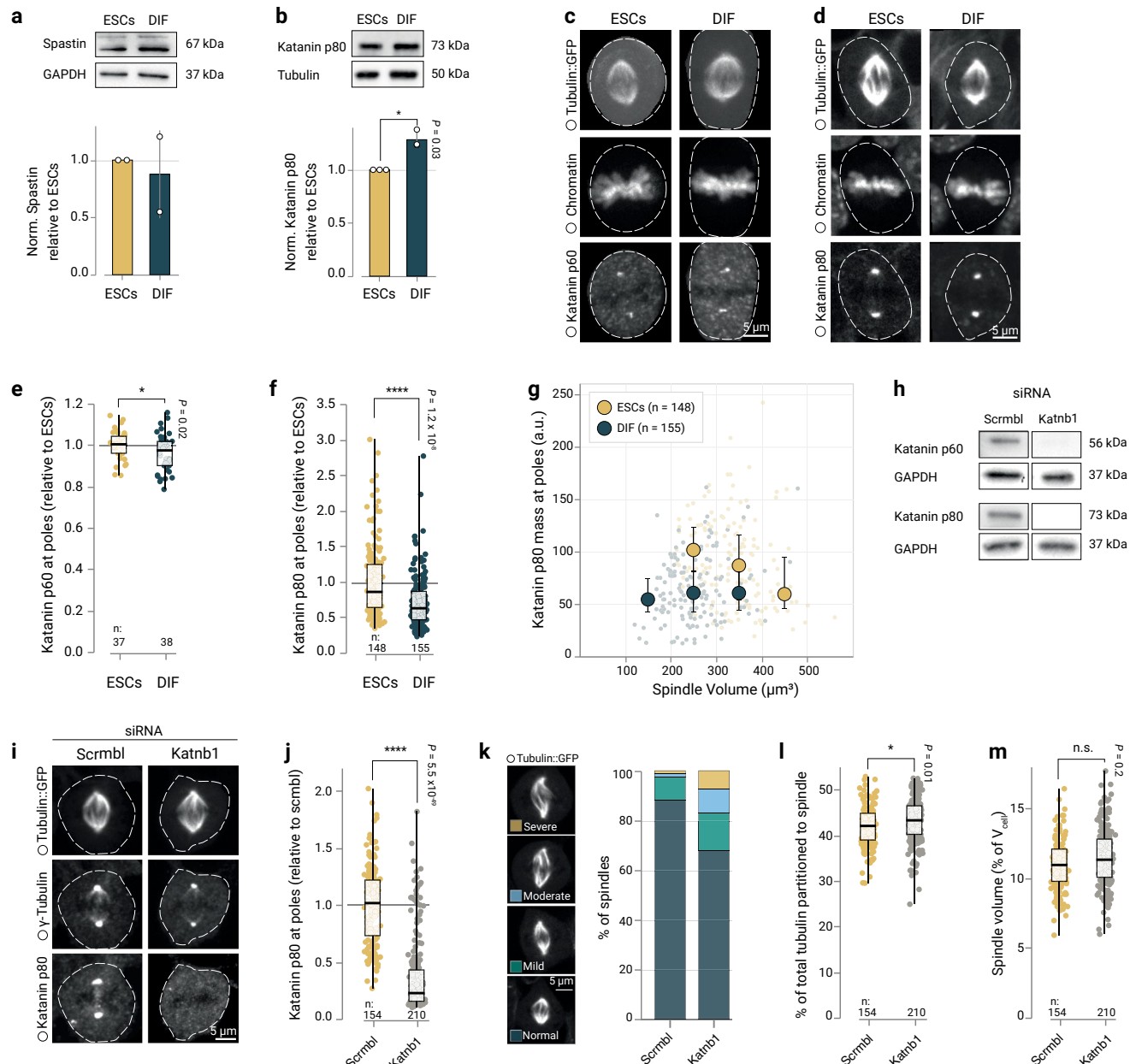

**Extended Data Fig. 5 | Spindle scaling in ESCs is independent of microtubule severing. a**. Cellular levels of spastin after 48 h of differentiation relative to ESCs, probed by western blotting and normalized to tubulin (N = 2, each time loading 2 independent batches of protein extracts). Bars show mean, errors show the standard deviation, circles show replicates. **b**. Cellular levels of katanin (subunit p80) after 48 h of differentiation relative to ESCs, probed by western blotting and normalized to tubulin (N = 3 biological replicates). Bars show mean, errors show standard deviation, circles show replicates. Welch's t-test (two-sided), $P$ = 0.03. **c**. Confocal micrographs (maximum projected, N = 3 experiments) of fixed ESCs or DIF at metaphase. Top: Tubulin::GFP signal, centre: chromatin (Hoechst), bottom: immunostained katanin p60. Dotted lines indicate cell boundaries. Scale bar: 5 μm. **d**. As in (c) but staining katanin p80 (N = 3 experiments). **e**. Average katanin p60 signal at spindle poles. Data points show individual cells (ESCs n = 37 cells and DIF n = 38 cells pooled from 3 independent experiments), boxes denote the interquartile ranges, horizontal lines inside the boxes denote the medians, whiskers show the minimum and maximum. Welch's t-test (two-sided), $P$ = 0.02. **f**. As in (e) but showing katanin p80 (ESCs n = 156 cells and DIF n = 168 cells pooled from 3 independent experiments). Welch's t-test (two-sided), $P$ = 1.2 x $10^{-8}$. **g**. Increased katanin p80 mass in ESCs is independent of spindle volume. Data points show individual cells (ESCs n = 148 cells and DIF n = 155 cells pooled from 3 independent experiments). Large circles show the medians of spindle volume

bins (bin size = 100 μm³), error bars show the interquartile ranges. **h**. Representative western blots (from N = 3 experiments) showing katanin p80 knockdown (and katanin p60 co-depletion, as has been described previously[115]) in ESCs using siRNAs (Katnb1) next to control using scrambled siRNAs (Scrmbl). GAPDH as loading control. **i**. Confocal micrographs (maximum projected, N = 3 experiments) of fixed ESCs after katanin p80 knockdown (Katnb1) or control (Scrmbl). Top: tubulin::GFP, centre: γ-tubulin immunostaining as spindle pole reference, bottom: immunostained katanin p80. Dotted lines indicate cell boundaries. Scale bar: 5 μm. **j**. As in f) but showing Katnb1 siRNA-treated ESCs and control ESCs (Scrmbl) (Katnb1 n = 210 cells and Scrmbl n = 154 cells pooled from 3 independent experiments). Welch's t-test (two-sided), $P$ = 5.5 x $10^{-49}$. **k**. Left: Representative micrographs (maximum projected, N = 3 experiments) of 4 spindle phenotypes in the RNAi experiment. Right: Percentages of the 4 phenotypes in population after Katnb1 siRNA treatment vs. control (Scrmbl). **l**. Fraction of tubulin::GFP partitioned to spindle bulk. Data points show individual cells (Katnb1 n = 210 cells and Scrmbl n = 154 cells from 3 independent experiments), boxes show interquartile ranges, horizontal bars show medians, whiskers show minima and maxima. Welch's t-test (two-sided), $P$ = 0.01. **m**. As in l) but showing spindle volume as a percentage of cell volume. $P$ = 0.2. *: $P$ < 0.05, ****: $P$ < 0.0001, n.s.: not significant, $P$ > 0.05.

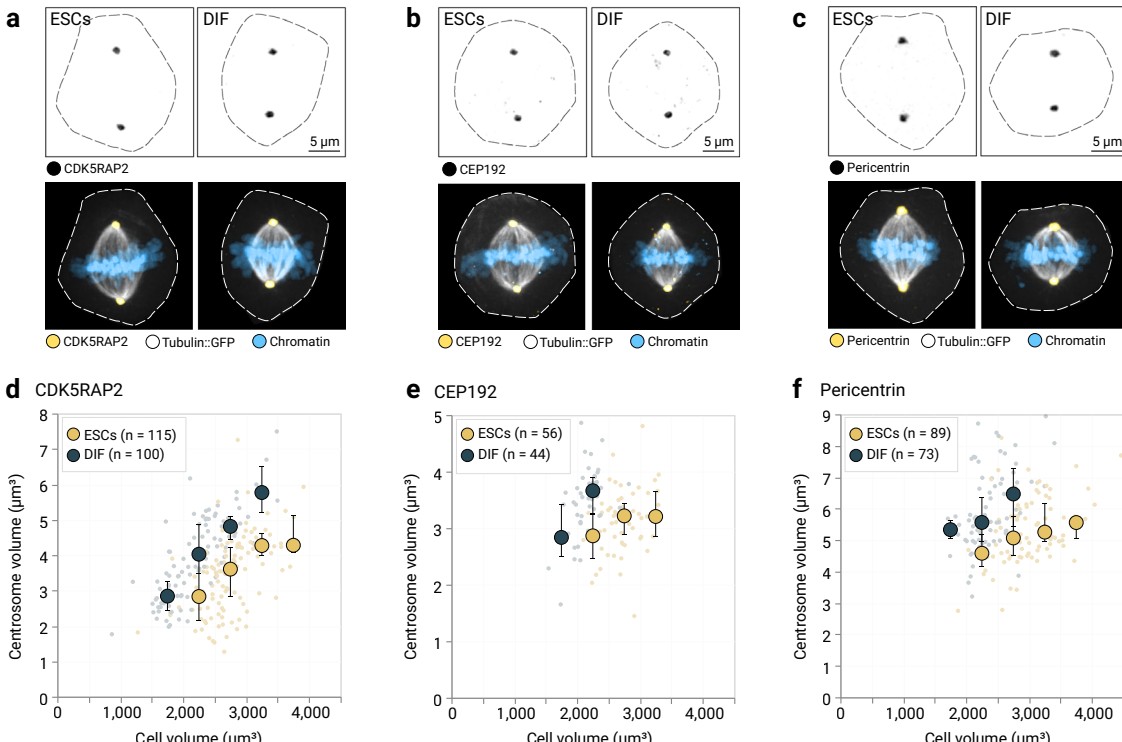

**Extended Data Fig. 6 | The pericentriolar material (PCM) superscales upon differentiation. a**. Confocal micrographs (maximum projected, representative from ESCs n = 113 cells and DIF n = 100 cells pooled from 3 independent experiments) of fixed ESCs or DIF at metaphase. Top: Immunostained CDK5RAP2 signal (inverted grayscale), bottom: immunostained CDK5RAP2 (yellow), tubulin::GFP (grey) and chromatin (Hoechst, blue). Dotted lines indicate cell boundaries. Scale bar: 5 μm. **b**. As in **a**) but using CEP192 (representative from ESCs n = 56 cells and DIF n = 44 cells from 1 experiment). **c**. As in **a**) but using pericentrin (representative from ESCs n = 89 cells and DIF n = 73 cells pooled from 3 experiments). **d**. Scaling relationship between cell volume and centrosome volume based on CDK5RAP2. Data points represent single cells (ESCs n = 113 cells and DIF n = 100 cells pooled from 3 independent experiments), large circles denote medians per cell volume bin, the error bars show interquartile ranges. **e**. As in **d**) but based on CEP192 (ESCs n = 56 cells and DIF n = 44 cells pooled from 1 experiment). **f**. As in **d**) but based on pericentrin (ESCs n = 89 cells and DIF n = 73 cells pooled from 3 experiments).

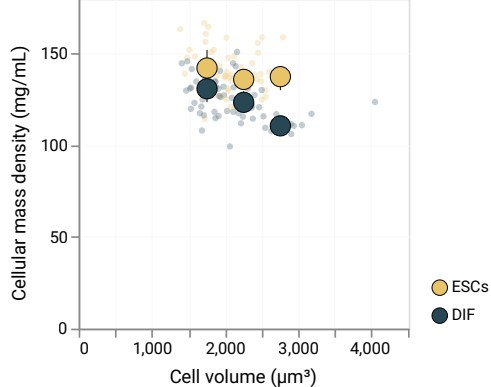

**Extended Data Fig. 7 | The cytoplasm in differentiating cells is diluted across all cell sizes.** Scaling relationship between cell volume and cellular mass density. Data points represent single cells (ESCs n = 61 cells, DIF n = 71 cells from 3 independent experiments, same data as in Fig. 5), the large circles denote the medians in each cell volume bin, the error bars show the interquartile ranges.

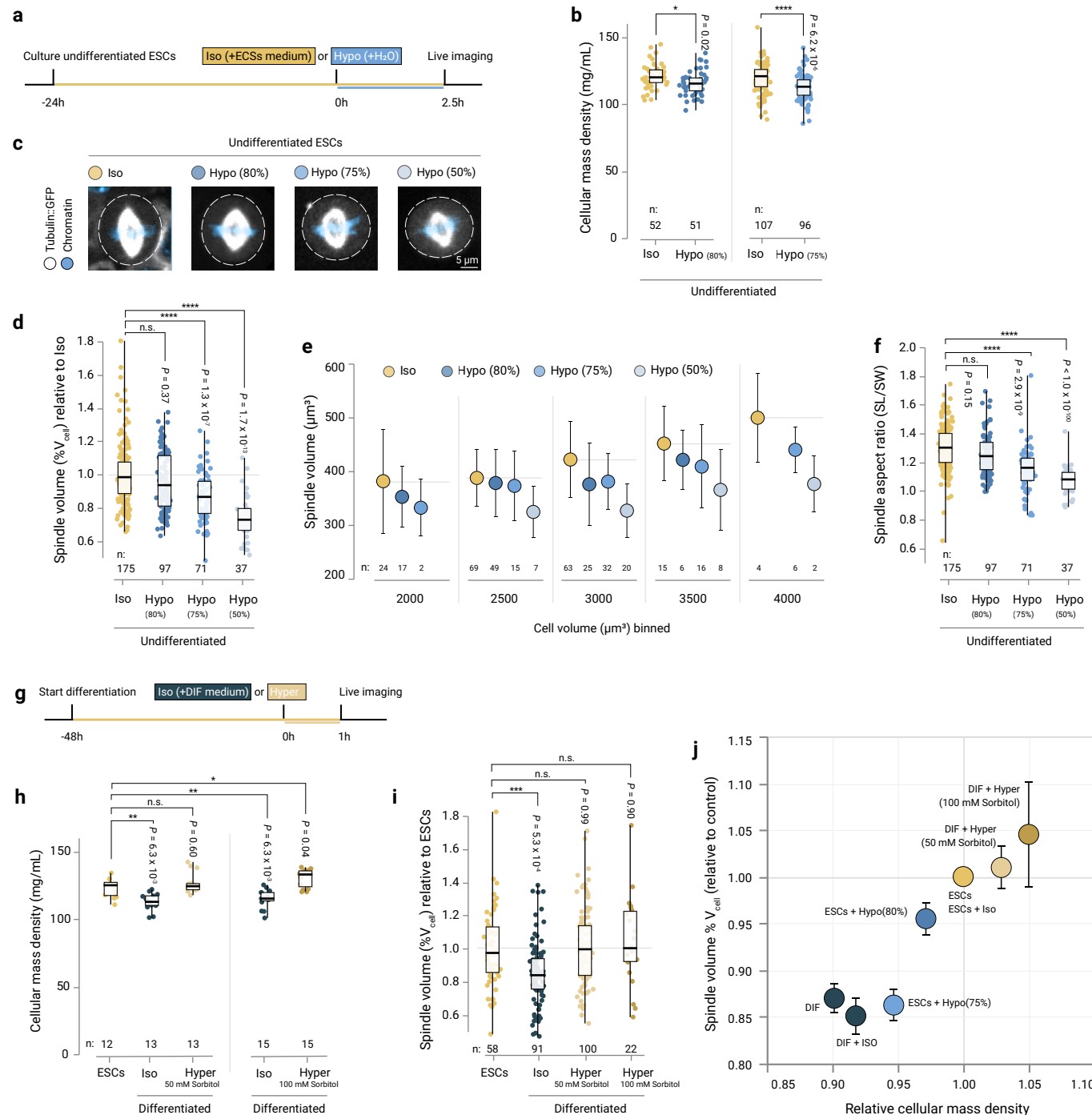

**Extended Data Fig. 8 | See next page for caption.**

**Extended Data Fig. 8 | Biophysical perturbation of cellular mass density modulates spindle scaling. a.** Hypoosmotic treatment of ESCs for live optical diffraction tomography (ODT) or confocal imaging. **b.** ODT-derived cellular mass densities after hypoosmotic treatment (left: 80/20, right: 75/25, % medium/ $H_2O$) of ESCs. Boxes show interquartile ranges, horizontal lines show medians, whiskers show minima and maxima. Data points show individual cells (80/20: n isosmotic: 52 cells, n hypoosmotic: 51 cells pooled from 3 independent experiments, 75/25: n isosmotic: 107 cells, n hypoosmotic: 96 cells pooled from 4 independent experiments). Welch's t-test (two-sided), Iso vs. Hypo (80%): $P = 0.02$; Iso vs. Hypo (75%): $P = 6.2 \times 10^{-6}$. **c.** Confocal live-cell imaging (central slices) of control ESCs (isosmotic treatment) versus increasing hypoosmotic challenges. Tubulin::GFP in grey, chromatin in blue. Dotted lines show cell boundaries. Scale bar: 5 μm. **d.** Cell volume-normalized spindle volumes during hypoosmotic challenge of ESCs, fold change relative to control ESCs (Iso). Boxes show interquartile ranges, horizontal lines show medians, whiskers show minima and maxima. Data points show individual cells (n isosmotic: 175 cells, n Hypo (80%): 97 cells, n Hypo (75%): 71 cells, and n Hypo (50%): 37 cells from 5 independent experiments). Welch's ANOVA and two-sided Games-Howell post-hoc testing, Iso vs. Hypo (80%): $P = 0.37$; Iso vs. Hypo (75%): $P = 1.3 \times 10^{-7}$; Iso vs. Hypo (50%): $P = 1.7 \times 10^{-13}$. **e.** Data from (d) re-analysed, showing spindle volumes within cell volume bins (n = 24, 17, 2, 0; 69, 49, 15, 7; 63, 25, 32, 20; 15, 6, 16, 8; 4, 0, 6, 2 for Iso, Hypo (80%), Hypo (70%), and Hypo (50%) conditions, respectively, binned from 2,000 μm³ through 4,000 μm³ in 500 μm³ bins). Circles show means, error bars show standard deviation. Horizontal grey lines indicate mean value of the control ESCs (Iso). **f.** Boxplots and data as (d) but re-analysed showing spindle aspect ratios (length/width). Welch's ANOVA and two-sided Games-Howell post-hoc testing, Iso vs. Hypo (80%): $P = 0.15$; Iso vs. Hypo (75%): $P = 2.9$

$\times 10^{-9}$; Iso vs. Hypo (50%): $P < 1.0 \times 10^{-100}$. **g.** Hyperosmotic treatment of DIF for live optical diffraction tomography (ODT) or confocal imaging. **h.** ODT-derived cellular mass densities after hyperosmotic treatment (left: +50 mM Sorbitol, right: +100 mM Sorbitol) of DIF, untreated ESCs as additional control. Boxes show the interquartile ranges, horizontal lines show medians, whiskers show minima and maxima. Data points show individual cells (+50 mM Sorbitol: n ESCs: 12 cells, n isosmotic DIF: 13 cells, n hyperosmotic DIF: 13 cells pooled from 1 independent experiment, +100 nM Sorbitol: n isosmotic DIF: 15 cells, n hyperosmotic DIF: 15 cells from 1 independent experiment) Welch's ANOVA and two-sided Games-Howell post-hoc testing, ESCs vs. DIF$_{Iso}$: $P = 6.3 \times 10^{-3}$; ESCs vs. DIF$_{50mM Sorbitol}$: $P = 0.60$; ESCs vs. DIF$_{100mM Sorbitol}$: $P = 0.04$. **i.** Cell volume-normalized spindle volumes during hyperosmotic challenge of DIF (fold change relative to ESCs). Boxes show interquartile ranges, horizontal lines show medians, whiskers show minima and maxima. Data points show individual cells (n ESCs: 58 cells, n isosmotic DIF: 91 cells, n hyperosmotic DIF (50 mM): 100, n hyperosmotic DIF (100 mM): 22 pooled from 2 independent experiments). Welch's ANOVA and two-sided Games-Howell post-hoc testing, ESCs vs. DIF$_{Iso}$: $P = 5.3 \times 10^{-4}$; ESCs vs. DIF$_{50mM Sorbitol}$: $P = 0.99$; ESCs vs. DIF$_{100mM Sorbitol}$: $P = 0.90$. **j.** Composite chart showing the cell volume-normalized spindle volumes (fold changes relative to controls) as a function of the mean cellular mass density (as fold change relative to controls). Circles show means, error bars show standard error of the mean. Data merged from this figure and Fig. 6 (n ESCs: 181 cells and n DIF: 119 cells pooled from 5 independent experiments; n ESCs+Iso: 175, n ESCs+Hypo80%: 97, n ESCs+Hypo75%: 71, n ESCs+Hypo50%: 37 pooled from 5 independent experiments; n DIF+Iso: 91 cells, n DIF+50mM: 100 cells, n DIF+100mM: 22 cells pooled from 2 experiments). *: $P < 0.05$, **: $P < 0.01$, ***: $P < 0.001$, ****: $P < 0.0001$, n.s.: not significant, $P > 0.05$.

# Reporting Summary

## Statistics

For all statistical analyses, confirm that the following items are present in the figure legend, table legend, main text, or Methods section.

| n/a | Confirmed | |
|---|---|---|
| ☐ | ☒ | The exact sample size (*n*) for each experimental group/condition, given as a discrete number and unit of measurement |
| ☐ | ☒ | A statement on whether measurements were taken from distinct samples or whether the same sample was measured repeatedly |
| ☐ | ☒ | The statistical test(s) used AND whether they are one- or two-sided *Only common tests should be described solely by name; describe more complex techniques in the Methods section.* |
| ☐ | ☒ | A description of all covariates tested |
| ☐ | ☒ | A description of any assumptions or corrections, such as tests of normality and adjustment for multiple comparisons |
| ☐ | ☒ | A full description of the statistical parameters including central tendency (e.g. means) or other basic estimates (e.g. regression coefficient) AND variation (e.g. standard deviation) or associated estimates of uncertainty (e.g. confidence intervals) |
| ☐ | ☒ | For null hypothesis testing, the test statistic (e.g. *F*, *t*, *r*) with confidence intervals, effect sizes, degrees of freedom and *P* value noted *Give P values as exact values whenever suitable.* |
| ☒ | ☐ | For Bayesian analysis, information on the choice of priors and Markov chain Monte Carlo settings |
| ☒ | ☐ | For hierarchical and complex designs, identification of the appropriate level for tests and full reporting of outcomes |
| ☐ | ☒ | Estimates of effect sizes (e.g. Cohen's *d*, Pearson's *r*), indicating how they were calculated |

*Our web collection on statistics for biologists contains articles on many of the points above.*

## Software and code

Policy information about availability of computer code

| Data collection | https://git.embl.de/grp-almf/automictools-zenblue-spindle-kletter<br>ZEN 2.3.64.0 (blue edition)<br>KNIME Analytics Platform<br>Leica LAS X Life Science<br>Nikon NIS-Elements AR<br>Image Lab Software 6.1 (Bio-Rad)<br>Image Studio 6.0 (LICORbio) |
|---|---|
| Data analysis | Fiji (ImageJ 2.16.0)(Schindelin et al. 2012)<br>Illastik (1.3.3)(Berg et al. 2019)<br>Python (3.7.1)<br>UniDec (Marty et al. 2015)<br>https://github.com/TobiasKletter/Scaling |

For manuscripts utilizing custom algorithms or software that are central to the research but not yet described in published literature, software must be made available to editors and reviewers. We strongly encourage code deposition in a community repository (e.g. GitHub). See the Nature Portfolio guidelines for submitting code & software for further information.

## Data

Policy information about availability of data

All manuscripts must include a data availability statement. This statement should provide the following information, where applicable:
- Accession codes, unique identifiers, or web links for publicly available datasets
- A description of any restrictions on data availability
- For clinical datasets or third party data, please ensure that the statement adheres to our policy

Source data are provided alongside the paper. Supporting data are available in the BioStudies database (http://www.ebi.ac.uk/biostudies) under accession code S-BIAD1680.
Mass spectrometry data that support the findings of this study have been deposited to the ProteomeXchange Consortium under accession code PXD061228.

## Research involving human participants, their data, or biological material

Policy information about studies with human participants or human data. See also policy information about sex, gender (identity/presentation), and sexual orientation and race, ethnicity and racism.

| | |
|---|---|
| Reporting on sex and gender | Does not apply |
| Reporting on race, ethnicity, or other socially relevant groupings | Does not apply |
| Population characteristics | Does not apply |
| Recruitment | Does not apply |
| Ethics oversight | Does not apply |

Note that full information on the approval of the study protocol must also be provided in the manuscript.

# Field-specific reporting

Please select the one below that is the best fit for your research. If you are not sure, read the appropriate sections before making your selection.

☒ Life sciences ☐ Behavioural & social sciences ☐ Ecological, evolutionary & environmental sciences

For a reference copy of the document with all sections, see nature.com/documents/nr-reporting-summary-flat.pdf

# Life sciences study design

All studies must disclose on these points even when the disclosure is negative.

| | |
|---|---|
| Sample size | No statistical methods were used to predetermine sample size. In retrospective, achieved sample sizes were determined to be adequate based on the magnitude and consistency of measurable differences between groups. |
| Data exclusions | No data points were excluded from the analyses. |
| Replication | All data are from at least 2 up to 9 independent experiments. All attempts at replication were successful. |
| Randomization | No randomisation was performed, since experiments were conducted with independent cell culture populations. |
| Blinding | Data collection and analysis were not performed blind to the conditions of the experiments. |

# Behavioural & social sciences study design

All studies must disclose on these points even when the disclosure is negative.

| | |
|---|---|
| Study description | Briefly describe the study type including whether data are quantitative, qualitative, or mixed-methods (e.g. qualitative cross-sectional, quantitative experimental, mixed-methods case study). |
| Research sample | State the research sample (e.g. Harvard university undergraduates, villagers in rural India) and provide relevant demographic information (e.g. age, sex) and indicate whether the sample is representative. Provide a rationale for the study sample chosen. For studies involving existing datasets, please describe the dataset and source. |

| | |
|---|---|
| Sampling strategy | *Describe the sampling procedure (e.g. random, snowball, stratified, convenience). Describe the statistical methods that were used to predetermine sample size OR if no sample-size calculation was performed, describe how sample sizes were chosen and provide a rationale for why these sample sizes are sufficient. For qualitative data, please indicate whether data saturation was considered, and what criteria were used to decide that no further sampling was needed.* |
| Data collection | *Provide details about the data collection procedure, including the instruments or devices used to record the data (e.g. pen and paper, computer, eye tracker, video or audio equipment) whether anyone was present besides the participant(s) and the researcher, and whether the researcher was blind to experimental condition and/or the study hypothesis during data collection.* |
| Timing | *Indicate the start and stop dates of data collection. If there is a gap between collection periods, state the dates for each sample cohort.* |
| Data exclusions | *If no data were excluded from the analyses, state so OR if data were excluded, provide the exact number of exclusions and the rationale behind them, indicating whether exclusion criteria were pre-established.* |
| Non-participation | *State how many participants dropped out/declined participation and the reason(s) given OR provide response rate OR state that no participants dropped out/declined participation.* |
| Randomization | *If participants were not allocated into experimental groups, state so OR describe how participants were allocated to groups, and if allocation was not random, describe how covariates were controlled.* |

# Ecological, evolutionary & environmental sciences study design

All studies must disclose on these points even when the disclosure is negative.

| | |
|---|---|
| Study description | *Briefly describe the study. For quantitative data include treatment factors and interactions, design structure (e.g. factorial, nested, hierarchical), nature and number of experimental units and replicates.* |
| Research sample | *Describe the research sample (e.g. a group of tagged Passer domesticus, all Stenocereus thurberi within Organ Pipe Cactus National Monument), and provide a rationale for the sample choice. When relevant, describe the organism taxa, source, sex, age range and any manipulations. State what population the sample is meant to represent when applicable. For studies involving existing datasets, describe the data and its source.* |
| Sampling strategy | *Note the sampling procedure. Describe the statistical methods that were used to predetermine sample size OR if no sample-size calculation was performed, describe how sample sizes were chosen and provide a rationale for why these sample sizes are sufficient.* |
| Data collection | *Describe the data collection procedure, including who recorded the data and how.* |
| Timing and spatial scale | *Indicate the start and stop dates of data collection, noting the frequency and periodicity of sampling and providing a rationale for these choices. If there is a gap between collection periods, state the dates for each sample cohort. Specify the spatial scale from which the data are taken* |
| Data exclusions | *If no data were excluded from the analyses, state so OR if data were excluded, describe the exclusions and the rationale behind them, indicating whether exclusion criteria were pre-established.* |
| Reproducibility | *Describe the measures taken to verify the reproducibility of experimental findings. For each experiment, note whether any attempts to repeat the experiment failed OR state that all attempts to repeat the experiment were successful.* |
| Randomization | *Describe how samples/organisms/participants were allocated into groups. If allocation was not random, describe how covariates were controlled. If this is not relevant to your study, explain why.* |
| Blinding | *Describe the extent of blinding used during data acquisition and analysis. If blinding was not possible, describe why OR explain why blinding was not relevant to your study.* |

Did the study involve field work? ☐ Yes ☐ No

# Field work, collection and transport

| | |
|---|---|
| Field conditions | *Describe the study conditions for field work, providing relevant parameters (e.g. temperature, rainfall).* |
| Location | *State the location of the sampling or experiment, providing relevant parameters (e.g. latitude and longitude, elevation, water depth).* |
| Access & import/export | *Describe the efforts you have made to access habitats and to collect and import/export your samples in a responsible manner and in compliance with local, national and international laws, noting any permits that were obtained (give the name of the issuing authority, the date of issue, and any identifying information).* |
| Disturbance | *Describe any disturbance caused by the study and how it was minimized.* |

# Reporting for specific materials, systems and methods

We require information from authors about some types of materials, experimental systems and methods used in many studies. Here, indicate whether each material, system or method listed is relevant to your study. If you are not sure if a list item applies to your research, read the appropriate section before selecting a response.

## Materials & experimental systems

| n/a | Involved in the study |
|---|---|
| ☐ | ☒ Antibodies |
| ☐ | ☒ Eukaryotic cell lines |
| ☒ | ☐ Palaeontology and archaeology |
| ☒ | ☐ Animals and other organisms |
| ☒ | ☐ Clinical data |
| ☒ | ☐ Dual use research of concern |
| ☒ | ☐ Plants |

## Methods

| n/a | Involved in the study |
|---|---|
| ☒ | ☐ ChIP-seq |
| ☒ | ☐ Flow cytometry |
| ☒ | ☐ MRI-based neuroimaging |

## Antibodies

| Antibodies used | Primary antibodies used:<br>CDK5RAP5 (rabbit, Sigma 06-1398)<br>CEP192 (rabbit, Proteintech 18832-1-AP)<br>Pericentrin (rabbit, Abcam 4448)<br>γ-tubulin (mouse, Sigma T6557)<br>Katanin p60 (rabbit, Proteintech 17560-1-AP)<br>Katanin p80 (rabbit, Proteintech 14969-1-AP)<br>CKAP2 (mouse, Proteintech 25486-1-AP)<br>KIF2C/MCAK (rabbit, Abcam ab71706)<br>Alpha-tubulin (mouse, DM1a (Sigma T-6199)<br>Yol1/34 (rat, Bio-rad MCA78G)<br>HAUS6 (rabbit, gift from Laurence Pelletier, University of Toronto, Canada)<br>TPX2 (mouse, gift from Alex Bird, MPI Dortmund, Germany)<br>Oct-4 (rabbit, Proteintech 11263-1-AP)<br>Nestin (mouse, R&D systems MAB2736)<br>PAX6 (mouse, DSHB Cat# pax6, RRID:AB_528427)<br>K40 acetylated tubulin (mouse, Sigma T7451)<br>Polyglutamylation (mouse, AdipoGen GT335)<br>Detyrosinated tubulin (rabbit, Abcam ab48389)<br>alpha/beta-tubulin (rabbit, Cell Signaling technology, #2148)<br>Tbr2 (rabbit, Abcam ab23345, GR153320-1)<br>ZO-1 (rabbit, Invitrogen 617300)<br>AKNA (mouse, gift from Magdalena Götz, LMU Munich, Germany)<br>GAPDH (mouse, Proteintech 60004-1-Ig)<br>TUBB3 (mouse, Nordic BioSite AMB-7318)<br><br>Secondary antibodies:<br>Goat anti-Mouse HRP conjugated (Proteintech 00001-1)<br>Goat anti-Rabbit HRP conjugated (Proteintech 00001-2)<br>Goat anti-Mouse IgG Alexa Fluor 647 (Thermo A-21235)<br>Goat anti-Rabbit IgG Alexa Fluor 647 (Thermo A-21244)<br>Goat anti-Rat IgG Alexa Fluor 647 (Thermo A-21247)<br>Goat anti-Mouse IgG Alexa Fluor 800 (Thermo A-32730) |
|---|---|
| Validation | Primary antibodies:<br>CDK5RAP5: Commercially available polyclonal against human CDK5 regulatory subunit-associated protein 2. Validated by manufacturer as follows: affinity isolated antibody, UNSPSC-Code: 12352203, eCl@ss: 32160702<br>CEP192: Commercially available polyclonal against centrosomal protein 192kDa. Validated by manufacturer as follows: affinity isolated antibody, validated for WB, IP, IHC, IF, ELISA for human and mouse. Cited in 12 publications.<br>Pericentrin: Commercially available rabbit polyclonal against Pericentrin. Validated by manufacturer as follows: affinity isolated antibody suitable for human and mouse samples in immunocytochemistry and immunofluorescence. Cited in 533 publications.<br>γ-tubulin: Commercially available mouse monoclonal against γ-tubulin. Validated by manufacturer as follows: Recognizes an epitope located in the N-terminal amino acids of γ-tubulin and validated for use in immunocytochemistry, indirect ELISA, western blot in human, bovine, dog, hamster, rat, mouse, chicken, and Xenopus.<br>Katanin p60: Commercially available rabbit polyclonal against the katanin p60 (ATPase-containing) subunit A 1. Validated by manufacturer as follows: affinity isolated, validated for use in western blot, immunoprecipitation, immunofluorescence, ELISA in human, mouse, rat. Cited in 25 publications.<br>Katanin p80: Commercially available rabbit polyclonal against the katanin p80 (WD repeat containing) subunit B 1. Validated by manufacturer as follows: affinity isolated, validated for use in western blot, immunoprecipitation, immunofluorescence, ELISA in human, mouse, rat. Cited in 22 publications. |

CKAP2: Commercially available rabbit polyclonal against cytoskeleton associated protein 2. Validated by manufacturer as follows: affinity isolated, validated for use in western blot, immunoprecipitation, immunofluorescence, ELISA in human. Cited in 4 publications.

KIF2C/MCAK: Commercially available rabbit polyclonal MCAK antibody. Validated by manufacturer as follows: suitable for western blot, immunoprecipitation, immunohistochemistry. Immunogen corresponding to synthetic peptide within mouse kinesin-like protein KIF2C. Cited in 3 publications.

Alpha-tubulin: Commercially available mouse monoclonal against alpha-tubulin produced in mouse, clone DM1A, ascites fluid. Validated by manufacturer as follows: suitable for western blot, immunoprecipitation in yeast, mouse, amphibian, human, rat, chicken, fungi, bovine. UNSPSC Code: 12352203, NACRES: NA.41

Yol1/34: Commercially available monoclonal rat anti alpha-tubulin. Validated by manufacturer as follows: prepared by affinity chromatography on Protein G from tissue culture supernatant. Suitable for ELISA, immunoprecipitation, immunohistochemistry, immunocytochemistry, radioimmunoassays, in yeast, mouse, amphibian, human, rat, chicken, fungi, bovine. Cited in 74 publications.

HAUS6: Validated in the referenced source publication.

TPX2: Validated in the referenced source publication.

Oct-4: Commercially available rabbit polyclonal Oct-4 antibody. Validated by manufacturer as follows: Affinity isolated, 11263-1-AP targets OCT4/POU5F1 in WB, IP, ELISA applications and shows reactivity with human, mouse, rat samples. Cited in 258 publications.

Nestin: Commercially available mouse monoclonal Nestin antibody. Validated by manufacturer as follows: Protein A or G purified from hybridoma culture supernatant. Immunogen: E. coli-derived recombinant rat Nestin, Detects mouse and rat Nestin in Western blots. Cited in 28 publications.

PAX6: Commercially available mouse monoclonal PAX6 antibody. Validated by manufacturer as follows: Immunogen: Recombinant partial protein (Chicken, N-terminal region, aa 1-223). Reactivity confirmed for mouse, human, chicken, zebrafish and others. Cited in 747 publications.

Tbr2: Commercially available rabbit polyclonal Tbr2 antibody. Validated by manufacturer as follows: Affinity isolated. Reactivity confirmed for mouse and human samples. Cited in over 530 publications.

ZO-1: Commercially available rabbit polyclonal ZO-1 antibody. Validated by manufacturer as follows: Affinity isolated, confirmed reactivity in avian, bovine, feline, human, rodent, amphibian, … samples. Immunogen: A 69 kD fusion protein corresponding to amino acids 463-1109 of human ZO-1 cDNA. This sequence lies N-terminal to the 80 amino acid region (the alpha-motif) present in the a+-isoform but absent in the a- isoform due to alternative splicing. 61-7300 has been successfully used in Western blot, Immunoprecipitation, Immunofluorescence, ELISA and Immunohistochemistry. Cited in 930 publications.

AKNA: Validated in the referenced source publication.

K40 acetylated tubulin: Commercially available mouse monoclonal K40 antibody. Validated by manufacturer as follows: Affinity isolated, confirmed reactivity in bovine, frog, invertebrates, human ,hamster, mouse, and others. The antibody recognizes an epitope located on the α3 isoform of Chlamydomonas axonemal α-tubulin, within four residues of Lys40 when this amino acid is acetylated. Monoclonal Anti-Acetylated Tubulin antibody produced in mouse has been used in immunofluorescence, Western blot, immunocytochemistry, ELISA, etc.

Polyglutamylation: Commercially available mouse monoclonal Polyglutamylated tubulin antibody. Purified from concentrated hybridoma tissue culture supernatant. Protein G-affinity purified. Immunogen: Octapeptide EGEGE*EEG, modified by the addition of two glutamyl units onto the fifth E (indicated by an asterisk). Reactivity confirmed in human, mouse, rat, and others. Used in ICC-IF, ICC, IHC, WB applications. Cited in 115 publications.

Detyrosinated alpha tubulin: Commercially available rabbit polyclonal detyrosinated tubulin antibody. Suitable for WB and reacts with Human samples. Immunogen corresponding to Synthetic Peptide within Human TUBA4A. Cited in 89 publications.

GAPDH: Commercially available mouse monoclonal GAPDH antibody. Validated by manufacturer as follows: Protein A purification, 60004-1-Ig targets GAPDH in WB, IHC, IF/ICC, FC (Intra), IP, CoIP, ELISA applications and shows reactivity with human, mouse, rat, pig, zebrafish, yeast, plant samples. Cited in 12333 publications.

alpha/beta-tubulin: Commercially available rabbit polyclonal tubulin antibody. Purified by protein A and peptide affinity chromatography. Used in Western blotting, immunohistochemistry, immunofluorescence, flow cytometry. Reactivity demonstrated for human, mouse, rabbit, zebrafish samples, and others. Cited in 793 publications.

TUBB3: Commercially available mouse monoclonal tubulin beta III antibody. Validated by manufacturer as follows: Immunogen: A synthetic peptide corresponding to amino acids 443-450 (ESEAQGPK) of human class III beta-tubulin conjugated to KLH. Reactivity demonstrated in rat, mouse, avian, human, bovine samples, and others. Used in ELISA, immunocytochemistry, western blot applications. Cited by 532 publications.

Secondary antibody
Goat anti-Mouse IgG Alexa Fluor 647, goat anti-Rabbit IgG Alexa Fluor 647, goat anti-Rat IgG Alexa Fluor 647 and Goat anti-Mouse IgG Alexa Fluor 800: Commercially available secondary antibody, validated by manufacturer and 1844, 1779, 1195 and 93 citations, respectively.

Goat anti-Mouse HRP conjugated: Commercially available goat polyclonal anti-mouse secondary antibody, conjugated with Horse Raddish Peroxidase. The antibody was purified from antisera by immunoaffinity chromatography using antigens coupled to agarose beads. Confirmed for ELISA, Western blot, Dot blot applications. Cited in 7528 publications.

Goat anti-Rabbit HRP conjugated: Commercially available goat polyclonal anti-rabbit secondary antibody, conjugated with Horse Raddish Peroxidase. The antibody was purified from antisera by immunoaffinity chromatography using antigens coupled to agarose beads. Confirmed for ELISA, Western blot, Dot blot applications. Cited in 10181 publications.

# Eukaryotic cell lines

*Policy information about* cell lines and Sex and Gender in Research

| Cell line source(s) | R1/E mouse embryonic stem cells (ESCs) stably transfected with bacterial artificial chromosomes harbouring the eGFP-fused coding region of human β5-tubulin and its regulatory sequences for native expression levels were a gift from the Hyman lab, MPI Dresden. Validation and characterisation in Poser, I., Sarov, M., Hutchins, J. R., Hériché, J. K., Toyoda, Y., Pozniakovsky, A., ... & Hyman, A. A. (2008). BAC TransgeneOmics: a high-throughput method for exploration of protein function in mammals. Nature methods, 5(5), 409-415 (PMC2871289). |
|---|---|
| Authentication | None of the cell lines were authenticated during the course of this study. |

| Mycoplasma contamination | Cell lines regularly tested negative for mycoplasma contamination. |
|---|---|
| Commonly misidentified lines (See ICLAC register) | None of the cell lines used in this study are in the ICLAC register. We have not used any of the misidentified lines. |

# Palaeontology and Archaeology

| Specimen provenance | *Provide provenance information for specimens and describe permits that were obtained for the work (including the name of the issuing authority, the date of issue, and any identifying information). Permits should encompass collection and, where applicable, export.* |
|---|---|
| Specimen deposition | *Indicate where the specimens have been deposited to permit free access by other researchers.* |
| Dating methods | *If new dates are provided, describe how they were obtained (e.g. collection, storage, sample pretreatment and measurement), where they were obtained (i.e. lab name), the calibration program and the protocol for quality assurance OR state that no new dates are provided.* |

☐ Tick this box to confirm that the raw and calibrated dates are available in the paper or in Supplementary Information.

| Ethics oversight | *Identify the organization(s) that approved or provided guidance on the study protocol, OR state that no ethical approval or guidance was required and explain why not.* |
|---|---|

Note that full information on the approval of the study protocol must also be provided in the manuscript.

# Animals and other research organisms

Policy information about studies involving animals; ARRIVE guidelines recommended for reporting animal research, and Sex and Gender in Research

| Laboratory animals | *For laboratory animals, report species, strain and age OR state that the study did not involve laboratory animals.* |
|---|---|
| Wild animals | *Provide details on animals observed in or captured in the field; report species and age where possible. Describe how animals were caught and transported and what happened to captive animals after the study (if killed, explain why and describe method; if released, say where and when) OR state that the study did not involve wild animals.* |
| Reporting on sex | *Indicate if findings apply to only one sex; describe whether sex was considered in study design, methods used for assigning sex. Provide data disaggregated for sex where this information has been collected in the source data as appropriate; provide overall numbers in this Reporting Summary. Please state if this information has not been collected. Report sex-based analyses where performed, justify reasons for lack of sex-based analysis.* |
| Field-collected samples | *For laboratory work with field-collected samples, describe all relevant parameters such as housing, maintenance, temperature, photoperiod and end-of-experiment protocol OR state that the study did not involve samples collected from the field.* |
| Ethics oversight | *Identify the organization(s) that approved or provided guidance on the study protocol, OR state that no ethical approval or guidance was required and explain why not.* |

Note that full information on the approval of the study protocol must also be provided in the manuscript.

# Clinical data

Policy information about clinical studies
All manuscripts should comply with the ICMJE guidelines for publication of clinical research and a completed CONSORT checklist must be included with all submissions.

| Clinical trial registration | *Provide the trial registration number from ClinicalTrials.gov or an equivalent agency.* |
|---|---|
| Study protocol | *Note where the full trial protocol can be accessed OR if not available, explain why.* |
| Data collection | *Describe the settings and locales of data collection, noting the time periods of recruitment and data collection.* |
| Outcomes | *Describe how you pre-defined primary and secondary outcome measures and how you assessed these measures.* |

# Dual use research of concern

Policy information about dual use research of concern

## Hazards

Could the accidental, deliberate or reckless misuse of agents or technologies generated in the work, or the application of information presented in the manuscript, pose a threat to:

No | Yes
☒ ☐ Public health
☒ ☐ National security
☒ ☐ Crops and/or livestock
☒ ☐ Ecosystems
☒ ☐ Any other significant area

## Experiments of concern

Does the work involve any of these experiments of concern:

No | Yes
☒ ☐ Demonstrate how to render a vaccine ineffective
☒ ☐ Confer resistance to therapeutically useful antibiotics or antiviral agents
☒ ☐ Enhance the virulence of a pathogen or render a nonpathogen virulent
☒ ☐ Increase transmissibility of a pathogen
☒ ☐ Alter the host range of a pathogen
☒ ☐ Enable evasion of diagnostic/detection modalities
☒ ☐ Enable the weaponization of a biological agent or toxin
☒ ☐ Any other potentially harmful combination of experiments and agents

# Plants

| | |
|---|---|
| Seed stocks | *Report on the source of all seed stocks or other plant material used. If applicable, state the seed stock centre and catalogue number. If plant specimens were collected from the field, describe the collection location, date and sampling procedures.* |
| Novel plant genotypes | *Describe the methods by which all novel plant genotypes were produced. This includes those generated by transgenic approaches, gene editing, chemical/radiation-based mutagenesis and hybridization. For transgenic lines, describe the transformation method, the number of independent lines analyzed and the generation upon which experiments were performed. For gene-edited lines, describe the editor used, the endogenous sequence targeted for editing, the targeting guide RNA sequence (if applicable) and how the editor was applied.* |
| Authentication | *Describe any authentication procedures for each seed stock used or novel genotype generated. Describe any experiments used to assess the effect of a mutation and, where applicable, how potential secondary effects (e.g. second site T-DNA insertions, mosiacism, off-target gene editing) were examined.* |

# ChIP-seq

## Data deposition

☐ Confirm that both raw and final processed data have been deposited in a public database such as GEO.

☐ Confirm that you have deposited or provided access to graph files (e.g. BED files) for the called peaks.

| | |
|---|---|
| Data access links<br>*May remain private before publication.* | *For "Initial submission" or "Revised version" documents, provide reviewer access links. For your "Final submission" document, provide a link to the deposited data.* |
| Files in database submission | *Provide a list of all files available in the database submission.* |
| Genome browser session<br>(e.g. UCSC) | *Provide a link to an anonymized genome browser session for "Initial submission" and "Revised version" documents only, to enable peer review. Write "no longer applicable" for "Final submission" documents.* |

## Methodology

| | |
|---|---|
| Replicates | *Describe the experimental replicates, specifying number, type and replicate agreement.* |
| Sequencing depth | *Describe the sequencing depth for each experiment, providing the total number of reads, uniquely mapped reads, length of reads and whether they were paired- or single-end.* |
| Antibodies | *Describe the antibodies used for the ChIP-seq experiments; as applicable, provide supplier name, catalog number, clone name, and lot number.* |

| Peak calling parameters | *Specify the command line program and parameters used for read mapping and peak calling, including the ChIP, control and index files used.* |
| Data quality | *Describe the methods used to ensure data quality in full detail, including how many peaks are at FDR 5% and above 5-fold enrichment.* |
| Software | *Describe the software used to collect and analyze the ChIP-seq data. For custom code that has been deposited into a community repository, provide accession details.* |

# Flow Cytometry

## Plots

Confirm that:

☐ The axis labels state the marker and fluorochrome used (e.g. CD4-FITC).

☐ The axis scales are clearly visible. Include numbers along axes only for bottom left plot of group (a 'group' is an analysis of identical markers).

☐ All plots are contour plots with outliers or pseudocolor plots.

☐ A numerical value for number of cells or percentage (with statistics) is provided.

## Methodology

| Sample preparation | *Describe the sample preparation, detailing the biological source of the cells and any tissue processing steps used.* |
| Instrument | *Identify the instrument used for data collection, specifying make and model number.* |
| Software | *Describe the software used to collect and analyze the flow cytometry data. For custom code that has been deposited into a community repository, provide accession details.* |
| Cell population abundance | *Describe the abundance of the relevant cell populations within post-sort fractions, providing details on the purity of the samples and how it was determined.* |
| Gating strategy | *Describe the gating strategy used for all relevant experiments, specifying the preliminary FSC/SSC gates of the starting cell population, indicating where boundaries between "positive" and "negative" staining cell populations are defined.* |

☐ Tick this box to confirm that a figure exemplifying the gating strategy is provided in the Supplementary Information.

# Magnetic resonance imaging

## Experimental design

| Design type | *Indicate task or resting state; event-related or block design.* |
| Design specifications | *Specify the number of blocks, trials or experimental units per session and/or subject, and specify the length of each trial or block (if trials are blocked) and interval between trials.* |
| Behavioral performance measures | *State number and/or type of variables recorded (e.g. correct button press, response time) and what statistics were used to establish that the subjects were performing the task as expected (e.g. mean, range, and/or standard deviation across subjects).* |

## Acquisition

| Imaging type(s) | *Specify: functional, structural, diffusion, perfusion.* |
| Field strength | *Specify in Tesla* |
| Sequence & imaging parameters | *Specify the pulse sequence type (gradient echo, spin echo, etc.), imaging type (EPI, spiral, etc.), field of view, matrix size, slice thickness, orientation and TE/TR/flip angle.* |
| Area of acquisition | *State whether a whole brain scan was used OR define the area of acquisition, describing how the region was determined.* |

Diffusion MRI   ☐ Used   ☐ Not used

## Preprocessing

| Preprocessing software | *Provide detail on software version and revision number and on specific parameters (model/functions, brain extraction, segmentation, smoothing kernel size, etc.).* |

| Normalization | *If data were normalized/standardized, describe the approach(es): specify linear or non-linear and define image types used for transformation OR indicate that data were not normalized and explain rationale for lack of normalization.* |
|---|---|
| Normalization template | *Describe the template used for normalization/transformation, specifying subject space or group standardized space (e.g. original Talairach, MNI305, ICBM152) OR indicate that the data were not normalized.* |
| Noise and artifact removal | *Describe your procedure(s) for artifact and structured noise removal, specifying motion parameters, tissue signals and physiological signals (heart rate, respiration).* |
| Volume censoring | *Define your software and/or method and criteria for volume censoring, and state the extent of such censoring.* |

## Statistical modeling & inference

| Model type and settings | *Specify type (mass univariate, multivariate, RSA, predictive, etc.) and describe essential details of the model at the first and second levels (e.g. fixed, random or mixed effects; drift or auto-correlation).* |
|---|---|
| Effect(s) tested | *Define precise effect in terms of the task or stimulus conditions instead of psychological concepts and indicate whether ANOVA or factorial designs were used.* |

Specify type of analysis: ☐ Whole brain ☐ ROI-based ☐ Both

| Statistic type for inference | *Specify voxel-wise or cluster-wise and report all relevant parameters for cluster-wise methods.* |
|---|---|

(See Eklund et al. 2016)

| Correction | *Describe the type of correction and how it is obtained for multiple comparisons (e.g. FWE, FDR, permutation or Monte Carlo).* |
|---|---|

## Models & analysis

| n/a | Involved in the study |
|---|---|
| ☐ | ☐ Functional and/or effective connectivity |
| ☐ | ☐ Graph analysis |
| ☐ | ☐ Multivariate modeling or predictive analysis |

| Functional and/or effective connectivity | *Report the measures of dependence used and the model details (e.g. Pearson correlation, partial correlation, mutual information).* |
|---|---|
| Graph analysis | *Report the dependent variable and connectivity measure, specifying weighted graph or binarized graph, subject- or group-level, and the global and/or node summaries used (e.g. clustering coefficient, efficiency, etc.).* |
| Multivariate modeling and predictive analysis | *Specify independent variables, features extraction and dimension reduction, model, training and evaluation metrics.* |

