## [Peer Review File · Nature Cell Biology]

Cell State-Specific Cytoplasmic Density Controls Spindle Architecture and Scaling

Corresponding Author: Professor Simone Reber

Version 0:

Decision Letter:

Revise extended OD

*Please delete the link to your author homepage if you wish to forward this email to co-authors.

Dear Professor Reber,

I am writing on behalf of my colleague Dr Daryl David who is currently out of the office. We apologize once again for the delay. As mentioned previously, our previous reviewer with expertise in computational modelling could not review your manuscript; we had since recruited a new reviewer (Reviewer #3) to assess your manuscript.

Your manuscript, "Cell State-Specific Cytoplasmic Material Properties Control Spindle Architecture and Scaling", has now been seen by 3 referees, who are experts in cell scaling (referee 1); cell division (referee 2); and mitotic spindle (referee 3). As you will see from their comments (attached below) they find this work of potential interest, but have raised substantial concerns, which in our view would need to be addressed with considerable revisions before we can consider publication in Nature Cell Biology.

Nature Cell Biology editors discuss the referee reports in detail within the editorial team, including the chief editor, to identify key referee points that should be addressed with priority, and requests that are overruled as being beyond the scope of the current study. To guide the scope of the revisions, I have listed these points below. We are committed to providing a fair and constructive peer-review process, so please feel free to contact me if you would like to discuss any of the referee comments further.

In particular, it would be essential to:

A.) Experimentally assess role of and effects on microtubule dynamics; this includes effects of polymerization and dynamics (all Reviewers), effects of MT binding proteins (Reviewers #2 and #3), and MT severing proteins (Reviewers #2 and #3).

B.) Provide further experimental data on potential effects of cytoplasmic dilution (shared by all Reviewers, however, please do see comments from Reviewers #2 and #3)

C.) Explain and provide rationale for the measurement of spindle size (Reviewers #2 and #3) with orthogonal methods of analysis given the current use of overexpression of markers (Reviewer #2)

D.) Justify the underlying assumptions of the mathematical model, taking into account the various MT assemblies and free tubulin in the cytoplasm (Reviewer #3).

) All other referee concerns pertaining to strengthening existing data, providing controls, methodological details, clarifications and textual changes, should also be addressed.

) Finally please pay close attention to our guidelines on statistical and methodological reporting (listed below) as failure to do so may delay the reconsideration of the revised manuscript. In particular please provide:

We would be happy to consider a revised manuscript that would satisfactorily address these points, unless a similar paper is published elsewhere, or is accepted for publication in Nature Cell Biology in the meantime.

- ensure that it conforms to our format instructions and publication policies (see below and www.nature.com/nature/authors/).

- provide a point-by-point rebuttal to the full referee reports verbatim, as provided at the end of this letter.

- provide the completed Editorial Policy Checklist (found here <https://www.nature.com/authors/policies/Policy.pdf>), and Reporting Summary (found here <https://www.nature.com/authors/policies/ReportingSummary.pdf>). This is essential for reconsideration of the manuscript and these documents will be available to editors and referees in the event of peer review. For more information see <http://www.nature.com/authors/policies/availability.html> or contact me.

Nature Cell Biology is committed to improving transparency in authorship. As part of our efforts in this direction, we are now requesting that all authors identified as 'corresponding author' on published papers create and link their Open Researcher and Contributor Identifier (ORCID) with their account on the Manuscript Tracking System (MTS), prior to acceptance. ORCID helps the scientific community achieve unambiguous attribution of all scholarly contributions. You can create and link your ORCID from the home page of the MTS by clicking on 'Modify my Springer Nature account'. For more information please visit <http://www.springernature.com/orcid>.

Link Redacted

We would like to receive a revised submission within six months. We would be happy to consider a revision even after this timeframe, however if the resubmission deadline is missed and the paper is eventually published, the submission date will be the date when the revised manuscript was received.

We hope that you will find our referees' comments, and editorial guidance helpful. Please do not hesitate to contact me if there is anything you would like to discuss.

Best wishes,

Sabrya Carim

Sabrya Carim, PhD
(she/her/hers)
Associate Editor, Nature Cell Biology
Nature Portfolio

Springer Nature
The Campus, 4 Crinan Street, London N1 9XW, UK
sabrya.carim@springernature.com
<https://orcid.org/0000-0001-9485-1938>

Reviewers' Comments:

Reviewer #1:
Remarks to the Author:

Kletter et al.

The authors set out to investigate mitotic spindle morphology during differentiation of neural stem cells and utilize a powerful long-term microscopy method as well as microtubule dynamics measurements, phase imaging, and quantitative biochemistry. They reveal a mechanism by which spindles shift in size and morphology as the cytoplasm becomes more dilute and identify a protein that contributes to the shift in microtubule nucleation from the spindle bulk to centrosomes.

This is an exceptionally nice paper. The observations are interesting, many different types of high quality data are presented and rigorously analyzed. The CPAP mechanism elucidated is convincing and of high significance. I believe this will be a landmark paper in the field. My comments on the text are minor.

It should be stated somewhere in the main text how cell volumes were measured.

Line 174: it would be useful for many readers to indicate that CKAP5 belongs to the XMAP215/TOG family. Is it neural-specific?

While it would be difficult to measure other parameters of microtubule dynamics, is it predicted that catastrophe frequency increases?

Line 216: perhaps "relative volume occupied by centrosomal proteins"

Line 232-233: "persisting centrosomes from large cells" is unclear and unrelated to the previous sentence. Was the question whether the cells were about to enter mitosis?

Line 271 and 290: May be a bit confusing, need to clearly distinguish microtubule density within the spindle bulk and "at the expense of spindle bulk". The size decreases but not the microtubule density.

Reviewer #2:

Remarks to the Author:

In this paper, Kletter and colleagues investigate the changes in spindle scaling during ESC differentiation towards the neural lineage. They find that spindle size scales with cell size, in accordance with the literature in other model systems, but that the relative spindle size/cell size changes during differentiation. They show convincingly that during differentiation nucleation of astral microtubules increases at the centrosome at the expense of the spindle bulk, due to a general dilution of the cytoplasm. They show that diluting the cytoplasm and freeing CPAP from tubulin both phenocopy differentiation in terms of spindle scaling and nucleation distribution. This paper is very elegant and nicely written, the figures are clear and beautiful. The question is interesting, and the approach is solid. It opens many fascinating questions that are outside of the scope of the paper, such as: why is the cytoplasm diluted in differentiated cells? Is it more important for a differentiated cells to have astral microtubules (maybe for spindle positioning)? Overall, I think the paper proposes a new mechanism for spindle scaling in a developmental context, and that the data presented is strong. I have a couple of comments/questions which I think would make the paper even stronger, but overall I am happy to recommend this paper for publication.

Major comments:

Several piece of crucial data relies on overexpression of markers. To be fully convincing, I think the authors should consider validating some of their finding using other labelling methods. For example, I think the key piece of data from Fig1 could be easily replicated using a microtubule dye eg SPY or SiR-tubulin.

Similarly, Fig2a would benefit from being repeated with an endogenous tagging of EB3, although I realize this is a lot more challenging thus should not condition the acceptance of the paper.

I do not understand what figure 3f-g bring. How does the fact that cell cycle duration changes during differentiation (also shown by doi: 10.1038/s41598-019-44537-0) demonstrates that the enlarged centrosomes are not remnants from larger stem cells. I am also not totally sure what a "persisting centrosome from larger stem cells" would be: are the authors implying that this cell would either not have divided at all/this would be a non differentiated ESC in a population of DIF cells ? Could the authors clarify what they hypothesis is here?

The authors look at microtubule severing using Katanin. They report differences between relative amounts of p80 between ESCs and DIF. What about p60, are there differences in relative amounts? Also, to conclude on the lack of role of microtubule severing, the authors should also look at Spastin (the authors could use a similar approach with siRNA or a Spastin inhibitor such as the one described here doi:10.1038/s41589-019-0225-6)

As far as I can tell, the authors did not show that diluting the cytoplasm actually leads to liberation of CPAP in this model system, this is solely based on literature and the fact that the 2 phenotypes phenocopy. I think this is fine, but the title of figure 6 should probably be toned down accordingly, as well as some aspects of the discussion (eg line 376).

Minor comments:

The authors see that cell volume is higher in ESC than their differentiated counterparts. However, the general belief is that differentiated cells are overall bigger than stem cells. Does this mean that stem cells swell more during mitosis than differentiated cells, or that this idea is a mere cliché, or perhaps due to previous non accurate measurements in 3D vs 2D? (this does not need to be added to the paper, I would just be curious to hear the authors' opinion).

I am not sure why the authors focus on the microtubule polymerase CKAP5 specifically. Could the authors justify this choice, or have a more systematic approach?

In supplementary 5i,j, should the authors be looking at scaling rather than volume?

Reviewer #3:

Remarks to the Author:

Kletter et al., 2024

Mitotic spindles vary significantly in size and morphology, often adapting to the specific size and function of different cell types. However, the biological mechanisms that drive these morphological adaptations remain poorly understood. In this manuscript, Kletter and colleagues explore spindle size scaling in mouse embryonic stem cells (ESCs) undergoing neural differentiation. One chief finding is that spindles are smaller in early differentiated (DIF) cells compared to pluripotent stem cells, with DIF cells exhibiting larger centrosomes and more abundant astral microtubules (MTs). Using optical diffraction tomography, the authors demonstrate that the cytoplasm of neurally differentiated cells is more dilute, resulting in lower free tubulin concentration. This decrease in tubulin may release CPAP from its binding to free tubulin, promoting centrosome growth and spindle scaling. Therefore, the study's significance lies in its proposal that cytoplasmic dilution serves as a key regulator of organelle size scaling during stem cell differentiation.

Overall, the manuscript is well-executed, incorporating several innovative quantitative microscopy techniques, numerous controls for tubulin biochemistry and spindle regulators, and mathematical models that support the conclusions. The paper is also generally well-written and presented. However, this reviewer finds that the main claim—that "cytoplasmic material properties control spindle architecture and scaling"—should be better substantiated before publication. Below are some major and minor concerns:

Spindle "Size" Definition

The authors use 3D volume to assess spindle size scaling. The rationale for this choice, as well as the definition of volume, should be clarified. Visual inspection suggests that spindles in DIF cells are wider and shorter, with differences in MT distribution compared to ESCs. Previous studies on spindle scaling have often used pole-to-pole length or chromosome plate width to measure spindle morphology, which should be quantified here as well. Given that spindles are porous structures with visible voids, the meaning of spindle volume remains ambiguous—does it represent the number of MTs, their mass, etc.? Additionally, why were astral MTs not included as part of spindle volume measurement?

Microtubule Regulation

The authors show that centrosomes are larger in DIF cells, potentially promoting astral MT nucleation at the expense of bulk MTs, thus reducing spindle size. However, it is unclear why larger centrosomes would only promote outward nucleation rather than inward. Moreover, the relationship between reduced bulk MTs and the formation of shorter, wider spindles needs clarification. The nucleation of MTs from chromosomes is another significant pathway, but its role in spindle scaling during differentiation is not examined. Also, while the authors note that MT growth speed is similar in ESCs and DIF cells, other aspects of MT dynamics, such as depolymerization speeds or catastrophe frequency, may play an important role. Finally, several depolymerizing kinesins (e.g., MCAK, Kif18b) that are known to influence spindle size are not tested here.

Cytoplasmic Dilution

Figure 6 is a central figure, as it aims to demonstrate that cytoplasmic dilution affects spindle size by influencing CPAP's impact on centrosome size. This aspect of the study should be expanded to provide more robust support for the paper's main claim. First, the authors should test a range of hypoosmotic shocks to determine whether spindle size declines linearly with cytoplasm concentration or exhibits a saturating behavior. Second, hyperosmotic shocks should be applied to concentrate the cytoplasm in DIF cells to see if this rescues spindle size. Third, to directly validate their model, the authors should attempt to reverse the effects of hypoosmotic shock by overexpressing tubulin and/or downregulating CPAP. Indeed, it is plausible that CPAP manipulation (via CCB02) and cytoplasmic dilution influence spindle size by very separate mechanisms, so further experiments are needed. Additionally, examining CPAP localization under different conditions could strengthen the proposed model.

Mathematical Model

The mathematical model presented is based on a Michaelis-Menten kinetics to explain the differences in astral MT numbers between ESCs and DIF cells, drawing on prior work by Rieckoff et al. However, Rieckoff's model applied this law to all MTs, not just astral MTs. Here, the authors assume that the total number of MTs remains constant, which is surprising given their claim that free tubulin concentration decreases in differentiated cells. Moreover, the potential impact of cytoplasm dilution on diffusion rates of tubulin and CPAP, as discussed in Molines et al. (2022), is not addressed here.

Minor Comments:

- Figure 2c-d: The region of interest (ROI) used to measure bulk EB1 signal should encompass the entire spindle width, as the images suggest that MTs are more concentrated at the spindle edges. The normalization in Figure 2d obscures the main effect, which is an increase in EB1 signal at centrosomes rather than a decrease in the spindle bulk. Please revise.
- Figure 4c: The presented blots appear to show increased deetyrosination levels, contradicting the authors' claim of no change. Please provide quantification to support this conclusion.
- Figure 5c: The quantification of tubulin-GFP covers the entire image, but it would be essential to separately document the levels of free tubulin versus polymerized tubulin.
- Figure 6j: The spindle volume is normalized by cell size, raising concerns that the observed hypoosmotic shock phenotypes may result from an increase in cell size rather than a reduction in spindle size. Please provide spindle size quantification similar to that in Figure 1i to clarify this point.

Methods should be written concisely, but should contain all elements necessary to allow interpretation and replication of the results. As a guideline, Methods sections typically do not exceed 3,000 words. The Methods should be divided into subsections listing reagents and techniques. When citing previous methods, accurate references should be provided and any alterations should be noted. Information must be provided about: antibody dilutions, company names, catalogue numbers and clone numbers for monoclonal antibodies; sequences of RNAi and cDNA probes/primers or company names and catalogue numbers if reagents are commercial; cell line names, sources and information on cell line identity and authentication. Animal studies and experiments involving human subjects must be reported in detail, identifying the committees approving the protocols. For studies involving human subjects/samples, a statement must be included confirming that informed consent was obtained. Statistical analyses and information on the reproducibility of experimental results should be provided in a section titled "Statistics and Reproducibility".

All Nature Cell Biology manuscripts submitted on or after March 21 2016 must include a Data availability statement at the end of the Methods section. For Springer Nature policies on data availability see <http://www.nature.com/authors/policies/availability.html>; for more information on this particular policy see <http://www.nature.com/authors/policies/data/data-availability-statements-data-citations.pdf>. The Data availability statement should include:

- Accession codes for primary datasets (generated during the study under consideration and designated as "primary accessions") and secondary datasets (published datasets reanalysed during the study under consideration, designated as "referenced accessions"). For primary accessions data should be made public to coincide with publication of the manuscript. A list of data types for which submission to community-endorsed public repositories is mandated (including sequence, structure, microarray, deep sequencing data) can be found here <http://www.nature.com/authors/policies/availability.html#data>.
- Unique identifiers (accession codes, DOIs or other unique persistent identifier) and hyperlinks for datasets deposited in an approved repository, but for which data deposition is not mandated (see here for details <http://www.nature.com/sdata/data-policies/repositories>).
- At a minimum, please include a statement confirming that all relevant data are available from the authors, and/or are included with the manuscript (e.g. as source data or supplementary information), listing which data are included (e.g. by figure panels and data types) and mentioning any restrictions on availability.
- If a dataset has a Digital Object Identifier (DOI) as its unique identifier, we strongly encourage including this in the Reference list and citing the dataset in the Methods.

We recommend that you upload the step-by-step protocols used in this manuscript to [protocols.io](https://www.protocols.io). More details can be found at <https://www.protocols.io/help/publish-articles>.

All imaging data should be accompanied by scale bars, which should be defined in the legend.

Cropped images of gels/blots are acceptable, but need to be accompanied by size markers, and to retain visible background signal within the linear range (i.e. should not be saturated). The boundaries of panels with low background have to be demarked with black lines. Splicing of panels should only be considered if unavoidable, and must be clearly marked on the figure, and noted in the legend with a statement on whether the samples were obtained and processed simultaneously. Quantitative comparisons between samples on different gels/blots are discouraged; if this is unavoidable, it should only be performed for samples derived from the same experiment with

gels/blots were processed in parallel, which needs to be stated in the legend.

- For line art, graphs, charts and schematics we prefer Adobe Illustrator (.AI), Encapsulated PostScript (.EPS) or Portable Document Format (.PDF). Files should be saved or exported as such directly from the application in which they were made, to allow us to restyle them according to our journal house style.
- We accept PowerPoint (.PPT) files if they are fully editable. However, please refrain from adding PowerPoint graphical effects to objects, as this results in them outputting poor quality raster art. Text used for PowerPoint figures should be Helvetica (preferred) or Arial.
- We do not recommend using Adobe Photoshop for designing figures, but we can accept Photoshop generated (.PSD or .TIFF) files only if each element included in the figure (text, labels, pictures, graphs, arrows and scale bars) are on separate layers. All text should be editable in 'type layers' and line-art such as graphs and other simple schematics should be preserved and embedded within 'vector smart objects' - not flattened raster/bitmap graphics.
- Some programs can generate Postscript by 'printing to file' (found in the Print dialogue). If using an application not listed above, save the file in PostScript format or email our Art Editor, Allen Beattie for advice (a.beattie@nature.com).

The total number of Supplementary Figures (not including the "unprocessed scans" Supplementary Figure) should not exceed the number of main display items (figures and/or tables (see our Guide to Authors and March 2012 editorial <http://www.nature.com/ncb/authors/submit/index.html#suppinfo>; <http://www.nature.com/ncb/journal/v14/n3/index.html#ed>). No restrictions apply to Supplementary Tables or Videos, but we advise authors to be selective in including supplemental data.

GUIDELINES FOR EXPERIMENTAL AND STATISTICAL REPORTING

REPORTING REQUIREMENTS – To improve the quality of methods and statistics reporting in our papers we have recently revised the reporting checklist we introduced in 2013. We are now asking all life sciences authors to complete two items: an Editorial Policy Checklist (found here https://www.nature.com/authors/policies/Policy.pdf) that verifies compliance with all required editorial policies and a reporting summary (found here https://www.nature.com/authors/policies/ReportingSummary.pdf) that collects information on experimental design and reagents. These documents are available to referees to aid the evaluation of the manuscript. Please note that these forms are dynamic 'smart pdfs' and must therefore be downloaded and completed in Adobe Reader. We will then flatten them for ease of use by the reviewers. If you would like to reference the guidance text as you complete the template, please access these flattened versions at http://www.nature.com/authors/policies/availability.html.

Version 1:

Decision Letter:

Our ref: NCB-A54744A

29th January 2025

Dear Dr. Reber,

Thank you for submitting your revised manuscript "Cell State-Specific Cytoplasmic Material Properties Control Spindle Architecture and Scaling" (NCB-A54744A). It has now been seen by the original referees and their comments are below. The reviewers find that the paper has improved in revision, and therefore we'll be happy in principle to publish it in Nature Cell Biology, pending minor revisions to satisfy the referees' final requests and to comply with our editorial and formatting guidelines.

Thank you again for your interest in Nature Cell Biology Please do not hesitate to contact me if you have any questions.

Sincerely,
Daryl

Daryl Jason Verzosa David, PhD

Senior Editor, Nature Cell Biology
Advisory Editor, npj Biological Physics and Mechanics
Nature Portfolio

Heidelberger Platz 3, 14197 Berlin, Germany
Email: daryl.david@nature.com
ORCID: <https://orcid.org/0000-0002-9253-4805>

Reviewer #1 (Remarks to the Author):

The authors have addressed all of my concerns and I believe this manuscript will be highly cited.

Reviewer #2 (Remarks to the Author):

The author have answered all the points I raised. I am happy to recommend this paper for publication, as it is both exciting and well executed!

Reviewer #3 (Remarks to the Author):

The authors have addressed most of my initial concerns; by incorporating many important new controls and quantifications. Especially, the new data presented in Ext Data Figure 8 provide a more rigorous assessment of the role of cytoplasm density on spindle size scaling in ES vs Differentiated cells.

I just have one minor final comment: The title uses "Cytosplamic Material Properties" but these properties are not being measured in this work. I would suggest to use "Cytoplasm Density" which better reflects the content of the paper; and does not take out anything from this very nice work.

Version 2:

Decision Letter:

Dear Dr Reber,

I am pleased to inform you that your manuscript, "Cell State-Specific Cytoplasmic Density Controls Spindle Architecture and Scaling", has now been accepted for publication in *Nature Cell Biology*.

Over the next few weeks, your paper will be copyedited to ensure that it conforms to *Nature Cell Biology* style. Once your paper is typeset, you will receive an email with a link to choose the appropriate publishing options for your paper and our Author Services team will be in touch regarding any additional information that may be required.

Publication is conditional on the manuscript not being published elsewhere and on there being no announcement of this work to any media outlet until the online publication date in *Nature Cell Biology*.

Please note that *Nature Cell Biology* is a Transformative Journal (TJ). Authors may publish their research with us through the traditional subscription access route or make their paper immediately open access through payment of an article-processing charge (APC). Authors will not be required to make a final decision about access to their article until it has been accepted. [Find out more about Transformative Journals](https://www.springernature.com/gp/open-research/transformative-journals)

If you have not already done so, we strongly recommend that you upload the step-by-step protocols used in this manuscript to protocols.io (<https://protocols.io>), an open online resource that allows researchers to share their detailed experimental know-how. All uploaded protocols are made freely available and are assigned DOIs for ease of citation. Protocols and Nature Portfolio journal papers in which they are used can be linked to one another, and this link is clearly and prominently visible in the online versions of both. Authors who performed the specific experiments can act as primary authors for the Protocol as they will be best placed to share the methodology details, but the Corresponding Author of the present research paper should be included as one of the authors. By uploading your Protocols onto protocols.io, you are enabling researchers to more readily reproduce or adapt the methodology you use, as well as increasing the visibility of your protocols and papers. You can also establish a dedicated workspace to collect your Lab Protocols. Further information can be found at <https://www.protocols.io/help/publish-articles>.

Nature Cell Biology encourages authors presenting evidence for cell, biological, molecular, and genetic interactions to consider communicating these findings using Biofactoid (<https://biofactoid.org/>). This tool helps users share a searchable representation of interactions (e.g. binding, gene expression, post-translational modification) between genes, gene products, or chemicals. Information added to Biofactoid, with author attribution, is shared on social media and public databases, such as Pathway Commons, where it can be discovered and analyzed in the context of a large and growing corpus of knowledge.

With kind regards,
Daryl

Daryl Jason Verzosa David, PhD

Senior Editor, Nature Cell Biology
Advisory Editor, npj Biological Physics and Mechanics
Nature Portfolio

Heidelberger Platz 3, 14197 Berlin, Germany
Email: daryl.david@nature.com
ORCID: <https://orcid.org/0000-0002-9253-4805>

** Visit the Springer Nature Editorial and Publishing website at http://editorial-jobs.springernature.com?utm_source=ejp_NCB_email&utm_medium=ejp_NCB_email&utm_campaign=ejp_NCB for more information about our career opportunities. If you have any questions please click [here](mailto:editorial.publishing.jobs@springernature.com).

Reviewers' Comments:

Reviewer #1

Remarks to the Author:

Kletter et al.

The authors set out to investigate mitotic spindle morphology during differentiation of neural stem cells and utilize a powerful long-term microscopy method as well as microtubule dynamics measurements, phase imaging, and quantitative biochemistry. They reveal a mechanism by which spindles shift in size and morphology as the cytoplasm becomes more dilute and identify a protein that contributes to the shift in microtubule nucleation from the spindle bulk to centrosomes.

This is an exceptionally nice paper. The observations are interesting, many different types of high quality data are presented and rigorously analyzed. The CPAP mechanism elucidated is convincing and of high significance. I believe this will be a landmark paper in the field. My comments on the text are minor.

We thank reviewer #1 for a careful reading of our manuscript and enthusiastic comments. We have addressed all of them (see below).

It should be stated somewhere in the main text how cell volumes were measured.

We now briefly state in the main text how we measure spindle and cell volumes (lines 130-134). In addition, we included a visual representation of our analysis rationale as Extended Data Fig. 2a.

Line 174: it would be useful for many readers to indicate that CKAP5 belongs to the XMAP215/TOG family. Is it neural-specific?

CKAP5 is the mouse gene that encodes for the XMAP215 homologue, we now specify this, see lines 166-168 of the revised manuscript. We now also include data on CKAP2, another microtubule polymerase (McAlear & Bechstedt 2022), and show that neither CKAP2 concentration nor localisation changes during early differentiation. See new experimental data in response to minor point #2 of reviewer #2 below and new Extended Data Fig. 3b-d of the revised manuscript.

While it would be difficult to measure other parameters of microtubule dynamics, is it predicted that catastrophe frequency increases?

To assess more parameters of microtubule dynamics, we now measured microtubule turnover via fluorescence recovery after photobleaching (FRAP), see new experimental data in response to major point #2 of reviewer #3 below and new man Fig. 2a-c of the revised manuscript. These measurements confirm our initial data that microtubule dynamics / turnover do not significantly

change during the first days of early neural differentiation and that total microtubule mass is conserved.

Line 216: perhaps “relative volume occupied by centrosomal proteins”

Agreed and changed accordingly.

Line 232-233: “persisting centrosomes from large cells” is unclear and unrelated to the previous sentence. Was the question whether the cells were about to enter mitosis?

As cells progress through the cell cycle, the centrosome is actively remodeled with centrioles being duplicated and centriolar protein composition being modified through protein recruitment and removal. The only point we wanted to convey here is that cells with enlarged centrosomes have dynamically remodelled them through more than one cell cycle. We now write (lines 220-224): “Commonly, to prepare for mitosis, microtubule remodelling starts with a dramatic increase in microtubule nucleation at the centrosomes driven by the recruitment and local activation of γ -TuRC (Piehl et al., 2004; Liu et al., 2020). Our long-term imaging strategy allowed us to track individual cells across several generations (Fig. 3f). This showed that cells classified as early-differentiated cells had on average divided 3 to 4 times (Fig. 3g) and thus dynamically remodelled their PCM.”

Line 271 and 290: May be a bit confusing, need to clearly distinguish microtubule density within the spindle bulk and “at the expense of spindle bulk”. The size decreases but not the microtubule density.

We thank the reviewer for this hint. We rephrased these lines, which now read:

Now lines 256-259: “This indicated a drop of total tubulin concentration concomitant with cytoplasmic dilution. While the reduction in total tubulin reduced spindle volume, it did not lead to a reduction in microtubule density within the spindle bulk (see Extended Data Figs 2j, 3a).”

and

Now lines 276-279: “Taken together, these data confirmed that a reduction in cellular mass density is sufficient to increase the centrosome’s nucleation capacity at the expense of the spindle bulk (Fig. 6j), resulting in a reduced spindle volume with constant microtubule density.”

Reviewer #2

Remarks to the Author:

In this paper, Kletter and colleagues investigate the changes in spindle scaling during ESC differentiation towards the neural lineage. They find that spindle size scales with cell size, in accordance with the literature in other model systems, but that the relative spindle size/cell size changes during differentiation. They show convincingly that during differentiation nucleation of astral microtubules increases at the centrosome at the expense of the spindle bulk, due to a general dilution of the cytoplasm. They show that diluting the cytoplasm and freeing CPAP from tubulin both phenocopy differentiation in terms of spindle scaling and nucleation distribution.

This paper is very elegant and nicely written, the figures are clear and beautiful. The question is interesting, and the approach is solid. It opens many fascinating questions that are outside of the scope of the paper, such as: why is the cytoplasm diluted in differentiated cells? Is it more important for a differentiated cell to have astral microtubules (maybe for spindle positioning)? Overall, I think the paper proposes a new mechanism for spindle scaling in a developmental context, and that the data presented is strong.

I have a couple of comments/questions which I think would make the paper even stronger, but overall I am happy to recommend this paper for publication.

We thank reviewer #2 for a careful assessment of our work. We have addressed all comments (see below), which certainly improved the quality of the manuscript.

Major comments:

Several piece of crucial data relies on overexpression of markers. To be fully convincing, I think the authors should consider validating some of their finding using other labelling methods. For example, I think the key piece of data from Fig1 could be easily replicated using a microtubule dye eg SPY or SiR-tubulin. Similarly, Fig2a would benefit from being repeated with an endogenous tagging of EB3, although I realize this is a lot more challenging thus should not condition the acceptance of the paper.

Importantly, we would like to emphasize that markers are NOT overexpressed, our stable cell lines are bacterial artificial chromosomes (BAC) cell lines, which express the respective proteins under their endogenous promoters. In particular, the tubulin cell line is stably transfected with BAC harbouring the eGFP-fused coding region of human \$\beta\$ 5-tubulin and its regulatory sequences for native expression levels (Poser et al., 2008) This information is provided in the Material and Methods section "Stem cell culture and differentiation", lines 606-638.

I do not understand what figure 3f-g bring. How does the fact that cell cycle duration changes during differentiation (also shown by doi: 10.1038/s41598-019-44537-0) demonstrates that the enlarged centrosomes are not remnants from larger stem cells. I am also not totally sure what a "persisting centrosome from larger stem cells" would be: are the authors implying that this cell would either not have divided at all/this would be a non differentiated ESC in a population of DIF cells ? Could the authors clarify what they hypothesis is here?

As cells progress through the cell cycle, the centrosome is actively remodeled with centrioles being duplicated and centriolar protein composition being modified through protein recruitment

and removal. The only point we wanted to convey here is that cells with enlarged centrosomes have dynamically remodelled them through more than one cell cycle. We now write (lines 220-224): “Commonly, to prepare for mitosis, microtubule remodelling starts with a dramatic increase in microtubule nucleation at the centrosomes driven by the recruitment and local activation of γ -TuRC (Piehl et al., 2004; Liu et al., 2020). Our long-term imaging strategy allowed us to track individual cells across several generations (Fig. 3f). This showed that cells classified as early-differentiated cells had on average divided 3 to 4 times (Fig. 3g) and thus dynamically remodelled their PCM.”

We also cite Waisman et al. 2019.

The authors look at microtubule severing using Katanin. They report differences between relative amounts of p80 between ESCs and DIF. What about p60, are there differences in relative amounts? Also, to conclude on the lack of role of microtubule severing, the authors should also look at Spastin (the authors could use a similar approach with siRNA or a Spastin inhibitor such as the one described here doi:10.1038/s41589-019-0225-6)

We thank the reviewer for this hint. The old version of the manuscript already included data on Katanin p60, which we found to co-deplete during RNAi of KATNB1 (p80), further substantiating our conclusion that MT severing does not explain spindle subscaling in our system. These data, however, were hidden in Extended Data Figure 5 and not explicitly mentioned in the main text. To complement our previous results, we now performed an analysis of Katanin p60 localisation during differentiation (see below and new Extended Data Figure 5c, e).

Katanin p60 loading at spindle poles is stable across differentiation.

a) Confocal micrographs (maximum-projected) of PFA-fixed undifferentiated ESCs or early differentiated cells at metaphase. Top: Tubulin::GFP, center: chromatin (Hoechst), bottom: Katanin p60 staining signal. Dotted lines indicate the cell boundaries. Scale bar: 5 μm.

b) Relative Katanin p60 signal on spindle poles (normalized by total cell katanin p60 fluorescence). Each data point represents a single cell (ESCs n = 37 and differentiation n = 38, data from N = 3 experiments), the large circles denote the medians in each cell volume bin, the error bars show the interquartile ranges.

As suggested, we now also tested for Spastin levels. While the Spastin antibody unfortunately did not work for immunofluorescence, the Western Blot data suggest that Spastin levels do not change during early differentiation. These data are now included as Extended Data Fig. 5a.

Microtubule severing protein Spastin levels are unchanged upon differentiation.

Cellular levels of KIF2C/MCAK after 48 h of differentiation relative to the undifferentiated ESCs, probed by western blotting and normalized to GAPDH (N = 2, each time loading 2 independent batches of protein extracts). Bars show the mean, errors show the standard deviation. Significance tested by Welch's t-test, n.s.: not significant, $p > 0.05$.

As far as I can tell, the authors did not show that diluting the cytoplasm actually leads to liberation of CPAP in this model system, this is solely based on literature and the fact that the 2 phenotypes phenocopy. I think this is fine, but the title of figure 6 should probably be toned down accordingly, as well as some aspects of the discussion (eg line 376).

We agree with reviewer #2. We now added new data, which provides further evidence that cytoplasmic density modulates spindle size (for details see major comment #2 of reviewer #3). In addition, we more carefully word the title of the paragraph and Figure 6, which now reads "Cytoplasmic dilution shifts spindle architecture by increasing centrosomal nucleation capacity".

Minor comments:

1. The authors see that cell volume is higher in ESC than their differentiated counterparts. However, the general belief is that differentiated cells are overall bigger than stem cells. Does this mean that stem cells swell more during mitosis than differentiated cells, or that this idea is a mere cliché, or perhaps due to previous non accurate measurements in 3D vs 2D? (this does not need to be added to the paper, I would just be curious to hear the authors' opinion).

We would not want to generalize our findings to "differentiated cells are overall bigger than stem cells". But in the case of our differentiation system, we clearly see that mESCs have a higher mitotic volume than differentiating cells (observed up to 6 days after induction of differentiation). Indeed, we have meanwhile measured volumes and densities of mitotic and corresponding interphase cells and see that mitotic swelling (and mitotic cytoplasmic dilution) are very much cell type dependent. This is a topic of current research in our lab.

2. I am not sure why the authors focus on the microtubule polymerase CKAP5 specifically. Could the authors justify this choice, or have a more systematic approach?

CKAP5 is the mouse homologue of XMAP215, the major microtubule polymerase, which has been shown to be a major determinant of spindle size in embryonic systems (Reber et al. 2013, Milunovic-Jevtic et al. 2018). We now also include data on CKAP2, another microtubule polymerase (McAlear & Bechstedt 2022). We find that neither CKAP2 concentration nor localisation changes during early differentiation. These data are now included as Extended Data Fig. 3b-d.

Microtubule polymerase CKAP2 levels and localization do not change between the differentiation states.

a) Cellular levels of CKAP2 after 48 h of differentiation relative to the undifferentiated ESCs, probed by western blotting and normalized to GAPDH (N = 3, each time loading 2 independent batches of protein extracts). Bars show the mean, errors show the standard deviation. Significance tested by Welch's t-test, n.s.: $p > 0.05$.

b) Confocal micrographs (maximum-projected) of methanol-fixed undifferentiated ESCs or early differentiated cells at metaphase. Top: Immunostained CKAP2 signal, bottom: Tubulin::GFP (grey) and chromatin counterstained by Hoechst (blue). Dotted lines indicate the cell boundaries. Scale bar: 5 μm .

c) Fluorescent CKAP2 signal on spindle (normalized by total cell CKAP2 fluorescence) summed, as a function of spindle volume. Each data point represents a single cell (ESCs n = 23 and differentiation n = 21, data from N = 2 experiments), the large circles denote the medians in each cell volume bin, the error bars show the interquartile ranges. Note that spindle volumes in general are smaller because of methanol fixation.

In supplementary 5i,j, should the authors be looking at scaling rather than volume?

Extended Data Fig. 5m (previously suppl. Fig. 5i) actually shows scaling as we plot spindle volume as % of cell volume (V_{cell}). This was not explicitly alluded to in the text, which we now changed.

Reviewer #3:

Remarks to the Author:

Kletter et al., 2024

Mitotic spindles vary significantly in size and morphology, often adapting to the specific size and function of different cell types. However, the biological mechanisms that drive these morphological adaptations remain poorly understood. In this manuscript, Kletter and colleagues explore spindle size scaling in mouse embryonic stem cells (ESCs) undergoing neural differentiation. One chief finding is that spindles are smaller in early differentiated (DIF) cells compared to pluripotent stem cells, with DIF cells exhibiting larger centrosomes and more abundant astral microtubules (MTs). Using optical diffraction tomography, the authors demonstrate that the cytoplasm of neurally differentiated cells is more dilute, resulting in lower free tubulin concentration. This decrease in tubulin may release CPAP from its binding to free tubulin, promoting centrosome growth and spindle scaling. Therefore, the study's significance lies in its proposal that cytoplasmic dilution serves as a key regulator of organelle size scaling during stem cell differentiation.

Overall, the manuscript is well-executed, incorporating several innovative quantitative microscopy techniques, numerous controls for tubulin biochemistry and spindle regulators, and mathematical models that support the conclusions. The paper is also generally well-written and presented. However, this reviewer finds that the main claim—that "cytoplasmic material properties control spindle architecture and scaling"—should be better substantiated before publication. Below are some major and minor concerns:

We thank reviewer #3 for their positive assessment of our work. Please see below for our answers to the specific comments.

1. Spindle "Size" Definition

The authors use 3D volume to assess spindle size scaling. The rationale for this choice, as well as the definition of volume, should be clarified. Visual inspection suggests that spindles in DIF cells are wider and shorter, with differences in MT distribution compared to ESCs. Previous studies on spindle scaling have often used pole-to-pole length or chromosome plate width to measure spindle morphology, which should be quantified here as well. Given that spindles are porous structures with visible voids, the meaning of spindle volume remains ambiguous—does it represent the number of MTs, their mass, etc.? Additionally, why were astral MTs not included as part of spindle volume measurement?

We thank the reviewer for this important comment. As suggested, we now provide definitions and measurements on spindle pole-to-pole length, spindle width, and spindle aspect ratio (see below and new Extended Data Fig. 2a-d). Upon differentiation, both spindle length and width drop together with cell volume and confirm the spindle subscaling phenotype that we described using spindle volume. The aspect ratio does slightly change upon differentiation, as the reviewer pointed out after visual inspection. Indeed, spindle length decreases more significantly than spindle width (see below). We interpret these data in a way that for spindle scaling, spindle width is very much restricted to maintain spindle function, i.e. accommodate all chromosomes at the metaphase

plate, while spindle length has more degrees of freedom. To sustain this, we now also provide measurements of chromatin volume, metaphase plate width and length (see below). Overall, chromatin volume stays relatively constant. Thus, in the differentiating population, subscale spindles need to accommodate comparably large metaphase plates, justifying changes in spindle aspect ratio when compared to the undifferentiated cells. Interestingly, we find the trend of decreasing aspect ratios recapitulated when we biophysically dilute the cytoplasm in undifferentiated stem cells (see below and new Extended Data Fig.8 f).

Spindle subscale during differentiation encompasses changes of spindle length and width.

a) We define spindle length (SL) as the pole-to-pole distance.

- b)** Globally, SL decreases upon differentiation. Boxes denote the interquartile ranges, horizontal lines inside the boxes show the medians, whiskers indicate the extrema. Data points show individual cells (ESCs $n = 1088$, DIF $n = 2921$). Welch's t-test, ****: $p < 0.0001$.
- c)** SL after binning the data into cell volume bins (bin size $500 \mu\text{m}^3$). Numbers inside the bins show the respective n of measurements. Boxes denote the interquartile ranges, horizontal lines inside the boxes show the medians, whiskers indicate the extrema. Welch's t-test, *: $p < 0.05$, **: $p < 0.001$, ****: $p < 0.0001$.
- d)** We define spindle width (SW) as the average diameter of the spindle at the equator.
- e)** As in b) but showing SW.
- f)** As in c) but showing SW. n.s.: not significant, $p > 0.05$.
- g)** We define the spindle aspect ratio as SL/SW .
- h)** As in b) but showing the spindle aspect ratio.
- i)** As in c) but showing the spindle aspect ratio.

Spindle volume is derived by finding a threshold above which tubulin voxels are incorporated into the spindle volume mask (see updated Materials and Methods part and a detailed technical description in Kletter et al. 2021). Importantly, void regions are excluded and spindle volume thus effectively equals spindle microtubule / tubulin mass. A visual representation of our analysis rationale can now be found in Extended Data Fig. 2a (see below). Taken together, the visualization of microtubules in the spindle bulk volume and the independent measure of astral microtubule number is the most robust and impactful to convey our main message, i.e. that during differentiation spindle volume reduces at constant microtubule density with a concomitant increase in centrosomal nucleation activity as evidenced by an increase in astral microtubule numbers.

Mitotic chromatin geometry changes during differentiation.

- a)** Confocal images (single slices) of mitotic cells showing Tubulin::GFP signals or chromatin labelled with SiR-DNA. Dotted lines indicate cell boundaries, straight lines show location of the coverslip. Scale bars: 5 μm .
- b)** Spindle volumes within chromatin volume bins (bin size = 50 μm^3). Boxes (yellow: ESCs, blue: DIF) denote the interquartile ranges, horizontal lines inside the boxes show the medians, whiskers indicate the extrema. Numbers inside the bins show the respective n of measurements. d: Cohen's d effect size. Welch's t-test, ****: $p < 0.0001$.
- c)** Chromatin volume within differentiation time bins (bin size = 48 h). Circles show the mean, error bars show the standard deviation. Numbers inside the bins show the respective n of measurements. d: Cohen's d effect size. Welch's t-test, *: $p < 0.05$. n.s.: not significant, $p > 0.05$.
- d)** As in c) but showing metaphase plate length (extent perpendicular to the spindle axis).**, $p < 0.01$, ****: $p < 0.0001$.
- e)** As in c) but showing the width (extent along the spindle axis) of the metaphase plate.

Visual representation of volumetric segmentation of mitotic cells expressing Tubulin::GFP.

Volumetric cell analysis, based on training pixel-classifier models in *Ilastik* (Berg et al. 2019) to distinguish mitotic cytoplasm via soluble tubulin::GFP, probability prediction and thresholding and manual validation of the probability masks. In parallel, spindle volume masks were generated and analyzed via *Spindle3D* (Kletter et al. 2021), based on adaptive thresholding of tubulin::GFP.

2. Microtubule Regulation

The authors show that centrosomes are larger in DIF cells, potentially promoting astral MT nucleation at the expense of bulk MTs, thus reducing spindle size. However, it is unclear why larger centrosomes would only promote outward nucleation rather than inward. Moreover, the relationship between reduced bulk MTs and the formation of shorter, wider spindles needs

clarification. The nucleation of MTs from chromosomes is another significant pathway, but its role in spindle scaling during differentiation is not examined. Also, while the authors note that MT growth speed is similar in ESCs and DIF cells, other aspects of MT dynamics, such as depolymerization speeds or catastrophe frequency, may play an important role. Finally, several depolymerizing kinesins (e.g., MCAK, Kif18b) that are known to influence spindle size are not tested here.

We agree with reviewer #3. To clarify the relationship between reduced bulk microtubules and the formation of shorter, wider spindles, we now include the spindle width and aspect ratio data (see comment above). Furthermore, to better assess nucleation of microtubules from chromosomes versus astral nucleation, we - in addition to measuring localisation and concentration of TPX2 and Augmin, both regulated by the chromatin-dependent Ran pathway - now assess the localisation and concentration of MCAK, a major microtubule depolymerase. While we see a slight reduction in MCAK concentration in differentiating cells by Western Blot, its concentration on centrosomes did not change (see below and new Extended Data Fig. 3f-h).

Microtubule depolymerase KIF2C/MCAK levels decrease upon differentiation, but spindle localisation is maintained.

a) Cellular levels of KIF2C/MCAK after 48 h of differentiation relative to the undifferentiated ESCs, probed by western blotting and normalised to tubulin (N = 3, each time loading 2 independent batches of protein extracts). Bars show the mean, errors show the standard deviation. Significance tested by Welch's t-test, *: p < 0.05.

b) Confocal micrographs (maximum-projected) of methanol-fixed undifferentiated ESCs or early differentiated cells at metaphase. Top: Immunostained KIF2C/MCAK signal, bottom: Tubulin::GFP (grey) and chromatin counterstained by Hoechst (blue). Dotted lines indicate the cell boundaries. Scale bar: 5 μ m.

c) Fluorescent KIF2C/MCAK signal on spindle (normalised by total cell KIF2C/MCAK fluorescence) summed, as a function of spindle volume. Each data point represents a single cell (ESCs n = 52 and differentiation n = 74, data from N = 2 experiments), the large circles denote the medians in each cell volume bin, the error bars show the interquartile ranges. Note that spindle volumes in general are smaller because of methanol fixation.

Finally, we assess turnover of spindle microtubules via fluorescence recovery after photobleaching (FRAP) measurements. We found the half-time of microtubule recovery in both spindle types to be similar and comparable to previous measurements (Walczak et al. 2010): 11.1 ± 4.6 s (n = 31) in stem cells and 9.7 ± 5.8 s (n = 31) in differentiating cells (see data below, included as new Fig. 2a-c). Taken together, these measurements confirm our initial data that microtubule dynamics and turnover do not significantly change during the first days of early neural differentiation and that total microtubule mass is conserved.

Spindle microtubule turnover is maintained between the differentiation states.

a) FRAP analysis of Tubulin::GFP turnover in mitotic spindles. Selected frames pre-bleach and post-bleach are shown. Regular photobleaching during the acquisitions was corrected by the signals within the control region of interest (ROI). Scale bar: 5 μ m.

b) Normalized recovery curves, lines show the mean (ESCs n = 31, DIF n = 31 from 2 independent experiments), bands show the standard deviation.

c) Recovery halftimes of Tubulin::GFP, large circles show the means, error bars indicate the standard deviation, small circles show individual cells. Significance tested by Welch's t-test, $p = 0.31$.

3. Cytoplasmic Dilution

Figure 6 is a central figure, as it aims to demonstrate that cytoplasmic dilution affects spindle size by influencing CPAP's impact on centrosome size. This aspect of the study should be expanded to provide more robust support for the paper's main claim. First, the authors should test a range of hypoosmotic shocks to determine whether spindle size declines linearly with cytoplasm concentration or exhibits a saturating behavior. Second, hyperosmotic shocks should be applied to concentrate the cytoplasm in DIF cells to see if this rescues spindle size. Third, to directly validate their model, the authors should attempt to reverse the effects of hypoosmotic shock by overexpressing tubulin and/or downregulating CPAP. Indeed, it is plausible that CPAP manipulation (via CCB02) and cytoplasmic dilution influence spindle size by very separate mechanisms, so further experiments are needed. Additionally, examining CPAP localization under different conditions could strengthen the proposed model.

We thank reviewer #3 for this critical comment. Unfortunately, we were not able to overexpress tubulin. But, as suggested, we now tested a range of osmolalities above and below the physiological range resulting in the effective dilution and concentration of the cytoplasm. Indeed, spindle size correlates with cytoplasmic density in both stem cells and neural progenitors (see

below and new Extended Data Fig. 8). We do not observe saturating behaviors, rather breaking points at which spindle assembly and / or mitosis fails. Furthermore, we now show that the phenotypes indeed result from a reduction in spindle size rather than an increase in cell size for all perturbation experiments (see new data replying to your comment #6 below). In addition, we tone down the title of the paragraph and of figure 6, which now reads “Cytoplasmic dilution shifts spindle architecture by increasing centrosomal nucleation capacity”.

Biophysical perturbation of cellular mass density modulates spindle scaling.

a) Hypoosmotic treatment of undifferentiated ESCs for live optical diffraction tomography (ODT) or confocal imaging.

b) ODT-derived cellular mass densities after hypoosmotic treatment (left: 80/20, right: 75/25, % medium/H₂O) of ESCs. Boxes show the interquartile ranges, horizontal lines show the medians, whiskers show the extrema. Data points show individual cells (80/20: n isosmotic: 52, n hypoosmotic: 51 from N = 3 experiments, 75/25: n isosmotic: 107, n hypoosmotic: 96 from N = 4 experiments). Significance tested by Welch's t-test, *: p < 0.05, ****: p < 0.0001.

c) Confocal live cell imaging (central slices) of control ESCs (isosmotic treatment) versus increasing hypoosmotic challenges. Tubulin::GFP in grayscale, chromatin in blue. Dotted lines show cell boundaries. Scale bar: 5 µm.

d) Cell volume-normalised spindle volumes during hypoosmotic challenge of ESCs, fold-change relative to control ESCs (Iso). Boxes show the interquartile ranges, horizontal lines show the medians, whiskers show the extrema. Data points

show individual cells (pooled from N = 5 experiments). Significances tested by Welch's ANOVA and Games-Howell post-hoc testing. ****: $p < 0.0001$, n.s. $p > 0.05$.

e) Data as in d), showing spindle volumes within cell volume bins. Circles show the means, error bars show the standard deviation. Horizontal gray lines indicate the mean value of the control ESCs (Iso).

f) As in d) but showing spindle aspect ratios (spindle length / spindle width).

g) Hyperosmotic treatment of early-differentiated cells for live optical diffraction tomography (ODT) or confocal imaging.

h) ODT-derived cellular mass densities after hyperosmotic treatment (left: +50 mM Sorbitol, right: +100 mM Sorbitol) of early-differentiated cells, untreated ESCs as additional control. Boxes show the interquartile ranges, horizontal lines show the medians, whiskers show the extrema. Data points show individual cells (+50 mM Sorbitol: n ESCs: 12 n, isosmotic DIF: 13, n hyperosmotic DIF: 13 from N = 1 experiment). Significances tested by Welch's ANOVA and Games-Howell post-hoc testing. **: $p < 0.01$, n.s. $p > 0.05$.

i) Cell volume-normalised spindle volumes during hyperosmotic challenge of DIF, fold-change relative to undifferentiated control ESCs). Boxes show the interquartile ranges, horizontal lines show the medians, whiskers show the extrema. Data points show individual cells (pooled from N = 2 experiments). Significances tested by Welch's ANOVA and Games-Howell post-hoc testing. ****: $p < 0.0001$, n.s. $p > 0.05$.

j) Composite chart showing the cell volume-normalised spindle volumes (as fold changes relative to controls) as a function of the mean cellular mass density (as fold change relative to controls). Circles show the mean, error bars show the standard error of the mean. Data merged from this figure and Figure 6.

Mathematical Model

The mathematical model presented is based on a Michaelis-Menten kinetics to explain the differences in astral MT numbers between ESCs and DIF cells, drawing on prior work by Rieckhoff et al. However, Rieckhoff's model applied this law to all MTs, not just astral MTs. Here, the authors assume that the total number of MTs remains constant, which is surprising given their claim that free tubulin concentration decreases in differentiated cells. Moreover, the potential impact of cytoplasm dilution on diffusion rates of tubulin and CPAP, as discussed in Molines et al. (2022), is not addressed here.

As in Rieckhoff et al., in our model the total number of microtubules follows the Michaelis-Menten law. This is because the number of bulk microtubules is proportional to the number of astral microtubules (equation (4) in the Supplementary Note). This respective proportionality was obtained experimentally (Fig. 2i). Thus, while we consider the number of astral and bulk microtubules separately (line 31 of Supplementary Note), their sum is the total microtubule number. Consequently, the sum of astral and bulk microtubules in our model naturally follows a Michaelis-Menten law.

We agree that the total number of microtubules remaining constant is surprising given that free tubulin concentration decreases. While we have no experimental evidence for this, our current working hypothesis is that local concentrations in the spindle and at the centrosomes define microtubule nucleation and not global cytoplasmic tubulin concentration. How much these local concentrations change upon global changes and how robust they remain is an exciting question for future research.

On the effects on diffusion: First, we would like to mention that the *S. pombe* cytoplasm (as in Molines et al. 2022) is different from the cytoplasm of mESC. In our measurements of cytoplasmic densities (Biswas et al. doi: <https://doi.org/10.1101/2023.09.05.556409>), the cytoplasm of *S. pombe* has a molecular density of approx. 200 mg/mL while the cytoplasmic density of stem cells is only 120 mg/mL, where the effect of crowding on larger particles as GEMs is stronger than on small molecules such as tubulin. This is consistent with the measured effective diffusion coefficients and stated in the Molines' paper: "Interestingly, the effective diffusion coefficient of the 40-nm GEM in fission yeast ($0.3 \text{ mm}^2/\text{s}$) is (...) slower than that in mammalian

cells ($0.5 \text{ mm}^2/\text{s}$). The more important arguments in this respect, however, are that (1) tubulin in both cell types is well within its respective reaction-diffusion volume as evidenced in spindle volume scaling with cell volume in both cell types (see Fig. 1h, i) and; (2) the possible changes of CPAP diffusion or reaction rates in differentiated and undifferentiated cells are captured by the different values of the fitted reaction-diffusion volumes (see Supplementary Note). Therefore, while diffusion rates can change due to cytoplasmic dilution, diffusion is not limiting for spindle assembly reactions. We thus think that we can sufficiently capture the observed differences in the different cell types by just accounting for the changes in concentrations of proteins.

We now cite Molines et al. 2022 in the main text as well as in the Supplementary Note and added a more detailed discussion of this matter therein.

Minor Comments:

- Figure 2c-d: The region of interest (ROI) used to measure bulk EB1 signal should encompass the entire spindle width, as the images suggest that MTs are more concentrated at the spindle edges. The normalization in Figure 2d obscures the main effect, which is an increase in EB1 signal at centrosomes rather than a decrease in the spindle bulk. Please revise.

We changed the visualisation of these data and now represent them as % of EB1 fractions, see main Fig. 3g.

- Figure 4c: The presented blots appear to show increased detyrosination levels, contradicting the authors' claim of no change. Please provide quantification to support this conclusion.

We agree with the reviewer that the data shown in the Western Blot might be interpreted as such. But compared to the detyrosination status of bovine tubulin, these changes are probably insignificant, and they do not appear in our mass spectrometry analysis (see main Figure 4a). We conclude that even if there were small changes, we do not see microtubule dynamics and turnover changed en gros.

- Figure 5c: The quantification of tubulin-GFP covers the entire image, but it would be essential to separately document the levels of free tubulin versus polymerized tubulin.

In general, we very much agree with the reviewer. In the theory, the values assumed for the concentration of polymerised tubulin in the spindle bulk are taken from our measurements, which cannot discern which fraction of bulk tubulin is free and polymerised. Technically, it would be very difficult to differentiate free tubulin and polymerised microtubules both, in the spindle bulk but also very close to the centrosomes. However, our measurements of microtubule growth velocities and turnover as well as EB1 comet density imply that the ratio between free and polymerised tubulin does not change in a magnitude that affects microtubule dynamics. Moreover, a potential overestimate of polymerised microtubule numbers would affect both cell states and consequently would not translate into a difference in scaling behaviour.

- Figure 6j: The spindle volume is normalized by cell size, raising concerns that the observed hypoosmotic shock phenotypes may result from an increase in cell size rather than a reduction in spindle size. Please provide spindle size quantification similar to that in Figure 1i to clarify this point.

This is a valid concern. We now provide a more comprehensive analysis using various degrees of hypoosmotic challenges to find that in cells of comparable volume, spindles are smaller when the cytoplasm is diluted. The data are now included as Extended Data Figure 8e (see below).

At a given cell volume, spindle volumes subscale upon cytoplasmic dilution.

Spindle volumes within cell volume bins (bin size = 500 µm³) upon osmotic challenge. Circles show the means, error bars indicate the standard deviations. Horizontal lines indicate the mean spindle volume of the isotonic control within each bin. Numbers inside the bins show the respective n of measurements for isotonic control ESCs or ESCs treated with increasingly diluted media (percentages indicate the media concentrations relative to iso after dilution, data are pooled from 4 independent experiments).

NCB-A54744A

Reviewers' Comments:

Reviewer #1 (Remarks to the Author):

The authors have addressed all of my concerns and I believe this manuscript will be highly cited.

Thank you.

Reviewer #2 (Remarks to the Author):

The author have answered all the points I raised. I am happy to recommend this paper for publication, as it is both exciting and well executed!

Thank you.

Reviewer #3 (Remarks to the Author):

The authors have adressed most of my initial concerns; by incorporating many important new controls and quantifications. Especially, the new data presented in Ext Data Figure 8 provide a more rigorous assessment of the role of cytoplasm density on spindle size scaling in ES vs Differentated cells.

I just have one minor final comment: The title uses "Cytosplamic Material Properties" but these properties are not being measured in this work. I would suggest to use "Cytoplasm Density" which better reflects the content of the paper; and does not take out anything from this very nice work.

Thank you. We have now changed the title of the paper to "Cell State-Specific Cytoplasmic Density Controls Spindle Architecture and Scaling".